# Diffusion Transformer Captures Spatial-Temporal Dependencies: A Theory for Gaussian Process Data

**Hengyu Fu**[*]     **Zehao Dou**[†]     **Jiawei Guo**[‡]     **Mengdi Wang**[§]     **Minshuo Chen**[¶]

## Abstract

Diffusion Transformer, the backbone of Sora for video generation, successfully scales the capacity of diffusion models, pioneering new avenues for high-fidelity sequential data generation. Unlike static data such as images, sequential data consists of consecutive data frames indexed by time, exhibiting rich spatial and temporal dependencies. These dependencies represent the underlying dynamic model and are critical to validate the generated data. In this paper, we make the first theoretical step towards bridging diffusion transformers for capturing spatial-temporal dependencies. Specifically, we establish score approximation and distribution estimation guarantees of diffusion transformers for learning Gaussian process data with covariance functions of various decay patterns. We highlight how the spatial-temporal dependencies are captured and affect learning efficiency. Our study proposes a novel transformer approximation theory, where the transformer acts to unroll an algorithm. We support our theoretical results by numerical experiments, providing strong evidence that spatial-temporal dependencies are captured within attention layers, aligning with our approximation theory.

## 1 Introduction

Diffusion models have emerged as a powerful new technology for generative AI, which is widely adopted in computer vision and audio generation (Song and Ermon, 2019; Song et al., 2020; Ho et al., 2020; Zhang et al., 2023), sequential data modeling (Alcaraz and Strodthoff, 2022; Tashiro et al., 2021; Tian et al., 2023), reinforcement learning and control (Pearce et al., 2023; Hansen-Estruch et al., 2023; Zhu et al., 2023; Ding and Jin, 2023), as well as computational biology (Xu et al., 2022; Guo et al., 2023). The basic functionality of diffusion models is to generate new samples replicating essential characteristics in the training data.

Diffusion models generate new samples by sequentially transforming Gaussian white noise. Each step of the transformation is driven by a so-called "score function", which is parameterized by a neural network. In order to train the score neural network, diffusion models utilize a forward process to produce noise corrupted data and the score neural network attempts to remove the added noise. In early implementations of diffusion models, the score neural network is typically chosen as the U-Net (Ronneberger et al., 2015). Afterwards, a few works also demonstrate the capability of using transformers as a score neural network (Peebles and Xie, 2023; Wu et al., 2024; Bao et al., 2023). Throughout the paper, we adopt the terminology in (Peebles and Xie, 2023) to name diffusion models with transformers as diffusion transformers.

Recently, the astounding success of applying diffusion models in *dynamic (sequential) data*, including video generation (Gupta et al., 2023; Liu et al., 2024b) and financial data augmentation (Gao et al., 2024), strengthens the seemingly unlimited potentiality of diffusion models . These models advocate transformers over the traditional U-Net for parameterizing the score function. A high-level intuition is that video data comprises rich spatial and temporal dependencies, induced by the underlying dynamics of objects and background. For example, the movement of an object should be continuous along

---

[*]Work done at Stanford University. Email: `fhycz2018@gmail.com`

[†]Yale University. Email: `zehao.dou@yale.edu`

[‡]Work done at Northwestern University. Email: `guokaku666@gmail.com`

[§]Princeton University. Email: `mengdiw@princeton.edu`

[¶]Northwestern University. Email: `minshuo.chen@northwestern.edu`

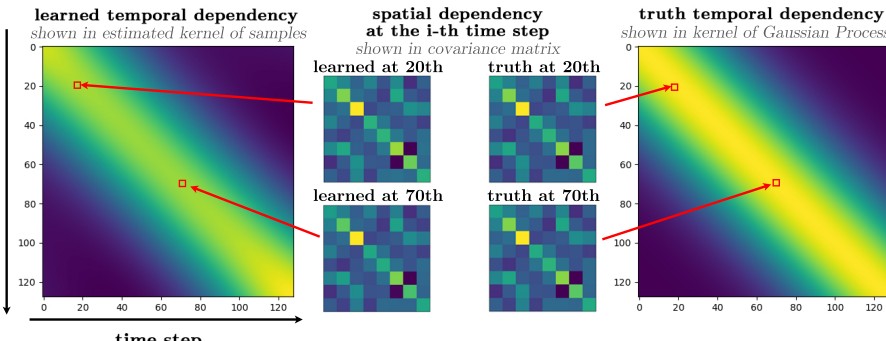

Figure 1: *Diffusion transformer learns spatial-temporal dependencies. The diffusion transformer is trained with data sampled from a stationary Gaussian process consisting of $128$ time steps. At each time step, the data dimension is $8$. We obtain 1000 generated samples at each time step. The left large heat map demonstrates the estimated temporal correlation (see Appendix F for the estimation method) in the process between different time steps, which aligns well with the ground truth on the right. The smaller heat maps are the estimated covariance matrix of data at a single time step, which demonstrate the spatial dependencies in data. They also align well with the ground truth.*

the time. These dependencies resonate well with the self-attention mechanism in transformers for capturing token-wise correlation, suggesting benefits for learning with sequential data. We illustrate diffusion transformer accurately learning spatial-temporal dependencies in Figure 1.

Despite the empirical success, there lacks a rigorous understanding of diffusion transformers for sequential data modeling. Different from static data, sequential data consists of a series of dependent data frames. The data sequence can be extensively long. For instance, a one-minute video would contain over 1440 image frames, and intraday data in financial applications may be even longer.

Therefore, naïvely treating the entire sequential data solely as high-dimensional data without considering its inner spatial-temporal correlations will lead to a large dimensionality dependence and inefficient learning. As a result, existing results of diffusion models for static data can provide few insights on the following fundamental questions:

*Can diffusion transformers efficiently capture spatial-temporal dependencies in sequential data? If yes, how do spatial-temporal dependencies affect the learning efficiency?*

We answer these questions for the first time by studying using diffusion transformers for learning Gaussian process data. Besides its simplicity, Gaussian process exhibits intriguing and salient properties. Firstly, Gaussian process data can be *high-dimensional*, highlighting the influence of data dimensionality in diffusion transformer. Secondly, the spatial-temporal dependencies are the *defining quantities* of a Gaussian process. This necessitates an effective learning of these dependencies. In fact, Gaussian process can encode a wide variety of complicated correlations in real-world applications (Seeger, 2004; Williams and Rasmussen, 2006). For instance, Brownian motion falls into the category of Gaussian process for describing particle movements in a fluid. Gaussian process is also a powerful statistical tool for regression, classification and forecasting problems (Banerjee et al., 2013; Deringer et al., 2021; Chen et al., 2021; Borovitskiy et al., 2021).

**Contributions** Our results show that by construction, transformers can adapt to the spatial-temporal dependencies so as to promote the learning efficiency. Furthermore, we show sample complexity bounds of diffusion transformers, demonstrating the influence of the decay of correlation in the sequential data. We summarize our contributions as follows.

• We propose a novel score function approximation scheme for Gaussian process data, which represents the score function by a gradient descent algorithm (Lemma 1). Then we construct a transformer architecture to unroll the gradient descent algorithm in Theorem 1. We particularly highlight that the attention layer effectively captures the spatial-temporal dependencies. Meanwhile, the decay pattern of those dependencies influences the approximation efficiency.

• Built upon our score function approximation theory, we establish the first sample complexity bound for diffusion transformers in learning sequential data (Theorem 2). We show that the generalization error scales with $1/\sqrt{n}$, where $n$ is the sample size. Our generalization error also demonstrates the influence of dependency decay speed and the length of sequences.

• We provide numerical results to support our theory by showing the learning performance under various settings. More interestingly, we demonstrate that a well-trained diffusion transformer reproduces the ground-truth spatial-temporal dependencies accurately within an attention layer, emphasizing the applicability of our theoretical insights.

Our theories are the first to explain how diffusion transformers model sequential data, while most existing theories of diffusion models focus on static data. Due to space limit, we provide a discussion about related works and our technical novelty to Appendix B.

**Notation** We use bold letters to denote vectors and matrices. For a vector $\mathbf{v}$, we denote its Euclidean norm as $\|\mathbf{v}\|_2$. For a matrix $\mathbf{A}$, we denote its operator, Frobenius norm as $\|\mathbf{A}\|_2$ and $\|\mathbf{A}\|_F$, respectively. Moreover, we denote $\|\mathbf{A}\|_\infty = \max_{i,j} |\mathbf{A}_{i,j}|$. We denote the condition number of a positive definite matrix $\mathbf{A}$ by $\kappa(\mathbf{A}) = \lambda_{\max}(\mathbf{A})/\lambda_{\min}(\mathbf{A})$, where $\lambda_{\max}$ and $\lambda_{\min}$ denote the maximum and minimum eigenvalues. We denote $f \lesssim g$ if $f \leq Cg$ holds for a constant $C > 0$.

## 2 GAUSSIAN PROCESS AND DIFFUSION TRANSFORMER

In this section, we formalize our data modeling and sampling problem with Gaussian process data. Meanwhile, we briefly introduce diffusion processes and transformer architectures.

**Gaussian Process** We denote $\{\mathbf{X}_h\}_{h \in [0,H]}$ as a continuous-time Gaussian process in the time interval $[0, H]$. The process lives in the $d$-dimensional Euclidean space, i.e., $\mathbf{X}_h \in \mathbb{R}^d$ for any $h \in [0, H]$. A defining property of Gaussian process is that for any finite collection of time indices $h_1, \ldots, h_N$ with $N \in \mathbb{N}^+$, the joint distribution of $\{\mathbf{X}_{h_1}, \ldots, \mathbf{X}_{h_N}\}$ is still Gaussian. As a particular example, for a fixed time $h$, the marginal distribution of $\mathbf{X}_h$ is Gaussian.

To fully describe a Gaussian process, we need the concept of mean and covariance functions. Roughly speaking, a mean function $\boldsymbol{\mu}(h) = \mathbb{E}[\mathbf{X}_h]$ characterizes the expected evolution trend of the process. A covariance function $\boldsymbol{\Gamma}(h_1, h_2) = \mathbb{E}[(\mathbf{X}_{h_1} - \boldsymbol{\mu}(h_1))(\mathbf{X}_{h_2} - \boldsymbol{\mu}(h_2))^\top]$ captures the correlation between two time indices in the process. Note that, when $h_1 = h_2 = h$, the covariance function computes the covariance matrix of $\mathbf{X}_h$. Remarkably, the covariance function determines many basic properties of the continuous-time process, such as its stationarity, periodicity, and smoothness (Williams and Rasmussen, 2006). In our later study, we reveal an intimate connection between the behavior of the covariance function to the learning efficiency of diffusion transformer.

Throughout the paper, we focus on Gaussian processes whose covariance functions only depend on the gap between time indices. Accordingly, we reparameterize the covariance function as $\boldsymbol{\Gamma}(h_1, h_2) = \gamma(h_1, h_2)\boldsymbol{\Sigma}$, where $\gamma(\cdot, \cdot)$ is a scalar-output function and $\boldsymbol{\Sigma} = \mathrm{Cov}[\mathbf{X}_h]$ (identical for any $h$).

**Sequential Data Sampled from Gaussian Process** In real-world scenarios, an underlying continuous-time process is often perceived by a sequence of data sampled at discrete times. For example, a video typically consists of 24 to 30 image frames per second. When played back, these frames appear seamless to the human eye, which cannot distinguish between individual frames as if the video is continuous. Following the same spirit, we denote $h_1, \ldots, h_N$ for a sufficiently large $N \in \mathbb{N}^+$ as a uniform grid on the interval $[0, H]$ and form a discrete sequence $\{\mathbf{X}_{h_1}, \ldots, \mathbf{X}_{h_N}\}$ observed at those time indices from an underlying Gaussian process. By the definition of Gaussian process, if we stack $\mathbf{X}_{h_1}, \ldots, \mathbf{X}_{h_N}$ consecutively as a vector in $\mathbb{R}^{dN}$, it follows a Gaussian distribution. The mean is $\boldsymbol{\mu} = [\boldsymbol{\mu}_1^\top, \ldots, \boldsymbol{\mu}_N^\top]^\top$ with $\boldsymbol{\mu}_i = \boldsymbol{\mu}(h_i)$ and the covariance is a block-wise matrix represented as $\boldsymbol{\Gamma} \otimes \boldsymbol{\Sigma}$, where

$$\boldsymbol{\Gamma} = \begin{bmatrix} \gamma(h_1, h_1) & \cdots & \gamma(h_1, h_N) \\ \vdots & \ddots & \vdots \\ \gamma(h_N, h_1) & \cdots & \gamma(h_N, h_N) \end{bmatrix} \quad \text{and} \quad \boldsymbol{\Gamma} \otimes \boldsymbol{\Sigma} = \begin{bmatrix} \boldsymbol{\Gamma}_{11}\boldsymbol{\Sigma} & \cdots & \boldsymbol{\Gamma}_{1N}\boldsymbol{\Sigma} \\ \vdots & \ddots & \vdots \\ \boldsymbol{\Gamma}_{N1}\boldsymbol{\Sigma} & \cdots & \boldsymbol{\Gamma}_{NN}\boldsymbol{\Sigma} \end{bmatrix}.$$

Here, $\otimes$ is the matrix Kronecker product. Notably, $\boldsymbol{\Gamma}$ captures the *temporal dependency* between time indices and $\boldsymbol{\Sigma}$ captures the *spatial dependency* of entries in $\mathbf{X}_h$. Since $h_1, \ldots, h_N$ form a uniform grid, $\boldsymbol{\Gamma}$ is a symmetric Toeplitz matrix with entries taking at most $N$ different values.

Gaussian process data provides explicit description of the spatial-temporal dependencies, but still remains highly relevant to real-world diffusion models. These models utilize a pre-trained Variational AutoEncoder (VAE) to map data into a low-dimensional latent representation (Wang et al., 2023; Blattmann et al., 2023). The typical prior distribution of the low-dimensional representation in VAEs is assumed to be Gaussian. Empirical results have demonstrated the effectiveness of Gaussian latent prior for sequential data modeling in some variants of VAEs (Casale et al., 2018; Fortuin et al., 2020). Our study aligns well with these empirical observations.

In a learning setting, we collect $n$ i.i.d. realizations of the discrete sequence $\{\mathbf{X}_{h_1}, \ldots, \mathbf{X}_{h_N}\}$, denoted as $\mathcal{D} = \{\mathbf{x}_{1,j}, \ldots, \mathbf{x}_{N,j}\}_{j=1}^n$. We aim to use a diffusion transformer for learning and generating new samples mimicking the distribution of the discrete sequence. The subtlety here is that naïvely learning the joint distribution of $\{\mathbf{X}_{h_1}, \ldots, \mathbf{X}_{h_N}\}$ is subject to a large dimension factor $N$, heavily exaggerating the problem dimension and jeopardizing the learning efficiency. Fortunately, we will show that the behavior of the covariance function may induce benign temporal dependencies and largely promote the sample complexity.

**Diffusion Processes** A diffusion model generates new data by progressively removing noise using the so-called "score function". We adopt a continuous-time perspective for a brief review of diffusion models. Interested readers may refer to recent surveys for a comprehensive exposure (Chen et al., 2024; Tang and Zhao, 2024; Chan, 2024). A diffusion model utilizes a forward and a backward process for training and sample generation, respectively:

$$\mathrm{d}\mathbf{X}_t = -\frac{1}{2}\mathbf{X}_t\mathrm{d}t + \mathrm{d}\mathbf{W}_t, \text{ for } t \in [0, T] \text{ and } \mathbf{X}_0 \sim P_0, \tag{Forward}$$

$$\mathrm{d}\mathbf{X}_t^{\leftarrow} = \left[\frac{1}{2}\mathbf{X}_t^{\leftarrow} + \nabla \log p_{T-t}(\mathbf{X}_t^{\leftarrow})\right]\mathrm{d}t + \mathrm{d}\overline{\mathbf{W}}_t, \text{ for } t \in [0, T] \text{ and } \mathbf{X}_0^{\leftarrow} \sim \mathsf{N}(\mathbf{0}, \mathbf{I}). \tag{Backward}$$

Here, $\mathbf{W}_t$ and $\overline{\mathbf{W}}_t$ are independent Wiener processes and $T$ is a finite terminal diffusion timestep. The initial distribution of $\mathbf{X}_0$ is $P_0$, which is also the clean data distribution, and we denote $P_t$ as the marginal distribution of $\mathbf{X}_t$. Since we are corrupting $P_0$ by Gaussian noise, $P_t$ has a density function $p_t$. Thus, $\nabla \log p_t$ in the backward process is recognized as the score function, which is unknown and requires learning. Typically, the score function will be parameterized by a neural network and trained by optimizing a loss function, which we will introduce in Section 5. When generating new samples, we simulate a discretized version of the backward process using the learned score function.

For Gaussian process data, we interpret $\mathbf{X}_t$ as a vector in $\mathbb{R}^{dN}$ by stacking $N$ observations. We term each observation as a patch. Equivalently, the forward process is to add independent Gaussian noise to each patch simultaneously. However, the backward process cannot be decomposed according to patches, as the score function encodes their correlation; see Section 3 for a detailed discussion on the structure of $\nabla \log p_t$.

**Transformer Architecture** A transformer comprises a series of blocks and each block encompasses a multi-head attention layer and a feedforward layer. Let $\mathbf{Y} = [\mathbf{y}_1, \ldots, \mathbf{y}_N] \in \mathbb{R}^{D \times N}$ be the (column) stacking matrix of $N$ patches. In a transformer block, multi-head attention computes

$$\mathtt{Attn}(\mathbf{Y}) = \mathbf{Y} + \sum_{m=1}^{M} \mathbf{V}^m \mathbf{Y} \cdot \sigma\left((\mathbf{Q}^m\mathbf{Y})^\top \mathbf{K}^m\mathbf{Y}\right), \tag{1}$$

where $\mathbf{V}^m, \mathbf{Q}^m$, and $\mathbf{K}^m$ are weight matrices of corresponding sizes in the $m$-th attention head, and $\sigma$ is an activation function. The attention layer is followed by a feedforward layer, which computes

$$\mathtt{FFN}(\mathbf{Y}) = \mathbf{Y} + \mathbf{W}_1 \cdot \mathrm{ReLU}(\mathbf{W}_2\mathbf{Y} + \mathbf{b}_2\mathbf{1}^\top) + \mathbf{b}_1\mathbf{1}^\top.$$

Here, $\mathbf{W}_1, \mathbf{W}_2$ are weight matrices, $\mathbf{b}_1$ and $\mathbf{b}_2$ are offset vectors, $\mathbf{1}$ denotes a vector of ones, and the ReLU activation function is applied entrywise. For our study, the raw input to a transformer is $N$ patches of $d$-dimensional vectors and diffusion timestep $t$ in the backward process. We refer to $\mathcal{T}(D, L, M, B, R)$ as a transformer architecture defined by

$$\begin{aligned}\mathcal{T}(D, L, M, B, R) = \{f : f &= f_{\mathrm{out}} \circ (\mathtt{FFN}_L \circ \mathtt{Attn}_L) \circ \cdots \circ (\mathtt{FFN}_1 \circ \mathtt{Attn}_1) \circ f_{\mathrm{in}}, \\ &\mathtt{Attn}_i \text{ uses entrywise ReLU activation for } i = 1, \ldots, L, \\ &\text{number of heads in each } \mathtt{Attn} \text{ is bounded by } M, \\ &\text{the Frobenius norm of each weight matrix is bounded by } B, \\ &\text{the output range } \|f\|_2 \text{ is bounded by } R\}.\end{aligned}$$

See Figure 2 for an illustration of $f_{\mathrm{in}}$ and $f_{\mathrm{out}}$. For attention layers, we consider ReLU activation for technical convenience and postpone a discussion with softmax activation to Appendix D.5.

## 3 REPRESENT SCORE FUNCTION AS THE LAST ITERATE OF GRADIENT DESCENT

Diffusion transformer learns the sequential data distribution through estimating the score function. In this section, we study how can transformers effectively represent the score function of our Gaussian process data. We are particularly interested in understanding how can transformers capture the spatial-temporal dependencies via multi-head attention.

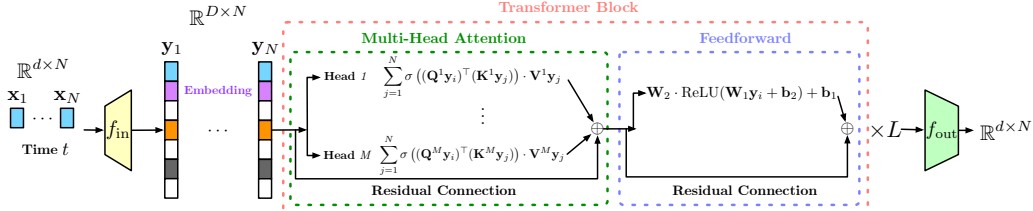

Figure 2: *Transformer architecture. Here $f_{\mathrm{in}}$ is a linear layer to lift input patch to $\mathbb{R}^D$, which appends the input raw data with time index embedding and other useful information. After passing through $L$ transformer blocks, $f_{\mathrm{out}}$ projects each patch into the data original dimension $\mathbb{R}^d$ and clip the output range by $R$. We allow the output range to be diffusion timestep $t$ dependent (denoted as $R_t$).*

### 3.1 SCORE FUNCTION FOR GAUSSIAN PROCESS DATA

Recall from Section 2 that the joint distribution of our $N$-patch Gaussian process data is still Gaussian. We can show that at diffusion timestep $t \in (0, T]$, given $\mathbf{v}_t = [\mathbf{x}_{1,t}^\top, \ldots, \mathbf{x}_{N,t}^\top]^\top$ in the backward process, the score function is

$$\nabla \log p_t(\mathbf{v}_t) = -(\alpha_t^2 \mathbf{\Gamma} \otimes \mathbf{\Sigma} + \sigma_t^2 \mathbf{I})^{-1}(\mathbf{v}_t - \alpha_t \boldsymbol{\mu}) \quad \text{for} \quad \alpha_t = e^{-t/2} \text{ and } \sigma_t = \sqrt{1 - e^{-t}}, \quad (2)$$

where $\boldsymbol{\mu}$ is the mean vector, and $\alpha_t$ and $\sigma_t$ are determined by the diffusion forward process. We defer the derivation to Appendix C.1. Observe that the score function for the $i$-th patch depends on the evolution of all patches, reflecting temporal dependencies in sequential data. Moreover, the influence of each patch is determined by the covariance function $\mathbf{\Gamma}_{ij} = \gamma(h_i, h_j)$. In the extreme case of $\mathbf{\Gamma}_{ij} = \mathbb{1}\{i = j\}$, i.e., there is no correlation between patches, the score function reduces to $(\alpha_t^2 \mathbf{I} \otimes \mathbf{\Sigma} + \sigma_t^2 \mathbf{I})^{-1}(\mathbf{v}_t - \alpha_t \boldsymbol{\mu})$, which isolates patches and reproduces the score function of the Gaussian distribution $\mathsf{N}(\boldsymbol{\mu}_i, \mathbf{\Sigma})$ for each patch.

Although the score function assumes the closed-form expression in (2), effectively representing it may suffer from difficulties. In fact, there are dependencies among $N$ patches and the correlation is encoded by the inverse covariance matrix $(\alpha_t^2 \mathbf{\Gamma} \otimes \mathbf{\Sigma} + \sigma_t^2 \mathbf{I})^{-1}$. Directly representing it using a transformer is subject to high complexity and a large dimension depending on $N$. To overcome the challenge, we resort to an algorithm unrolling perspective and leveraging the attention mechanism in transformers. The key insight is to approximate the score function as the last iterate of a gradient descent algorithm, where the algorithm can be efficiently implemented by a transformer.

### 3.2 SCORE FUNCTION IS THE OPTIMIZER OF A CONVEX FUNCTION AND GRADIENT DESCENT FINDS IT

We consider a fixed diffusion timestep $t \in (0, T]$. It is straightforward to check by the first-order optimality that $\nabla \log p_t(\mathbf{v}_t)$ is the minimizer of the following quadratic objective function,

$$\nabla \log p_t(\mathbf{v}_t) = \underset{\mathbf{s} \in \mathbb{R}^{dN}}{\operatorname{argmin}} \ \mathcal{L}_t(\mathbf{s}) := \frac{1}{2}\mathbf{s}^\top \left(\alpha_t^2(\mathbf{\Gamma} \otimes \mathbf{\Sigma}) + \sigma_t^2 \mathbf{I}\right)\mathbf{s} + (\mathbf{v}_t - \alpha_t \boldsymbol{\mu})^\top \mathbf{s}. \quad (3)$$

Examining (3) reveals that the objective function is strongly convex as long as $t > 0$. Moreover importantly, the formulation (3) is free of matrix inverse and the optimal solution can be found by a gradient descent algorithm. Specifically, in the $k$-th iteration, gradient descent for $\mathcal{L}_t$ computes

$$\mathbf{s}^{(k+1)} = \mathbf{s}^{(k)} - \eta_t \cdot \left(\left(\alpha_t^2(\mathbf{\Gamma} \otimes \mathbf{\Sigma}) + \sigma_t^2 \mathbf{I}\right)\mathbf{s}^{(k)} + (\mathbf{v}_t - \alpha_t \boldsymbol{\mu})\right), \quad (4)$$

where $\eta_t$ is a proper step size. Comparing to (2), the gradient descent iteration avoids the matrix inverse, and we can further decompose the update (4) according to each patch. With well-conditioned covariance matrix $\mathbf{\Gamma} \otimes \mathbf{\Sigma}$, the gradient descent algorithm converges exponentially fast for approximating the ground-truth score function. Moreover, we could substitute $\mathbf{\Gamma}$ in (4) by any of its approximation $\bar{\mathbf{\Gamma}}$. We quantify the representation error after sufficient gradient descent iterations.

**Lemma 1** (**Gradient Descent Iterate Approximates the Score Function**). For an arbitrarily fixed $t \in [0, T]$ and $\mathbf{v}_t$, given an error tolerance $\epsilon > 0$ and any integer $J < N$, if $\bar{\mathbf{\Gamma}}$ with $\bar{\mathbf{\Gamma}}_{ij} = \mathbf{\Gamma}_{ij}\mathbb{1}\{|i - j| < J\}$ is positive semidefinite, then running gradient descent in (4) with a suitable step size $\eta_t$ for $K = \mathcal{O}(\kappa_t \log(1/\epsilon))$ iterations gives rise to

$$\left\|\mathbf{s}^{(K)}(\mathbf{v}_t) - \nabla \log p_t(\mathbf{v}_t)\right\|_2 \leq \underbrace{\frac{1}{\sigma_t^2}\|\mathbf{v}_t - \alpha_t \boldsymbol{\mu}\|_2 \epsilon}_{\mathcal{E}_1:\text{GD representation error}} + \underbrace{\frac{\|\mathbf{\Sigma}\|_{\mathrm{F}}\|\mathbf{v}_t - \alpha_t \boldsymbol{\mu}\|_2}{\sigma_t^4}\sqrt{\sum_{|i-j|\geq J}\mathbf{\Gamma}_{ij}^2}}_{\mathcal{E}_2:\text{correlation truncation error}},$$

where $\kappa_t = \kappa\big(\alpha_t^2(\bar{\boldsymbol{\Gamma}} \otimes \boldsymbol{\Sigma}) + \sigma_t^2\mathbf{I}\big)$ is the condition number of $\alpha_t^2(\bar{\boldsymbol{\Gamma}} \otimes \boldsymbol{\Sigma}) + \sigma_t^2\mathbf{I}$.

The proof is deferred to Appendix C.2. To interpret the lemma, we first consider $J = N$, which implies $\mathcal{E}_2 = 0$ and $\mathcal{E}_1$ recovers the typical convergence guarantee of gradient descent algorithm: for a strongly convex and smooth function, gradient descent algorithm converges exponentially fast.

**Controlling the Length of Dependencies**  The special treatment in Lemma 1 concentrates on the truncation length $J$. On the high level, $J$ defines the maximum length of temporal dependencies we aim to model in the score function. In particular, instead of working with (4), we consider a surrogate gradient descent iteration driven by replacing $\boldsymbol{\Gamma}$ by $\bar{\boldsymbol{\Gamma}}$, which neglects correlations beyond length $J$. We discuss sufficient conditions for ensuring $\bar{\boldsymbol{\Gamma}}$ being positive semidefinite after Assumption 1. Introducing $\bar{\boldsymbol{\Gamma}}$ incurs the truncation error $\mathcal{E}_2$, whose magnitude depends on the pattern of temporal dependencies. Apparently, with long-horizon dependencies, the truncation error $\mathcal{E}_2$ tends to be large. However, advantage appears in the presence of decaying dependencies, since we can neglect faint correlation to promote the representation and learning efficiency.

### 3.3 Bounding Correlation Truncation Error $\mathcal{E}_2$

In many dynamical systems, temporal dependencies decay rather fast as a function of the time gap. The decay pattern may lead to a well controlled truncation error $\mathcal{E}_2$. As the decay is determined by the covariance function in a Gaussian process, we impose the following assumption.

**Assumption 1.** Let $\mathbf{e}_1, \ldots, \mathbf{e}_N \in \mathbb{R}^{d_e}$ be $d_e$-dimensional time embedding of $h_1, \ldots, h_N$ with $\|\mathbf{e}_i\|_2 = r$ for each $i$ such that there exists a positive and increasing function $f$ with $\|\mathbf{e}_i - \mathbf{e}_j\|_2 = f(|i - j|) \geq c|i - j|$ for an absolute constant $c > 0$. Further, the covariance function $\gamma$ satisfies

$$\gamma(h_i, h_j) = \exp\big(- \|\mathbf{e}_i - \mathbf{e}_j\|_2^\nu /\ell\big) \quad \text{for} \quad \nu \in [1, 2] \quad \text{and} \quad \ell > 0.$$

Firstly, Assumption 1 says that the time embedding (a.k.a. position embedding) preserves the gap between real times. For example, a common time embedding used in sequential data modeling (Vaswani et al., 2017) is sinusoidal transformations, where $\mathbf{e}_i = [r\sin(2i\pi/C), r\cos(2i\pi/C)]^\top \in \mathbb{R}^2$ for a positive radius $r$ and a large constant $C > 0$. We can check that $\|\mathbf{e}_i - \mathbf{e}_j\|_2 = 2r\sin(|i - j|\pi/C) \geq 4r|i - j|/C$ is positive and approximately linearly increasing for a sufficiently large $C$.

Secondly, the covariance function decays exponentially fast and the speed is controlled by the exponent $\nu$ and the bandwidth $\ell$. Large $\ell$ or small $\nu$ indicates that the correlation between different time indices decays relatively slowly. Thus, the sequential data has some long-horizon dependencies. The range of $\nu$ includes the well-known quadratic-exponential (Gaussian) covariance function ($\nu = 2$). Moreover, when $\nu = 1$, the covariance function coincides with the correlation in a Brownian motion. Varying $\nu \in [1, 2]$ can capture abundant temporal dependency patterns.

Besides, under Assumption 1, we can provide a sufficient condition for ensuring $\bar{\boldsymbol{\Gamma}}$ being positive semidefinite for any $J$. Specifically, when $\ell \leq c^\nu$, $\boldsymbol{\Gamma}$ is symmetric diagonally dominant, i.e., the diagonal entry has a larger magnitude than the sum of the magnitudes of off diagonal entries. This implies that for any truncation length $J$, $\bar{\boldsymbol{\Gamma}}$ is always positive semidefinite. See Remark 1 in Appendix C.3 for a formal justification. Apparently, requiring $\ell \leq c^\nu$ is not necessary and as long as the covariance function decays sufficiently fast, $\bar{\boldsymbol{\Gamma}}$ can be positive semidefinite. We now establish the following corollary to demonstrate a reasonable choice of $J$.

**Corollary 1 (Correlation Truncation with Decay).** Suppose Assumption 1 holds with $\ell \leq c^\nu$. For any $\epsilon < \lambda_{\min}(\boldsymbol{\Gamma})$, under the setup of Lemma 1, by setting $J = \mathcal{O}\big((\ell \log(N/(\epsilon\sigma_t)))^{1/\nu}\big)$, it holds that

$$\left\|\mathbf{s}^{(K)}(\mathbf{v}_t) - \nabla \log p_t(\mathbf{v}_t)\right\|_2 \leq 2\sigma_t^{-2} \|\mathbf{v}_t - \alpha_t\boldsymbol{\mu}\|_2\, \epsilon.$$

The proof is deferred to Appendix C.3, where we keep the $\ell \leq c^\nu$ condition for technical convenience. This should not be considered restrictive, as our theory holds as long as the truncation of $\boldsymbol{\Gamma}$ at length $J$ is positive semidefinite. Corollary 1 shows that the truncation error $\mathcal{E}_2$ can be controlled at the same order of $\mathcal{E}_1$. The truncation length $J$ is logarithmically dependent on the full length of the sequence, indicating that we can focus on relatively short-horizon dependencies proportional to the bandwidth $\ell$. This observation is the key to promote the representation and learning efficiency of diffusion transformer.

## 4 APPROXIMATION THEORY OF SCORE FUNCTION USING TRANSFORMERS

This section devotes to establishing a transformer approximation theory of the score function. Different from the existing universal approximation theories (Cybenko, 1989; Yarotsky, 2018), we construct a transformer to unroll the gradient descent algorithm for representing the score function. We show this perspective leads to an efficient approximation in the following theorem.

**Theorem 1 (Score Approximation by Transformers).** Suppose Assumption 1 holds with $\ell \leq c^\nu$. Given any $t_0 \in (0, T]$ and a small $\epsilon < \lambda_{\min}(\mathbf{\Gamma})$, there exists a transformer architecture $\mathcal{T}(D, L, M, B, R_t)$ such that with proper weight parameters, it yields an approximation $\widetilde{\mathbf{s}}$ to the score function $\nabla \log p_t$ with

$$\int \|\widetilde{\mathbf{s}}(\mathbf{v}_t) - \nabla \log p_t(\mathbf{v}_t)\|_2^2 \, p_t(\mathbf{v}_t) \mathrm{d}\mathbf{v}_t \leq \sigma_t^{-2}\epsilon \quad \text{for all} \quad t \in [t_0, T].$$

The transformer architecture satisfies

$$D = \mathcal{O}(d + d_e), \quad L = \mathcal{O}\big(\kappa_{t_0} \log^2(Nd/(\epsilon\sigma_{t_0}))\big), \quad M = \mathcal{O}\Big((\ell \log(Nd \|\mathbf{\Sigma}\|_{\mathrm{F}} /(\epsilon\sigma_{t_0})))^{1/\nu}\Big),$$

$$B = \mathcal{O}\big(\log(Nd/(\epsilon\sigma_{t_0}))\sigma_{t_0}^{-2}Nd(r^2 + \|\mathbf{\Sigma}\|_\infty)\big), \quad R_t = \mathcal{O}\Big(\log(Nd/(\epsilon\sigma_{t_0}))\sqrt{Nd}/\sigma_t\Big).$$

The proof is deferred to Appendix D.1. Here we observe that the approximation error depends on $\sigma_t$, which matches existing results for studying score approximation using neural networks (Oko et al., 2023; Chen et al., 2023a; Tang and Yang, 2024). Yet we remark that our algorithm unrolling approach is very different from these existing works. Our results also hold for softmax activated transformer architectures, which is discussed in Appendix D.5. We remark on other interpretations.

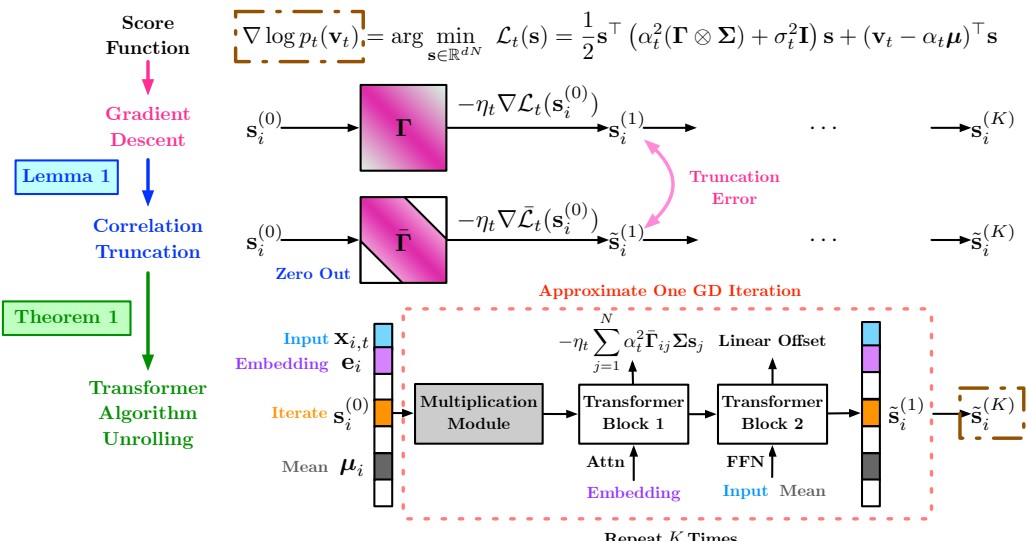

Figure 3: *Construction of score function approximation using a transformer. By rewriting the score function as the optimizer of a quadratic objective function, we use gradient descent algorithm to approximate the optimizer. We allow correlation truncation to manipulate the maximum length of temporal dependencies to model in Lemma 1. Then we construct a transformer architecture to unroll the gradient descent algorithm for score approximation in Theorem 1. Each gradient descent iteration is realized by a multiplication module followed by two transformer blocks. In the first transformer block, its attention layer calculates correlation $\mathbf{\Gamma}_{ij}$ utilizing time embedding. The second transformer block calculates the linear offset $-\eta_t \sigma_t^2 \mathbf{s} - \eta_t(\mathbf{v}_t - \alpha_t \boldsymbol{\mu})$ in (4).*

**A Glimpse of Transformer Architecture** The constructed transformer architecture is demonstrated in Figure 3. To achieve an approximation, the transformer unrolls the gradient descent algorithm. For realizing a single step, a multiplication module calculates time dependent rescaling parameters, e.g., $\alpha_t$ and $\sigma_t$. Then two transformer blocks implement the iteration in (4). As can be seen, the raw input is lifted into a higher dimensional vector, containing time embedding and other useful information. It is worth mentioning that the total number of transformer blocks is proportional to

the condition number $\kappa_{t_0}$. For sharply decaying covariance functions, $\kappa_{t_0}$ is a constant. However, for slowly decaying covariance functions, $\kappa_{t_0}$ can be large, indicating the fundamental difficulty of capturing long-horizon dependencies. While explicitly bounding $\kappa_{t_0}$ for a finite $N$ goes beyond the current technical limit, we discuss its asymptotic behavior in Appendix G.4.

**What Is Represented in Self-Attention** We zoom into our constructed transformer architecture in Figure 3 to understand the role of multi-head attention layer in Transformer Block 1. Here the attention layer is constructed with proper $\mathbf{Q}$ and $\mathbf{K}$ matrices, so that it finds the correlation between a pair of time embedding $\mathbf{e}_i$ and $\mathbf{e}_j$. Specifically, it calculates the inner product $\mathbf{e}_i^\top \mathbf{e}_j$ for approximating the correlation coefficient $\gamma(h_i, h_j)$. Interestingly, our numerical results in Section 6 support this construction, providing evidence that a well-trained diffusion transformer puts large weight corresponding to time embedding. We suspect that our construction provides practical insights on how correlation is learned in attention layers.

## 5 SAMPLE COMPLEXITY OF DIFFUSION TRANSFORMER

Given a properly transformer architecture, this section studies the sample complexity of diffusion transformer for learning Gaussian process data. As mentioned in Section 2, the training of diffusion transformer is to estimate the score function. Conceptually, we can use a quadratic loss,

$$\operatorname{argmin}_{\mathbf{s}\in\mathcal{T}} \int_0^T \mathbb{E}_{\mathbf{v}_t\sim P_t} \|\mathbf{s}(\mathbf{v}_t, t) - \nabla \log p_t(\mathbf{v}_t)\|_2^2 \, dt, \tag{5}$$

However, this loss function is not directly implementable due to $\nabla \log p_t$ being unknown and numerical instability when $t$ approaches zero. Therefore, we consider the following loss,

$$\operatorname{argmin}_{\mathbf{s}\in\mathcal{T}} \int_{t_0}^T \mathbb{E}_{\mathbf{v}_0} \mathbb{E}_{\mathbf{v}_t\sim\mathsf{N}(\alpha_t\mathbf{v}_0, \sigma_t^2\mathbf{I})} \left\| \mathbf{s}(\mathbf{v}_t, t) - \frac{\alpha_t\mathbf{v}_0 - \mathbf{v}_t}{\sigma_t^2} \right\|_2^2 \, dt. \tag{6}$$

Here, $t_0$ is an early-stopping time and $\frac{\alpha_t\mathbf{v}_0 - \mathbf{v}_t}{\sigma_t^2}$ substitutes the unknown score function $\nabla \log p_t$. The equivalence of (6) to (5) is established in the seminal works (Vincent, 2011; Hyvärinen and Dayan, 2005). When given the collected data set $\mathcal{D} = \{\mathbf{x}_{1,j}, \ldots, \mathbf{x}_{N,j}\}_{j=1}^n$, we replace the population expectation $\mathbb{E}_{\mathbf{v}_0}$ in (6) by a sample empirical distribution. We denote $\mathbf{v}_0^i = [\mathbf{x}_{i,1}^\top, \ldots, \mathbf{x}_{i,N}^\top]^\top$ as the stacking vector of a data sequence. Then the estimated score function $\widehat{\mathbf{s}}$ can be written as an empirical risk minimizer,

$$\widehat{\mathbf{s}} \in \operatorname{argmin}_{\mathbf{s}\in\mathcal{T}} \frac{1}{n} \sum_{i=1}^n \int_{t_0}^T \mathbb{E}_{\mathbf{v}_t\sim\mathsf{N}(\alpha_t\mathbf{v}_0^i, \sigma_t^2\mathbf{I})} \left\| \mathbf{s}(\mathbf{v}_t, t) - \frac{\alpha_t\mathbf{v}_0^i - \mathbf{v}_t}{\sigma_t^2} \right\|_2^2 \, dt.$$

To generate new samples, diffusion transformer uses $\widehat{\mathbf{s}}_n$ in the backward process. Correspondingly, we denote the distribution learned by such a diffusion transformer as $\widehat{P}$. We bound the divergence of $\widehat{P}$ to our ground-truth data distribution $P_0$ in the following theorem.

**Theorem 2 (Sample Complexity of Diffusion Transformer).** Suppose Assumption 1 holds with $\ell \le c^\nu$. Assume there exists a constant $C > 1$ such that $C^{-1} \le \lambda_{\min}(\mathbf{\Sigma}) \le \lambda_{\max}(\mathbf{\Sigma}) \le C$ and $\|\boldsymbol{\mu}\|_\infty \le C$. We choose the transformer architecture $\mathcal{T}(D, L, M, B, R_t)$ as in Theorem 1 with $\epsilon = 1/n$. By setting the terminal diffusion timestep $T = \log(n)$, considering $\widehat{P}$ generated by the empirical risk minimizer $\widehat{\mathbf{s}}$, we have

$$\mathbb{E}_{\mathcal{D}}\left[\mathrm{TV}(\widetilde{P}_0, \widehat{P})\right] \lesssim \sqrt{\frac{\ell^{1/\nu}\kappa_{t_0}^2 N d^3}{n}} \cdot \log^{\frac{5\nu+1}{2\nu}}\left(\kappa_{t_0} n d N t_0^{-1}\right), \tag{7}$$

where $\widetilde{P}_0$ is a perturbed data distribution satisfying $W_2(P_0, \widetilde{P}_0) \lesssim \ell\sqrt{t_0 N d}$.

The proof is deferred to Appendix E. We remark that the emergence of $\widetilde{P}_0$ owes to the early-stopping in score estimation, which is obtained by evolving $P_0$ along the forward process for timestep $t_0$. The optimal choice on $t_0$ depends on the condition number $\kappa_{t_0}$. For fast decay covariance functions, $\kappa_{t_0} = \mathcal{O}(1)$ and we can choose $t_0 = 1/n$ and the generalization error is in the order of $\widetilde{\mathcal{O}}(\sqrt{\ell N d^3/n})$. In this case, we have a relatively weak dependence on $N$, demonstrating the efficiency of diffusion transformers in sequential data modeling. On the other hand, when $\kappa_{t_0}$ is large, i.e., with the presence of long-horizon dependencies, the learning efficiency suffers from heavier dependence on $N$. We experiment with various decaying patterns in Section 6 and show the corresponding performance.

## 6 NUMERICAL RESULTS

### 6.1 EXPERIMENTS ON GAUSSIAN PROCESS DATA

In this section, we conduct experiments on diffusion transformer for learning synthetic Gaussian process data. We test various aspects of our theory, including learning efficiency and the capture of spatial-temporal dependencies.

**Experiment Setup** We consider synthetic Gaussian process data with $d = 8$ and $N = 128$. We stick to the setting where the covariance matrix is generated by $\mathbf{\Sigma} = \mathbf{A}^\top \mathbf{A} \in \mathbb{R}^{8\times8}$ for a Gaussian random matrix $\mathbf{A} \in \mathbb{R}^{8\times8}$. We set the mean vector $\boldsymbol{\mu} = \mathbf{0}$ for simplicity. The covariance function is chosen as in Assumption 1 with $\gamma(h_i, h_j) = \exp\left(-|h_i - h_j|^\nu/\ell\right)$. We set $\ell$ and $\nu$ as hyperparameters and report their influences on the learning. We generate $n \in \{1000, 3200, 10000, 32000, 100000\}$ sequences from the Gaussian process as our training data in different settings. The diffusion transformer is implemented based on the DiT (Peebles and Xie, 2023) code base. We set the number of transformer blocks to be 12 throughout all the experiments. We modify the patchify module in DiT to cope with our Gaussian process data. Additional implementation details can be found in Appendix F.

**Influence of Covariance Function Decay on Capturing Spatial-Temporal Dependencies** We study the influence of covariance functions on the performance of diffusion transformers. We vary the exponent $\nu$ and bandwidth $\ell$ as follows: 1) we keep $\ell = 64$, while $\nu$ varies in $\{1, 2^{1/4}, 2^{1/2}, 2^{3/4}, 2\}$; 2) we keep $\nu = 2$, while $\ell$ varies in $\{4, 16, 64, 256, 1024\}$. After training, we generate 1000 samples at each time step for performance evaluation. The metric is a relative error $\epsilon$ of the estimated sample covariance matrix to its ground-truth (see a definition in Appendix F), which is reported in Figure 4(a).

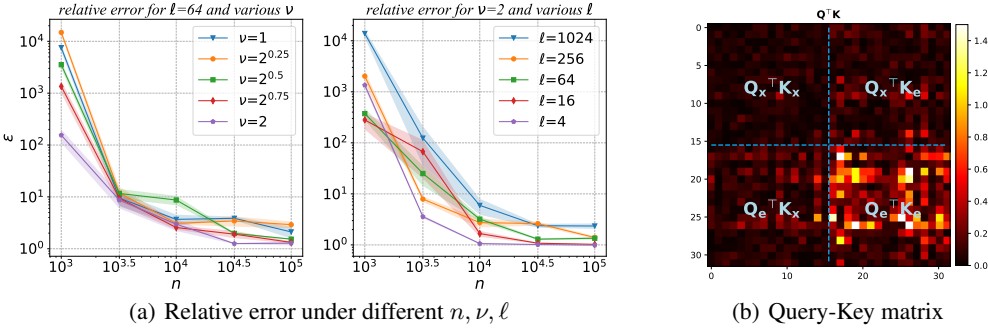

(a) Relative error under different $n, \nu, \ell$   (b) Query-Key matrix

Figure 4: *In panel (a), we observe that the relative error decreases as the sample size increases. Meanwhile, larger $\nu$ or smaller $\ell$ leads to better performance, supporting our generalization bound in Theorem 2. In panel (b), we split $\mathbf{Q}$ to $\mathbf{Q} = [\mathbf{Q}_x, \mathbf{Q}_e]$ where $\mathbf{Q}_x \in \mathbb{R}^{16\times32}$ and $\mathbf{Q}_e \in \mathbb{R}^{16\times32}$ corresponds to time embedding $\mathbf{e}_i$. We also split the key matrix $\mathbf{K}$ as $[\mathbf{K}_x, \mathbf{K}_e]$. The sub-block $\mathbf{Q}_e^\top \mathbf{K}_e$ has dominant weights compared to other sub-blocks.*

**Transformer's Query-Key Matrices Coincides with Our Approximation Theory** We dive into transformer blocks to understand how attention layer captures dependencies. Inside a transformer block, the input is a concatenation of a data vector and a corresponding time embedding written as $[\mathbf{z}_{i,t}^\top, \mathbf{e}_i^\top]^\top \in \mathbb{R}^{32}$, where $\mathbf{e}_i \in \mathbb{R}^{16}$ is the time embedding and $\mathbf{z}_{i,t} \in \mathbb{R}^{16}$ is the output of the patchify module in DiT, obtained by a linear transformation on raw data at diffusion timestep $t$. We plot the heat map of query and key matrices in the 5th transformer block in Figure 4(b). Plots for other transformer blocks are provided in Appendix F. As can be seen, the interaction between time embedding (bottom-right block) is dominant, which aligns with our approximation theory for constructing the transformer architecture.

**Backward Diffusion Process Unveils the Kernel Matrix in Attention Scores** To further understand how DiT captures the temporal dependencies, i.e., matrix $\mathbf{\Gamma}$ for Gaussian process data, we plot the evolution of score matrices $(\mathbf{QY})^\top \mathbf{KY}$ in the attention layers at different steps in the backward diffusion process. Besides, we demonstrate the gradual change of score matrices in different attention layers. Specifically, Figure 5 presents that with progressive denoising in the backward process, the attention score matrix becomes more and more similar to the ground truth $\mathbf{\Gamma}$. In addition, in the first few attention layers, e.g., the first and the second layers, the dependencies are not well-structured and as predicted by our theory, these layers are still realizing some transformations on their inputs. Starting from the 3rd layer, the attention score matrices gradually exhibit the pattern of the ground

truth temporal dependencies. In further subsequent layers, we observe that the learned pattern of temporal dependencies is kept. We refer readers to Figure 12 in Appendix F for a complete plot of score matrices in each attention layer.

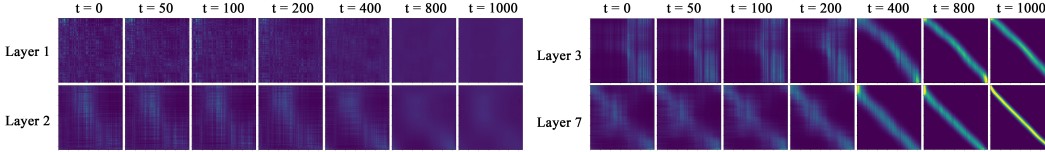

Figure 5: *We demonstrate score matrices in different attention layers and at different backward denoising steps. The learned temporal dependencies gain more and more clarity as the denoising in the backward process proceeds. Meanwhile, we observe that the temporal dependencies are well captured starting from the 3rd layer.*

## 6.2 EXPERIMENTS ON SEMI-SYNTHETIC VIDEO DATA

To further demonstrate the capability of diffusion transformers in capturing spatial-temporal dependencies, we consider learning 2D motions of a ball. The motion is described by a sequence of gray-scale image frames of resolution $64 \times 64$, which characterizes the ball that starts moving toward a random direction in a cube and bounces back when hitting a wall. Because of the bouncing-back mechanism, the dynamic of the ball goes beyond the class of Gaussian process and exemplifies more complex dependencies with abrupt changes. We train a latent diffusion model (Rombach et al., 2022), where we first generate 20000 image frames for training a 2D Variational Autoencoder (VAE). The 2D VAE sets the latent representation in $\mathbb{R}^2$. Once the VAE is trained, we fix it and generate independently 10000 motion videos, each consisting of 240 image frames. The diffusion transformer is trained on the latent representations of the generated videos. As shown in Figure 6, we observe that spatial-temporal dependencies are well captured. We provide generated video frames in Appendix F, which features appealing time-consistency and satisfies the bouncing-back mechanism.

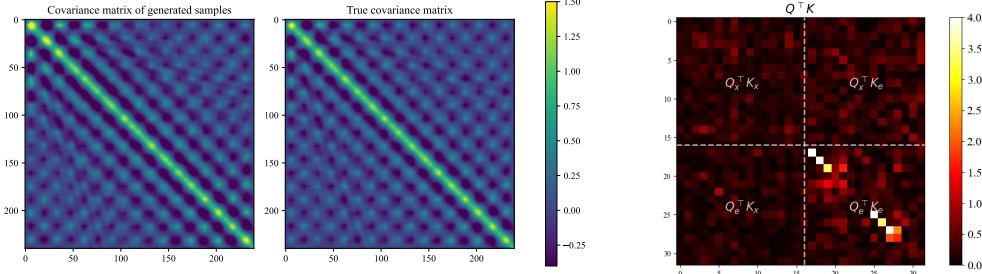

(a) Learned spatial-temporal dependencies (left) compared with the ground truth (right).

(b) The query-key matrix of the first diffusion transformer block.

Figure 6: In panel (a), we observe that the spatial-temporal dependencies (represented by the covariance matrix) have been well captured by diffusion transformers. We collect 1000 generated samples to calculate the dependencies. In panel (b), similar to Panel (b) in Figure 4, the learned query-key matrix demonstrates dominant scores between time embedding (bottom-right block).

## 7 CONCLUSION AND DISCUSSION

We have studied diffusion transformers for learning Gaussian process data. We have developed a score function approximation theory, by leveraging transformers to unroll a gradient descent algorithm. Further, we have established sample complexities of diffusion transformer and discussed the influence of spatial-temporal dependencies on learning efficiency. While Gaussian process data enjoys mathematical simplicity and is relatively preliminary, our theoretical insights and experimental findings can provide invaluable intuition to analyze and design sequential data modeling using diffusion processes. An interesting future direction is to consider generic dynamic models. We expect broad and positive societal impact on advancing diffusion models for sequential data synthesis, forecasting and editing, including video and audio snippets.

# 8 ACKNOWLEDGMENTS

Mengdi Wang acknowledges support by NSF grants DMS-1953686, IIS-2107304, CMMI-1653435, and ONR grant 1006977.

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

## A  ADDITIONAL EXPERIMENTS

We provide additional numerical results to support and validate our theory: 1) We present comparisons between diffusion transformers and vanilla diffusion models with UNet, and 2) We vary the size of the diffusion transformers and demonstrate the training dynamic of gradually capturing spatial-temporal dependencies.

**Comparison of DiT to diffusion with UNet**    To conduct a comparison between DiT and diffusion with UNet, we choose one instance of our Gaussian process data with $d = 8$ and $N = 128$. The covariance function is Gaussian kernel with $\nu = 2$ and $\ell = 64$. We collect $n = 10000$ sequences for training a DiT with 12 transformer blocks and a UNet-based diffusion model with 4 down/up sampling procedures. Each of the down/up sampling in UNet consists of 3 residual and convolution layers so that the DiT and UNet have approximately the same model size.

Each model is sufficiently trained for $400$ epochs, when the training error has converged. In the testing stage, we collect $10000$ samples generated separately from each model and estimate the spatial-temporal dependencies for comparison. As shown in the first column of Figure 7, both DiT and UNet-based diffusion model capture the decay pattern in the temporal correlation. However, DiT exhibits a much better learning result, matching the ground truth. The temporal correlation of the samples generated by UNet-based diffusion model presents a "piecewise" pattern, not as smooth as the ground truth. We conjecture that it is caused by the size of the filter in convolution layers that prevents UNets from learning complete temporal correlation.

Moreover, DiT exhibits significant strength in learning the spatial correlation. As shown in the remaining columns of Figure 7, DiT successfully captures the spatial correlation between two tokens even sufficiently separated, i.e., the correlation is rather weak. In contrast, UNet-based diffusion struggles in learning spatial correlation. We find clear inconsistent patterns of the estimated spatial correlation in the third row, not to mention that the pattern deviates from the ground truth. This result not only demonstrates the surprising learning and generalization capabilities of DiT in sequential data, but also indicates some advantage of DiT over UNet-based diffusion models.

**Performance of DiT with varying network size and sequence length**    We study the influence of the sequence length $N$ and the number of transformer blocks $L$ on the performance. We choose the sequence length $N$ and the number of transformer blocks $L$ as follows:

1) we fix $L = 16$, while change $N$ in $\{2^4, 2^5, 2^6, 2^7, 2^8\}$;

2) we fix $N = 128$, while change $L$ in $\{1, 2, 4, 8, 16\}$.

We adopt a uniform Gaussian process setting of $d = 8$, $\nu = 2$ and $\ell = 64$ as considered in the experiments shown in Figure 7. After training, we independently generate 10000 sequences for performance evaluation. The metric is the relative error $\epsilon$ of the estimated sample covariance matrix to its ground truth (see a definition in Appendix F), which is plotted in Figure 8. As shown in the left panel of the figure, the relative error increases mildly as the sequence length increases, which aligns with our $\sqrt{N}$-dependence in Theorem 2, demonstrating DiT's strength in handling long sequences. The right panel of the figure shows that the performance of DiT improves as the number of transformer blocks increases, yet at a marginally diminishing speed when $L$ is sufficiently large. Using 8 transformer blocks suffices for an efficient learning in this case, which approximately verifies our construction of a transformer with $\mathcal{O}(\log N)$ blocks.

**Training dynamic of DiT**    To further examine how DiT captures spatial-temporal dependencies, we present the training dynamic of DiT. As shown in Figure 9, the spatial correlation is captured relatively accurately at an early stage of the training (after the first few epochs), while the temporal correlation is learned slower (after epoch 50). We observe the gradual change in the temporal pattern to match eventually the ground truth. In particular, large temporal dependencies are easier to learn, while weak temporal dependencies require more epochs to learn.

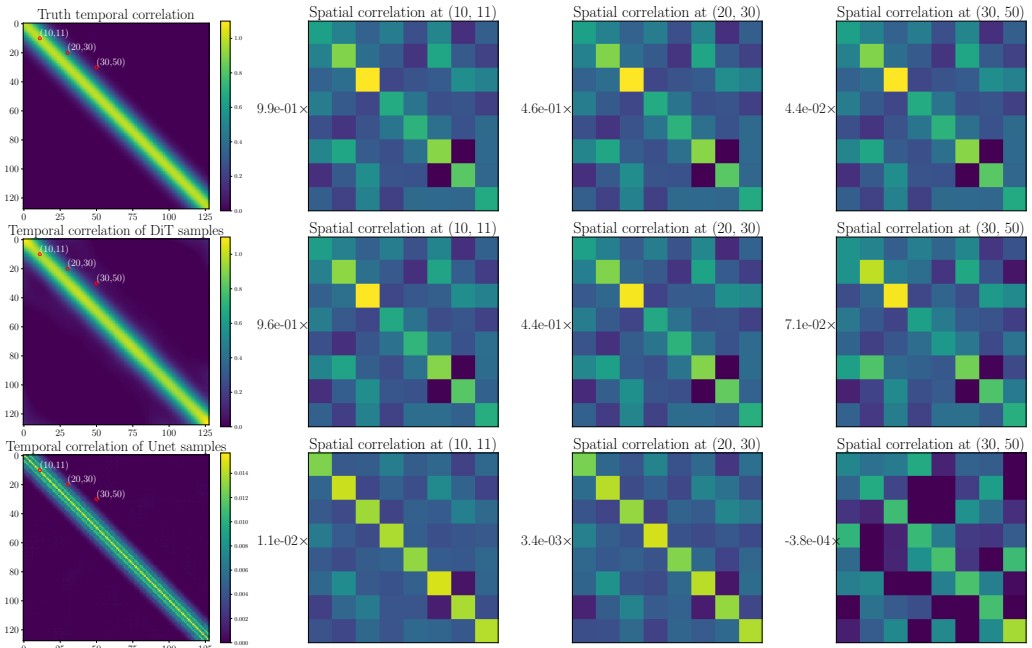

Figure 7: Visualization of ground truth spatial-temporal dependencies (the first row), spatial-temporal dependencies of the DiT-generated samples (the second row), and the UNet-generated samples (the third row). We visualize the temporal dependencies in the first column, while visualize the spatial dependencies of three pairs $(i,j) \in \{(10,11),(20,30),(30,50)\}$ of tokens respectively, representing short-horizon, medium-horizon, and long-horizon spatial dependencies. For each pair $(i,j)$, we compute the (empirical) covariance matrix between the $i$-th token and the $j$-token and visualize the matrix in a normalized version $\widehat{\gamma}_{i,j} \times \widehat{\boldsymbol{\Sigma}}$, where we set $\|\widehat{\boldsymbol{\Sigma}}\|_{\mathrm{F}} = \|\boldsymbol{\Sigma}\|_{\mathrm{F}}$ for a better comparison with the true spatial covariance $\boldsymbol{\Sigma}$.

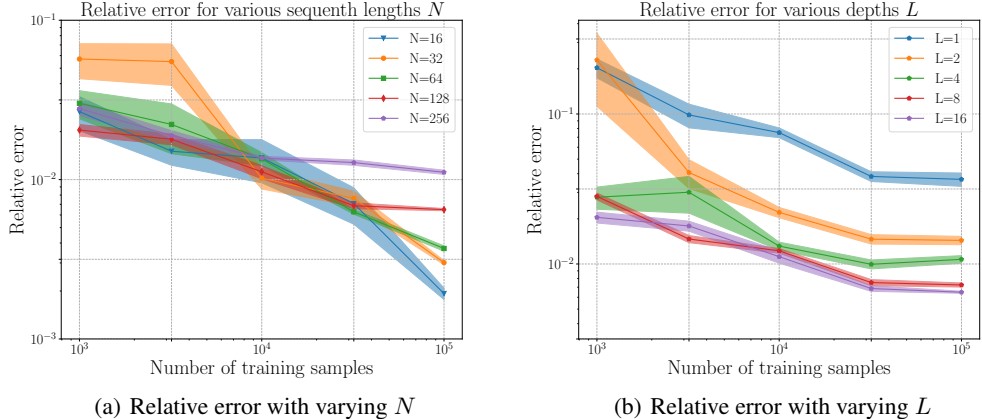

(a) Relative error with varying $N$

(b) Relative error with varying $L$

Figure 8: Relative error with different sequence lengths and numbers of transformer blocks. We observe a mild error increase when increasing the length of the sequence. Meanwhile, increasing the number of blocks in transformer leads to an initial gain of the performance. However, the performance gain saturated when the number of transformer blocks is sufficiently large. These observations support our theoretical results.

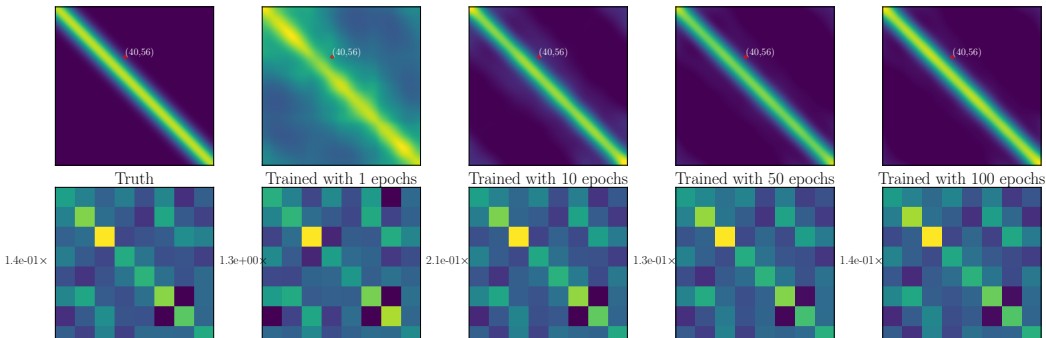

Figure 9: Spatial-temporal correlations of samples generated by DiT trained with different number of epochs. We train DiT with 10000 samples and use Adam optimizer with a mini-batch size of 128. We visualize the temporal correlation in the first row, and in the second row, we plot the estimated spatial correlation between the 40-th token and the 60-token using generated sequences.

## B    RELATED WORK

Our work establishes score approximation and distribution estimation theories of diffusion transformer with sequential data. Prior works focus on static data and provide sampling and learning guarantees of diffusion models. In particular, assuming access to a relatively accurate estimated score function, Benton et al. (2022; 2023; 2024); Li et al. (2024a; 2023); Chen et al. (2022b); Lee et al. (2022a;b); Chen et al. (2023c;b) show that the generated distribution of diffusion models stays close to the ground-truth distribution. Towards an end-to-end analysis, i.e., involving the score estimation procedure, Chen et al. (2023a); Oko et al. (2023); Li et al. (2024b); Mei and Wu (2023); Tang and Yang (2024); Jiao et al. (2024) all provide sample complexity bounds of diffusion models for various types of data, including manifold data and graphical models. Yet these results are not directly applicable to understanding how spatial-temporal dependencies are captured by diffusion transformers in sequential data.

For score approximation using transformers, we adopt an algorithm unrolling approach (Monga et al., 2021). In particular, we view the score function as the last iterate of a gradient descent algorithm and utilize transformers to implement gradient descent iterations. We are aware of Mei and Wu (2023); Mei (2024) showing U-Net performing algorithm unrolling in diffusion models.

On the empirical side, Diffusion Transformer (DiT) (Peebles and Xie, 2023) challenges the common choice of U-Net for image generation, providing state-of-the-art performance. Moreover, diffusion transformer exhibits appealing scalability towards better generation qualities with larger model sizes. More recently, diffusion transformers are leveraged in video generation (Gupta et al., 2023; Liu et al., 2024b), where the video data is patchified and rearranged into a long sequence. Besides, diffusion transformers are also used for other sequential data (Sun et al., 2022; Austin et al., 2021; Campbell et al., 2022; Gao et al., 2024), such as language, music and financial data.

## C    OMITTED PROOFS IN SECTION 3

Before diving into detailed proofs of our results (Theorem 1 and Theorem 2), we list our assumptions for a quick reference.

• **Data assumption**. We consider Gaussian process data in $\mathbb{R}^d$ within interval $[0, H]$, which is defined in Section 2. The covariance function of the underlying Gaussian process verifies Assumption 1. For technical convenience, we also assume the mean function $\boldsymbol{\mu}(\cdot)$ of the Gaussian process can be efficiently represented by neural networks in Assumption 2.

• **Transformer architecture**. We denote a transformer architecture by $\mathcal{T}(D, L, M, B, R)$, where $D, L, M, B, R$ are hyperparameters defining the size of the transformer. In Theorem 1, we will choose these hyperparameters depending on the desired approximation accuracy. In Theorem 2, we will further choose these hyperparameters depending on the training sample size $n$.

## C.1 Deriving Score Function for Gaussian Process Data

By the definition of Gaussian process, we know that the stacking vector $[\mathbf{x}_1^\top, \ldots, \mathbf{x}_N^\top]^\top \in \mathbb{R}^{dN}$ follows a Gaussian distribution $\mathsf{N}(\boldsymbol{\mu}, \boldsymbol{\Gamma} \otimes \boldsymbol{\Sigma})$. Along the forward process, we progressively add Gaussian noise to the initial data distribution and hence, for any $\mathbf{v}_t$, we have

$$p_t(\mathbf{v}_t) = \int \underbrace{\frac{1}{(2\pi\sigma_t^2)^{dN/2}} \exp\left(-\frac{1}{2\sigma_t^2}\|\mathbf{v}_t - \alpha_t\mathbf{v}_0\|_2^2\right)}_{(A)}$$

$$\cdot \underbrace{\frac{1}{(2\pi)^{dN/2}\sqrt{\det(\boldsymbol{\Gamma}\otimes\boldsymbol{\Sigma})}} \exp\left(-\frac{1}{2}(\mathbf{v}_0 - \boldsymbol{\mu})^\top(\boldsymbol{\Gamma}\otimes\boldsymbol{\Sigma})^{-1}(\mathbf{v}_0 - \boldsymbol{\mu})\right)}_{(B)} \, \mathrm{d}\mathbf{v}_0.$$

Note that $(A)$ is the Gaussian transition kernel corresponding to the forward process and $(B)$ is the clean data density function. It is clear that $p_t(\mathbf{v}_t)$ is again a Gaussian distribution. By completing the squares and some algebraic manipulation, we have

$$p_t(\mathbf{v}_t) \propto \int \exp\left(-\frac{1}{2}(\mathbf{v}_t - \alpha_t\boldsymbol{\mu})^\top(\sigma_t^2\mathbf{I} + \alpha_t^2\boldsymbol{\Gamma}\otimes\boldsymbol{\Sigma})^{-1}(\mathbf{v}_t - \alpha_t\boldsymbol{\mu})\right).$$

As a sanity check, $p_t$ is now a Gaussian density function of $\mathsf{N}(\alpha_t\boldsymbol{\mu}, \sigma_t^2\mathbf{I} + \alpha_t^2\boldsymbol{\Gamma}\otimes\boldsymbol{\Sigma})$, matching the marginal distribution of the forward process. Therefore, the score function is

$$\nabla \log p_t(\mathbf{v}_t) = -\left(\alpha_t^2\boldsymbol{\Gamma}\otimes\boldsymbol{\Sigma} + \sigma_t^2\mathbf{I}\right)^{-1}(\mathbf{v}_t - \alpha_t\boldsymbol{\mu}).$$

## C.2 Proof of Lemma 1

To prove the Lemma, we first state a standard result in convex optimization.

**Lemma 2** (Theorem 3.12 in Bubeck (2015)). Let $f$ be $\beta$-smooth and $\alpha$-strongly convex on $\mathbb{R}^d$ and $\mathbf{x}^\star$ is the global minimizer. Then gradient descent with $\eta = \frac{2}{\alpha+\beta}$ satisfies

$$\left\|\mathbf{x}^{(k+1)} - \mathbf{x}^\star\right\|_2 \leq \left(\frac{\kappa - 1}{\kappa + 1}\right)\left\|\mathbf{x}^{(k)} - \mathbf{x}^\star\right\|_2, \quad k = 0, 1, \ldots.$$

Here $\mathbf{x}^{(k+1)} = \mathbf{x}^{(k)} - \eta\nabla f(\mathbf{x}^{(k)})$ is the outcome in $(k+1)$-th iteration of GD and $\kappa = \beta/\alpha$.

With the lemma above, the proof of Lemma 1 is quite straightworward.

*Proof of Lemma 1.* Denote the truncated score function by

$$\bar{\mathbf{s}}(\mathbf{v}_t) = -\left(\alpha_t^2(\bar{\boldsymbol{\Gamma}}\otimes\boldsymbol{\Sigma}) + \sigma_t^2\mathbf{I}\right)^{-1}(\mathbf{v}_t - \alpha_t\boldsymbol{\mu}).$$

Let's consider the following quadratic target function

$$\bar{\mathcal{L}}_t(\mathbf{s}) = \frac{1}{2}\mathbf{s}^\top\left(\alpha_t^2(\bar{\boldsymbol{\Gamma}}\otimes\boldsymbol{\Sigma}) + \sigma_t^2\mathbf{I}\right)\mathbf{s} + (\mathbf{v}_t - \alpha_t\boldsymbol{\mu})^\top\mathbf{s},$$

And we iterate $\mathbf{s}^{(k)}$ by gradient descent on this target function with $\mathbf{s}^{(0)} = \mathbf{0}$. Since $\bar{\mathcal{L}}_t$ is $\lambda_{\max}\left(\alpha_t^2(\bar{\boldsymbol{\Gamma}}\otimes\boldsymbol{\Sigma}) + \sigma_t^2\mathbf{I}\right)$-smooth and $\lambda_{\min}\left(\alpha_t^2(\bar{\boldsymbol{\Gamma}}\otimes\boldsymbol{\Sigma}) + \sigma_t^2\mathbf{I}\right)$-strongly convex, by the approximate GD formula (4) and Lemma 1, for any $k \geq 0$, we have

$$\left\|\mathbf{s}^{(k+1)}(\mathbf{v}_t) - \bar{\mathbf{s}}(\mathbf{v}_t)\right\|_2 \leq \left(\frac{\bar{\kappa}_t - 1}{\bar{\kappa}_t + 1}\right)\left\|\mathbf{s}^{(k)}(\mathbf{v}_t) - \bar{\mathbf{s}}(\mathbf{v}_t)\right\|_2 \leq \exp\left(-\frac{2(k+1)}{\bar{\kappa}_t + 1}\right)\|\bar{\mathbf{s}}(\mathbf{v}_t)\|_2.$$

Here $\bar{\kappa}_t$ is the condition number of $\alpha_t^2(\bar{\boldsymbol{\Gamma}}\otimes\boldsymbol{\Sigma}) + \sigma_t^2\mathbf{I}$. Moreover, we have

$$\begin{aligned}
\|\bar{\mathbf{s}}(\mathbf{v}_t)\|_2 &\leq \left\|\left(\alpha_t^2(\bar{\boldsymbol{\Gamma}}\otimes\boldsymbol{\Sigma}) + \sigma_t^2\mathbf{I}\right)^{-1}\right\|_2 \|\mathbf{v}_t - \alpha_t\boldsymbol{\mu}\|_2 \\
&= \lambda_{\min}^{-1}\left(\alpha_t^2(\bar{\boldsymbol{\Gamma}}\otimes\boldsymbol{\Sigma}) + \sigma_t^2\mathbf{I}\right)\|\mathbf{v}_t - \alpha_t\boldsymbol{\mu}\|_2 \\
&\leq \frac{\|\mathbf{v}_t - \alpha_t\boldsymbol{\mu}\|_2}{\sigma_t^2}
\end{aligned}$$

Thus, by taking the number of iterations $K = \lceil \frac{\bar{\kappa}_t + 1}{2} \log(1/\epsilon) \rceil$, we have

$$\left\| \mathbf{s}^{(K)}(\mathbf{v}_t) - \bar{\mathbf{s}}(\mathbf{v}_t) \right\|_2 \le \frac{\|\mathbf{v}_t - \alpha_t \boldsymbol{\mu}\|_2 \, \epsilon}{\sigma_t^2}.$$

Besides, denote $\boldsymbol{\Phi}_t = \alpha_t^2(\boldsymbol{\Gamma} \otimes \boldsymbol{\Sigma}) + \sigma_t^2 \mathbf{I}$ and $\bar{\boldsymbol{\Phi}}_t = \alpha_t^2(\bar{\boldsymbol{\Gamma}} \otimes \boldsymbol{\Sigma}) + \sigma_t^2 \mathbf{I}$. The difference between the truth score function and the truncated score function is

$$\begin{aligned}
\|\bar{\mathbf{s}}(\mathbf{v}_t) - \mathbf{s}(\mathbf{v}_t)\|_2 &= \left\| \left( \left( \alpha_t^2(\bar{\boldsymbol{\Gamma}} \otimes \boldsymbol{\Sigma}) + \sigma_t^2 \mathbf{I} \right)^{-1} - \left( \alpha_t^2(\boldsymbol{\Gamma} \otimes \boldsymbol{\Sigma}) + \sigma_t^2 \mathbf{I} \right)^{-1} \right)(\mathbf{v}_t - \alpha_t \boldsymbol{\mu}) \right\|_2 \\
&= \left\| \left( \bar{\boldsymbol{\Phi}}_t^{-1} - \boldsymbol{\Phi}_t^{-1} \right)(\mathbf{v}_t - \alpha_t \boldsymbol{\mu}) \right\|_2 \\
&\le \left\| \bar{\boldsymbol{\Phi}}_t^{-1} - \boldsymbol{\Phi}_t^{-1} \right\|_2 \|\mathbf{v}_t - \alpha_t \boldsymbol{\mu}\|_2 \\
&= \left\| \boldsymbol{\Phi}_t^{-1} \left( \boldsymbol{\Phi}_t - \bar{\boldsymbol{\Phi}}_t \right) \bar{\boldsymbol{\Phi}}_t^{-1} \right\|_2 \|\mathbf{v}_t - \alpha_t \boldsymbol{\mu}\|_2 \\
&\le \left\| \boldsymbol{\Phi}_t^{-1} \right\|_2 \left\| \bar{\boldsymbol{\Phi}}_t^{-1} \right\|_2 \left\| \boldsymbol{\Phi}_t - \bar{\boldsymbol{\Phi}}_t \right\|_2 \|\mathbf{v}_t - \alpha_t \boldsymbol{\mu}\|_2 \\
&\le \sigma_t^{-4} \left\| \boldsymbol{\Phi}_t - \bar{\boldsymbol{\Phi}}_t \right\|_2 \|\mathbf{v}_t - \alpha_t \boldsymbol{\mu}\|_2.
\end{aligned}$$

Let's focus on bounding $\left\| \boldsymbol{\Phi}_t - \bar{\boldsymbol{\Phi}}_t \right\|_2$. Without any assumptions on $\boldsymbol{\Phi}_t$, a natural bound is

$$\left\| \boldsymbol{\Phi}_t - \bar{\boldsymbol{\Phi}}_t \right\|_2 \le \left\| \boldsymbol{\Phi}_t - \bar{\boldsymbol{\Phi}}_t \right\|_{\mathrm{F}} = \alpha_t^2 \|\boldsymbol{\Sigma}\|_{\mathrm{F}} \sqrt{\sum_{|i-j|>J} \boldsymbol{\Gamma}_{ij}^2}. \tag{8}$$

Altogether, we have

$$\begin{aligned}
\left\| \mathbf{s}^{(K)}(\mathbf{v}_t) - \nabla \log p_t(\mathbf{v}_t) \right\|_2 &\le \frac{\|\mathbf{v}_t - \alpha_t \boldsymbol{\mu}\|_2 \, \epsilon}{\sigma_t^2} + \sigma_t^{-4} \alpha_t^2 \|\boldsymbol{\Sigma}\|_{\mathrm{F}} \sqrt{\sum_{|i-j|>J} \boldsymbol{\Gamma}_{ij}^2} \, \|\mathbf{v}_t - \alpha_t \boldsymbol{\mu}\|_2 \\
&\le \left( \epsilon + \frac{\|\boldsymbol{\Sigma}\|_{\mathrm{F}}}{\sigma_t^2} \sqrt{\sum_{|i-j|>J} \boldsymbol{\Gamma}_{ij}^2} \right) \sigma_t^{-2} \|\mathbf{v}_t - \alpha_t \boldsymbol{\mu}\|_2. \tag{9}
\end{aligned}$$

In the last inequality, we invoke $\alpha_t^2 \le 1$ for any $t \ge 0$. The proof is complete. $\square$

**GD with Approximation Error** In the process of using transformers to express GD, additional approximation error will be induced, i.e., the update formula becomes

$$\mathbf{s}^{(k+1)} = \mathbf{s}^{(k)} - \eta \nabla \mathcal{L}_t(\mathbf{s}^{(k+1)}) + \boldsymbol{\xi}^{(k)}.$$

Here $\boldsymbol{\xi}^{(k)}$ represents the approximation error. Then we have the following convergence analysis:

**Lemma 3.** Given $\epsilon > 0$, if the approximation error in each step satisfies $\left\| \boldsymbol{\xi}^{(k)} \right\|_2 \le \epsilon$, then after $K = \lceil \frac{\kappa_t + 1}{2} \log(1/\epsilon) \rceil$ steps of GD on minimizing the truncated target function $\bar{\mathcal{L}}_t$, we have

$$\left\| \mathbf{s}^{(K)}(\mathbf{v}_t) - \nabla \log p_t(\mathbf{v}_t) \right\|_2 \le \frac{(\kappa_t + 1)\epsilon}{2} + \left( \epsilon + \frac{\|\boldsymbol{\Sigma}\|_{\mathrm{F}}}{\sigma_t^2} \sqrt{\sum_{|i-j|>J} \boldsymbol{\Gamma}_{ij}^2} \right) \sigma_t^{-2} \|\mathbf{v}_t - \alpha_t \boldsymbol{\mu}\|_2.$$

*Proof of Lemma 3.* By a similar deduction in the proof of Lemma 1, for any $k \ge 0$, we have

$$\begin{aligned}
\left\| \mathbf{s}^{(k+1)} - \bar{\mathbf{s}} \right\|_2 &= \left\| \mathbf{s}^{(k)} - \eta \nabla \mathcal{L}_t(\mathbf{s}^{(k+1)}) + \boldsymbol{\xi}^{(k)} - \bar{\mathbf{s}} \right\|_2 \\
&\le \left\| \mathbf{s}^{(k)} - \eta \nabla \mathcal{L}_t(\mathbf{s}^{(k+1)}) - \bar{\mathbf{s}} \right\|_2 + \left\| \boldsymbol{\xi}^{(k)} \right\|_2 \\
&\le \left( \frac{\kappa_t - 1}{\kappa_t + 1} \right) \left\| \mathbf{s}^{(k)} - \bar{\mathbf{s}} \right\|_2 + \epsilon.
\end{aligned}$$

Thus, we have

$$\left\| \mathbf{s}^{(k+1)} - \bar{\mathbf{s}} \right\|_2 - \frac{(\kappa_t + 1)\epsilon}{2} \le \left( \frac{\kappa_t - 1}{\kappa_t + 1} \right) \left( \left\| \mathbf{s}^{(k)} - \bar{\mathbf{s}} \right\|_2 - \frac{(\kappa_t + 1)\epsilon}{2} \right).$$

Note that if there exists $k \leq K$ such that $\left\|\mathbf{s}^{(k)} - \bar{\mathbf{s}}\right\|_2 \leq (\kappa_t + 1)\epsilon/2$, we have for any $k_1 \geq k$, $\left\|\mathbf{s}^{(k_1)} - \bar{\mathbf{s}}\right\|_2 \leq (\kappa_t + 1)\epsilon/2$, so $\left\|\mathbf{s}^{(K)} - \bar{\mathbf{s}}\right\|_2 \leq (\kappa_t + 1)\epsilon/2$, which finishes the proof. Now we assume for any $0 \leq k \leq K$, $\left\|\mathbf{s}^{(K)} - \bar{\mathbf{s}}\right\|_2 \geq (\kappa_t + 1)\epsilon/2$. Then we have

$$
\begin{aligned}
\left\|\mathbf{s}^{(k)} - \bar{\mathbf{s}}\right\|_2 - \frac{(\kappa_t + 1)\epsilon}{2} &\leq \left(\frac{\kappa_t - 1}{\kappa_t + 1}\right)^k \left(\left\|\mathbf{s}^{(0)} - \bar{\mathbf{s}}\right\|_2 - \frac{(\kappa_t + 1)\epsilon}{2}\right) \\
&\leq \left(\frac{\kappa_t - 1}{\kappa_t + 1}\right)^k \left\|\mathbf{s}^{(0)} - \bar{\mathbf{s}}\right\|_2 \\
&\leq \exp\left(-\frac{2k}{\kappa_t + 1}\right) \left\|\mathbf{s}^{(0)} - \bar{\mathbf{s}}\right\|_2.
\end{aligned}
\tag{10}
$$

Substituting $k = K = \lceil \frac{\kappa_t + 1}{2} \log(1/\epsilon) \rceil$ into the inequality, we have

$$
\left\|\mathbf{s}^{(K)} - \bar{\mathbf{s}}\right\|_2 - \frac{(\kappa_t + 1)\epsilon}{2} \leq \epsilon \left\|\bar{\mathbf{s}}(\mathbf{v}_t)\right\|_2 \leq \sigma_t^{-2} \left\|\mathbf{v}_t - \alpha_t \boldsymbol{\mu}\right\|_2 \epsilon.
$$

Combining the result with (8), we complete our proof. $\qquad\square$

### C.3 PROOF OF COROLLARY 1

Let's first analyse the approximation error by truncating $\mathbf{\Gamma}$. For simplicity, we denote $\Delta\mathbf{\Gamma} = \mathbf{\Gamma} - \bar{\mathbf{\Gamma}}$.

**Lemma 4.** Suppose Assumption 1 holds. Then for any $\epsilon > 0$, by taking $J = \left\lceil \left(\frac{\ell}{2} \log(N\ell/\epsilon^2)\right)^{1/\nu} \right\rceil + 1$, we have $\|\Delta\mathbf{\Gamma}\|_{\mathrm{F}} \leq \epsilon$.

*Proof of Lemma 4.* By Assumption 1, we have $f(m) \geq cm$. According to the definition of $\Delta\mathbf{\Gamma}$, we have

$$
\begin{aligned}
\|\Delta\mathbf{\Gamma}\|_{\mathrm{F}}^2 &= \sum_{|i-j| \geq J} \gamma(h_i, h_j)^2 \\
&= \sum_{k=J}^{N-1} (2N - 2k) \exp\left(-\frac{2f(k)^\nu}{\ell}\right) \\
&\leq \sum_{k=J}^{N-1} (2N - 2k) \exp\left(-\frac{2(ck)^\nu}{\ell}\right) \\
&\leq 2N \sum_{k=J}^{N-1} \exp\left(-\frac{2c^\nu k^\nu}{\ell}\right) \\
&\leq 2N \int_{J-1}^{\infty} \exp\left(-\frac{2c^\nu t^\nu}{\ell}\right) \mathrm{d}t \\
&\leq c^{-\nu} N\ell \exp\left(-\frac{2c^\nu (J-1)^\nu}{\ell}\right).
\end{aligned}
$$

by taking $J = \left\lceil \left(\frac{\ell}{2c^\nu} \log(N\ell/\epsilon^2 c^\nu)\right)^{1/\nu} \right\rceil + 1 = \mathcal{O}(\ell \log(N/\epsilon)^{1/\nu})$, we ensure that $\|\Delta\mathbf{\Gamma}\|_{\mathrm{F}}^2 \leq \epsilon^2$. The proof is complete. $\qquad\square$

**Remark 1.** From the proof of Lemma 4, we also observe that $\mathbf{\Gamma}$ is diagonally dominant, i.e., $\mathbf{\Gamma}_{ii} \geq \sum_{j \neq i} |\mathbf{\Gamma}_{ij}|$, when we have $2 \sum_{k=1}^{N-1} \exp\left(-\frac{2f(k)^\nu}{\ell}\right) \leq 1$. According to the proof above, a sufficient condition for this to hold is $c^{-\nu}\ell \leq 1$, i.e., $\ell \leq c^\nu$. When truncating $\mathbf{\Gamma}$ by any length $J$, $\bar{\mathbf{\Gamma}}$ remians diagonally dominant, therefore, $\bar{\mathbf{\Gamma}}$ is positive semidefinite.

Now we turn back to the proof of Corollary 1. By taking

$$
J = \left\lceil \left(\frac{\ell}{2c^\nu} \log(N\ell/(\sigma_t^4 \epsilon^2 c^\nu))\right)^{1/\nu} \right\rceil = \mathcal{O}\left((\ell \log(N/(\epsilon\sigma_t)))^{1/\nu}\right),
$$

we ensure that $\|\Delta\boldsymbol{\Gamma}\|_F^2 \leq \sigma_t^4 \epsilon^2$. Then plugging the approximation error into (9) concludes our proof.

Moreover, we could bound the operator norm of $\boldsymbol{\Gamma}$ in the same way. For any $\mathbf{v} \in \mathbb{R}^N$ such that $\|\mathbf{v}\|_2 = 1$, we have

$$
\begin{aligned}
\left|\mathbf{v}^\top \boldsymbol{\Gamma} \mathbf{v}\right| &= \sum_{i,j} \boldsymbol{\Gamma}_{i,j} v_i v_j \\
&= 1 + \sum_{k=1}^{N-1} 2 \exp(-2c^\nu k^\nu/\ell) \sum_{i=1}^{N-k} v_i v_{i+k} \\
&\leq 1 + 2 \sum_{k=1}^{N-1} \exp(-2c^\nu k^\nu/\ell) \sqrt{\sum_{i=1}^{N-k} v_i^2 \sum_{i=1}^{N-k} v_{i+k}^2} \\
&\leq 1 + 2 \sum_{k=1}^{N-1} \exp(-2c^\nu k^\nu/\ell) \\
&\leq 1 + c^{-\nu}\ell.
\end{aligned}
$$

Thus, we have

$$
\|\boldsymbol{\Gamma}\|_2 \leq 1 + c^{-\nu}\ell \lesssim 1 + \ell. \tag{11}
$$

We will apply this bound in Lemma 12.

## D OMITTED PROOFS IN SECTION 4

### D.1 PROOF OF THEOREM 1

By the previous derivation, we know the truth score function is written as

$$
\nabla \log p_t(\mathbf{v}_t) = -\left(\alpha_t^2 (\boldsymbol{\Gamma} \otimes \boldsymbol{\Sigma}) + \sigma_t^2 \mathbf{I}\right)^{-1} (\mathbf{v}_t - \alpha_t \boldsymbol{\mu}). \tag{12}
$$

Due to the fast decay of $\boldsymbol{\Gamma}$, we consider a truncated score function

$$
\bar{\mathbf{s}}(\mathbf{v}_t) = -\left(\alpha_t^2 (\bar{\boldsymbol{\Gamma}} \otimes \boldsymbol{\Sigma}) + \sigma_t^2 \mathbf{I}\right)^{-1} (\mathbf{v}_t - \alpha_t \boldsymbol{\mu})
$$

with $\bar{\boldsymbol{\Gamma}}_{ij} = \boldsymbol{\Gamma}_{ij}$ if $|i - j| < J$ and $\bar{\boldsymbol{\Gamma}}_{ij} = 0$ otherwise. Denoting $\boldsymbol{\Gamma} = \bar{\boldsymbol{\Gamma}} + \Delta\boldsymbol{\Gamma}$ (see the formal statement in Lemma 4), by appropriately choosing $M = \mathcal{O}((\log(1/\epsilon))^{1/v})$, we could guarantee $\|\Delta\boldsymbol{\Gamma}\| \leq \epsilon$ for any $\epsilon > 0$. Now we decompose the $L_2$ error into the following two items:

$$
\|\widetilde{\mathbf{s}} - \nabla \log p_t\|_{L_2(P_t)} \leq \underbrace{\|\widetilde{\mathbf{s}} - \bar{\mathbf{s}}\|_{L_2(P_t)}}_{\text{Proposition 1}} + \underbrace{\|\bar{\mathbf{s}} - \nabla \log p_t\|_{L_2(P_t)}}_{\text{Lemma 5}}.
$$

Here $\|\mathbf{s}\|_{L_2(P_t)}^2 = \int \|\mathbf{s}(\mathbf{v}_t)\|_2^2 p_t(\mathbf{v}_t) \mathrm{d}t$ is the squared $L_2$ norm w.r.t. a density function $p_t$ of distribution $P_t$. We provide the error analysis for the two items in Proposition 1 and Lemma 5 separately. Under these two supporting statements, the proof of Theorem 1 is quite straightforward.

Now we state Proposition 1 and Lemma 5 with their detailed proof deferred to next sections. Proposition 1 provides an approximation guarantee on the difference between $\widetilde{\mathbf{s}}$ and $\bar{\mathbf{s}}$ in $L_2(P_t)$, which is our main result throughout the proof of Theorem 1.

**Proposition 1.** Suppose Assumption 1 holds. Given $t_0 \in (0, T]$, there exists a transformer architecture $\widetilde{\mathbf{s}} \in \mathcal{T}(D, L, M, B, R_t)$ such that with proper weight parameters, it yields an approximation $\widetilde{\mathbf{s}}$ to the truncated score function $\bar{\mathbf{s}}$ with

$$
\|\widetilde{\mathbf{s}} - \bar{\mathbf{s}}\|_{L_2(P_t)}^2 \leq \frac{\epsilon^2}{\sigma_t^2}, \quad t \geq t_0.
$$

The transformer architecture satisfies

$$
D = 9d + d_t + d_e + 1, L = \mathcal{O}\left(\kappa_{t_0} \log(Nd/(\epsilon\sigma_{t_0}))^2\right), M = \mathcal{O}\left((\ell \log(Nd \|\boldsymbol{\Sigma}\|_F /(\epsilon\sigma_{t_0})))^{1/\nu}\right),
$$

$$
B = \mathcal{O}\left(\log(Nd/(\epsilon\sigma_{t_0}))\sigma_{t_0}^{-2} Nd(r^2 + \|\boldsymbol{\Sigma}\|_\infty)\right), R_t = \mathcal{O}\left(\log(Nd/(\epsilon\sigma_{t_0}))\sqrt{Nd}/\sigma_t\right).
$$

The proof is provided in Appendix D.2.

Moreover, Lemma 5 bounds the difference between the truncated score function $\bar{\mathbf{s}}$ and the ground truth in $L_2(P_t)$ distance.

**Lemma 5.** Given any $\epsilon > 0$ and $t > 0$, by choosing

$$J = \mathcal{O}\Big( \big( \ell \log(N \left\| \mathbf{\Sigma} \right\|_{\mathrm{F}} / (\epsilon \sigma_t)) \big)^{1/\nu} \Big),$$

it holds that for any $t > 0$,

$$\left\| \bar{\mathbf{s}} - \nabla \log p_t \right\|_{L_2(P_t)} \leq \frac{\epsilon}{\sigma_t}.$$

The proof is provided in Appendix D.4.

Now back to the proof of Theorem 1, recall that we can decompose the score error as in (12). Thus, combining Proposition 1 and Lemma 5 and adjusting the constants to bound the error by $\epsilon/\sigma_t$, we finish the proof of Theorem 1.

### D.2 PROOF OF PROPOSITION 1

To prove Proposition 1, we first need a uniform approximation theory of $\bar{\mathbf{s}}$ on a bounded region, which is the backbone of the proof.

**Lemma 6.** Given any radius $R_0 \geq 1$, error level $0 \leq \epsilon < 1$ and $t_0 > 0$, there exists a transformer architecture $\mathcal{T}(D, L, M, B, R)$ that gives rise to $\widetilde{\mathbf{s}}$ satisfying

$$\left\| \widetilde{\mathbf{s}}(\mathbf{v}_t) - \bar{\mathbf{s}}(\mathbf{v}_t) \right\|_2 \leq \epsilon, \quad \text{for any } \left\| \mathbf{s}(\mathbf{v}_t) \right\|_2 \leq R_0 \sigma_t^{-1}, \ \ t \geq t_0.$$

The transformer architecture satisfies

$$D = 9d + d_t + d_e + 1, \quad L = \mathcal{O}\big( \kappa_{t_0} \log(R_0 N d/(\epsilon \sigma_{t_0}))^2 \big), \quad M = 4J,$$
$$B = \mathcal{O}\Big( \log(R_0 N d/(\epsilon \sigma_{t_0})) \sqrt{N} R_0 d \sigma_{t_0}^{-2} (r^2 + \left\| \mathbf{\Sigma} \right\|_\infty) \Big), \quad R_t = \mathcal{O}\big( R_0 \sigma_t^{-1} \big).$$

The proof is provided in Appendix D.3. Then we could upper bound the second moments of the truncated score function $\bar{\mathbf{s}}$ in $P_t$.

**Lemma 7.** For any $0 \leq \epsilon < 1$, by setting

$$J > \left\lceil \left( \frac{1}{2} \ell \log(N \ell \left\| \mathbf{\Sigma} \right\|_{\mathrm{F}}^2 / \sigma_t^2) \right)^{1/\nu} \right\rceil,$$

we have the second moment of the truncated score function bounded by $\left\| \bar{\mathbf{s}} \right\|_{L_2(P_t)} \leq \frac{\sqrt{2Nd}}{\sigma_t}$.

Moreover, we need to derive a uniform upper bound of the true score function for convenience of truncation arguments.

**Lemma 8.** Suppose Assumption 1 holds, then with probability $1 - 2\exp(-C\delta)$, the range of truth score function can be bounded by $\left\| \nabla \log p_t(\mathbf{v}_t) \right\|_2^2 \leq \sigma_t^{-2} \Big( N + \delta \sqrt{Nd} \Big)$. Here $C$ is an absolute constant.

The proofs of the lemmas are provided in Appendix D.4. Now we are ready to prove Proposition 1.

*Proof of Proposition 1.* By Lemma 6, we obtain a transformers such that $\left\| \widetilde{\mathbf{s}}(\mathbf{v}_t) - \mathbf{s}(\mathbf{v}_t) \right\|_2 \leq \epsilon$ for any $\left\| \mathbf{s}(\mathbf{v}_t) \right\|_2 \leq R_0 \sigma_t^{-1}$ and $t \geq t_0$. Moreover, we have $\left\| \widetilde{\mathbf{s}}(\mathbf{v}_t) \right\|_2 \leq C R_0 \sigma_t^{-1}$ for some absolute

constant $C$. Now we can decompose the $L_2$ error as

$$
\begin{aligned}
\|\widetilde{\mathbf{s}} - \bar{\mathbf{s}}\|^2_{L_2(P_t)} &= \mathbb{E}_{\mathbf{v}_t}\left[\|\widetilde{\mathbf{s}}(\mathbf{v}_t) - \bar{\mathbf{s}}(\mathbf{v}_t)\|^2_2\right] \\
&\leq \mathbb{E}_{\mathbf{v}_t}\left[\|\widetilde{\mathbf{s}}(\mathbf{v}_t) - \bar{\mathbf{s}}(\mathbf{v}_t)\|^2_2 \, \mathbf{1}\left\{\|\mathbf{s}(\mathbf{v}_t)\|_2 \leq R_0 \sigma_t^{-1}\right\}\right] \\
&\quad + \mathbb{E}_{\mathbf{v}_t}\left[\|\widetilde{\mathbf{s}}(\mathbf{v}_t) - \bar{\mathbf{s}}(\mathbf{v}_t)\|^2_2 \, \mathbf{1}\left\{\|\mathbf{s}(\mathbf{v}_t)\|_2 > R_0 \sigma_t^{-1}\right\}\right] \\
&\leq \epsilon^2 + 2\mathbb{E}_{\mathbf{v}_t}\left[\frac{C^2 R_0^2}{\sigma_t^2} \mathbf{1}\left\{\|\mathbf{s}(\mathbf{v}_t)\|_2 > R_0 \sigma_t^{-1}\right\}\right] \\
&\quad + 2\mathbb{E}_{\mathbf{v}_t}\left[\|\bar{\mathbf{s}}(\mathbf{v}_t)\|^2_2 \mathbf{1}\left\{\|\mathbf{s}(\mathbf{v}_t)\|_2 > R_0 \sigma_t^{-1}\right\}\right] \\
&\leq \epsilon^2 + \frac{2C^2 R_0^2}{\sigma_t^2} \Pr\left[\|\mathbf{s}(\mathbf{v}_t)\|_2 > R_0 \sigma_t^{-1}\right] \\
&\quad + 2\mathbb{E}_{\mathbf{v}_t}\left[\|\bar{\mathbf{s}}(\mathbf{v}_t)\|^4_2\right]^{1/2} \Pr\left[\|\mathbf{s}(\mathbf{v}_t)\|_2 > R_0 \sigma_t^{-1}\right]^{1/2} \\
&\leq \epsilon^2 + \frac{1}{\sigma_t^2} 2C^2 R_0^2 \Pr\left[\|\mathbf{s}(\mathbf{v}_t)\|_2 > R_0 \sigma_t^{-1}\right] \\
&\quad + C_{2,4}\mathbb{E}_{\mathbf{v}_t}\left[\|\bar{\mathbf{s}}(\mathbf{v}_t)\|^2_2\right] \Pr\left[\|\mathbf{s}(\mathbf{v}_t)\|_2 > R_0 \sigma_t^{-1}\right]^{1/2} \\
&\leq \epsilon^2 + \frac{1}{\sigma_t^2}\left(2C^2(R_0 + \|\boldsymbol{\mu}\|_2^2) + C_{2,4} 2Nd\right) \Pr\left[\|\mathbf{s}(\mathbf{v}_t)\|_2 > R_0 \sigma_t^{-1}\right]^{1/2}.
\end{aligned}
$$

Here we invoke Lemma 25 in the second-to-last inequality and invoke Lemma 7 in the last inequality.

By Lemma 8, choosing $R_0 = C_1\sqrt{Nd}\log(\|\boldsymbol{\mu}\|_2 CNd\epsilon^{-1}\sigma_{t_0}^{-1})$ for some absolute constant $C_1$, we can bound $\Pr\left[\|\mathbf{s}(\mathbf{v}_t)\|_2 > R_0\sigma_t^{-1}\right]$ by

$$
\Pr\left[\|\mathbf{s}(\mathbf{v}_t)\|_2 > R_0\sigma_t^{-1}\right] \leq \frac{\epsilon^2}{4(R_0 + \|\boldsymbol{\mu}\|_2^2)C^2 Nd},
$$

thus bounding the $L_2$ error by $\mathcal{O}(\epsilon^2/\sigma_t^2)$. By adjusting the constant, we can ensure that there exists a transformer architecture $\mathcal{T}(D, L, M, B, R_t)$ that gives rise to $\widetilde{\mathbf{s}}$ with

$$
\begin{gathered}
D = 9d + d_t + d_e + 1, \quad L = \mathcal{O}\big(\kappa_{t_0}\log(R_0 Nd/(\epsilon\sigma_{t_0}))^2\big) = \mathcal{O}\big(\kappa_{t_0}\log(Nd/(\epsilon\sigma_{t_0}))^2\big), \\
M = \mathcal{O}\Big((\ell\log(dN\|\boldsymbol{\Sigma}\|_{\mathrm{F}}/(\epsilon\sigma_{t_0})))^{1/\nu}\Big), \quad B = \mathcal{O}\big(\log(Nd/(\epsilon\sigma_{t_0}))\sigma_{t_0}^{-2} Nd(r^2 + \|\boldsymbol{\Sigma}\|_\infty)\big), \\
R_t = \mathcal{O}\Big(\log(Nd/(\epsilon\sigma_{t_0}))\sqrt{Nd}/\sigma_t\Big)
\end{gathered}
$$

such that

$$
\|\widetilde{\mathbf{s}} - \bar{\mathbf{s}}\|^2_{L_2(P_t)} \leq \epsilon^2/\sigma_t^2.
$$

The proof is complete. $\qquad\square$

### D.3  PROOF OF LEMMA 6

### D.3.1  TRANSFORMER ARCHITECTURE

Following Figure 3, we construct our targeted transformer architecture as

$$
f = f_{\mathrm{out}} \circ f_{\mathrm{GD}} \circ \cdots \circ f_{\mathrm{GD}} \circ f_{\mathrm{pre}} \circ f_{\mathrm{in}}.
$$

**Encoder**  For simplicity, we suppose the encoder converts the initial input into $\mathbf{Y} = f_{\mathrm{in}}([\mathbf{x}_1, \mathbf{x}_2, \ldots, \mathbf{x}_N]) = [\mathbf{y}_1^\top, \ldots, \mathbf{y}_N^\top] \in \mathbb{R}^{D \times N}$ satisfies

$$
\mathbf{y}_i = [\mathbf{x}_i^\top, \mathbf{e}_i^\top, \boldsymbol{\phi}^\top(t), \mathbf{0}_{5d}^\top, 1, \mathbf{0}_{3d}^\top]^\top,
$$

where $\boldsymbol{\phi}(t) = [\eta_t, \alpha_t, \sigma_t^2, \alpha_t^2]^\top \in \mathbb{R}^{d_t}$ with $d_t = 4$. Here $\mathbf{0}_{5d}^\top$ and $\mathbf{0}_{3d}^\top$ serve as the buffer space for storing the components necessary for expressing gradient descent algorithm.

**Transformer Components**    Besides the encoder and decoder, $f_{\text{pre}}$ represents a multi-layer trans-formers that prepare the necessary components for the gradient descent block, such as the mean $\boldsymbol{\mu}_i$ for each input token $\mathbf{x}_i$. $f_{\text{GD}}$ represents a multi-layer transformers that approximately express one step of gradient descent, which is the key component of our network. We elaborate on the construction of these subnetworks in D.3.2.

**Decoder**    Suppose the output tokens from past layers has produced a score approximator in matrix shape, we design the decoder as follows:

$$f_{\text{out}} = f_{\text{norm}} \circ f_{\text{linear}},$$

where $f_{\text{linear}} : \mathbb{R}^{D \times N} \to \mathbb{R}^{dN}$ extracts a $d \times N$ block from the input and flattens it into a vector to align with the dimension of the score function. $f_{\text{norm}} : \mathbb{R}^{dN} \to \mathbb{R}^{dN}$ controls the output range of the network by $R_t$, which can be mathematically written as

$$f_{\text{norm}}(\mathbf{s}) = \begin{cases} \mathbf{s}, & \text{if } \|\mathbf{s}\|_2 \leq R_t, \\ \dfrac{R_t}{\|\mathbf{s}\|_2} \mathbf{s}, & \text{otherwise.} \end{cases}$$

We remark that such clipping strategy is also applied in other theoretical works Oko et al. (2023) to better adapt to the magnitude of the score function at different diffusion time $t$, since the score function of a degenerate Gaussian distribution could blow up as $t \to 0$. Moreover, such clipping layer can be easily expressed by finite layers of feed-forward networks if we take both $\mathbf{s}$ and $R_t$ as the input. See Equation 5 in Lemma 2 of Chen et al. (2022a) for an example of the construction.

**Raw Transformer Network**    In the proof of Lemma 6, we mainly focus on the transformer blocks instead of the encoders and decoders. We denote the raw transformer network as

$$\begin{aligned} \mathcal{T}(D, L, M, B) = \{ f : f = (\text{FFN}_L \circ \text{Attn}_L) \circ \cdots \circ (\text{FFN}_1 \circ \text{Attn}_1), \\ \text{The input and out dimension is } D, \\ \text{Attn}_i \text{ uses entrywise ReLU activation for } i = 1, \ldots, L, \\ \text{number of heads in each } \text{Attn} \text{ is bounded by } M, \\ \text{the Frobenius norm of each weight matrix is bounded by } B \}. \end{aligned}$$

### D.3.2    APPROXIMATE EACH TRANSFORMER SUBNETWORKS

For simplicity, for the subnetwork $f_{\text{pre}}$, we assume that the mean vectors $\boldsymbol{\mu}_i$ can be directly expressed by a constant number of transformer blocks:

**Assumption 2.** There exists a raw transformer $f_\mu \in \mathcal{T}_{\text{raw}}(D, L_\mu, M_\mu, \mathcal{O}(d))$ such that for any input token $\mathbf{y}_i = [\mathbf{x}_i^\top, \mathbf{e}_i^\top, \boldsymbol{\phi}^\top(t), \mathbf{0}_{5d}^\top, 1, \mathbf{0}_{3d}^\top]^\top$, we have

$$f_\mu(\mathbf{y}_i) = [\mathbf{x}_i^\top, \mathbf{e}_i^\top, \boldsymbol{\phi}^\top(t), \mathbf{0}_{4d}^\top, \boldsymbol{\mu}_i^\top, 1, \mathbf{0}_{3d}^\top]^\top.$$

Here $L_\mu$ and $M_\mu$ are all constants. This assumption is mild, given the approximation ability of transformers.

After applying $f_\mu$, we begin to implement gradient descent algorithm using transformer blocks. Starting at $\mathbf{s}_i^{(0)} = \mathbf{0}_d$ for $i \in [N]$, the first iteration can be written as

$$\mathbf{s}_i^{(1)} = \mathbf{s}_i^{(0)} - \eta_1 \left( \sum_{j=1}^N (\alpha_t^2 \bar{\boldsymbol{\Gamma}}_{ij} \boldsymbol{\Sigma}) \mathbf{s}_j^{(0)} + \sigma_t^2 \mathbf{s}_i^{(0)} + (\mathbf{x}_i - \alpha_t \boldsymbol{\mu}_i) \right) = -\eta_1 (\mathbf{x}_i - \alpha_t \boldsymbol{\mu}_i).$$

Thus, we only need apply a multiplication module $f_{\text{mult}}(\alpha_t, \boldsymbol{\mu}_i) \approx \alpha_t \boldsymbol{\mu}_i$ that approximately re-alizes the product operation. We defer the detailed construction of the multiplication module to Appendix G.3, which utilizes $\mathcal{O}(\log(\max_i \|\boldsymbol{\mu}_i\|_2 / \epsilon_{\text{mult}}))$ transformer blocks to reach the accuracy $\|f_{\text{mult}}(\alpha_t, \boldsymbol{\mu}_i) - \alpha_t \boldsymbol{\mu}_i\|_\infty \leq \epsilon_{\text{mult}}$ for any $i \in [N]$.

**Lemma 9** (Construct the first step of GD). Suppose the input is $\mathbf{Y} = [\mathbf{y}_1^\top, \ldots, \mathbf{y}_N^\top] \in \mathbb{R}^{D \times N}$, where each token is

$$\mathbf{y}_i = [\mathbf{x}_i^\top, \mathbf{e}_i^\top, \boldsymbol{\phi}^\top(t), \mathbf{0}_{4d}^\top, \boldsymbol{\mu}_i^\top, 1, \mathbf{0}_{3d}^\top]^\top \in \mathbb{R}^{9d + d_e + d_t + 1}.$$

Given error level $\epsilon < 1$ and learning rate $\eta_t > 0$, there exists a transformer $f_{\text{GD},1} \in \mathcal{T}_{\text{raw}}(D, L, M, B)$ such that

$$f_{\text{GD},1}(\mathbf{y}_i) = \left[ \mathbf{x}_i^\top, \mathbf{e}_i^\top, \boldsymbol{\phi}^\top(t), f_{\text{mult}}(\eta_t, \mathbf{x}_i - f_{\text{mult}}(\alpha_t, \boldsymbol{\mu}_i))^\top, \mathbf{0}_d^\top, \mathbf{0}_d^\top, \mathbf{0}_d^\top, \right.$$

$$\left. f_{\text{mult}}(\alpha_t, \boldsymbol{\mu}_i)^\top, 1, \mathbf{0}_{3d}^\top \right]^\top.$$

Here $\mu_0 = \|\boldsymbol{\mu}\|_\infty$, and the two multiplication modules satisfy $\left\| f_{\text{mult}}(\alpha_t, \boldsymbol{\mu}_i)^\top - \alpha_t \boldsymbol{\mu}_i \right\|_2 \le \epsilon/\sqrt{N}$ and also $\| f_{\text{mult}}(\eta_t, \mathbf{x}_i - f_{\text{mult}}(\alpha_t, \boldsymbol{\mu}_i)) - \eta_t(\mathbf{x}_i - \alpha_t \boldsymbol{\mu}_i) \|_2 \le \epsilon/\sqrt{N}$ for $i \in [N]$, and the parameters of the networks satisfy

$$D = 9d + d_e + d_t + 1, \quad L = \mathcal{O}(\log(\|\mathbf{x}\|_\infty \|\mathbf{s}\|_\infty Nd/\epsilon)),$$
$$M = 1, \quad B = \mathcal{O}(d(\|\mathbf{x}\|_\infty + \|\mathbf{s}\|_\infty)).$$

*Proof of Lemma 9.* To prove the lemma, we apply Corollary 3 and use the last $3d$ dimensions $\mathbf{0}_{3d}$ as the buffer space to approximate the product operation $f_{\text{mult}} : \mathbb{R}^D \to \mathbb{R}^D$ such that

$$f_{\text{mult}}(\mathbf{Y}) = \begin{bmatrix} \mathbf{x}_1 & \cdots & \mathbf{x}_N \\ \mathbf{e}_1 & \cdots & \mathbf{e}_N \\ \boldsymbol{\phi}(t) & \cdots & \boldsymbol{\phi}(t) \\ \mathbf{0}_{4d} & \cdots & \mathbf{0}_{4d} \\ f_{\text{mult}}(\alpha_t, \boldsymbol{\mu}_1) & \cdots & f_{\text{mult}}(\alpha_t, \boldsymbol{\mu}_N) \\ 1 & \cdots & 1 \\ \mathbf{0}_{3d} & \cdots & \mathbf{0}_{3d} \end{bmatrix},$$

where $f_{\text{mult}}(\alpha_t, \boldsymbol{\mu}_i) = \alpha_t \boldsymbol{\mu}_i + \boldsymbol{\epsilon}_{\mu,i}$ with $\|\boldsymbol{\epsilon}_{\mu,i}\|_2 \le \epsilon/\sqrt{N}$. After obtaining $f_{\text{mult}}$, we use one transformer block $\mathcal{TB} = \text{FFN} \circ \text{Attn}$ with the attention block being trivial, i.e., all weight parameters being zero so that $\text{Attn}(\mathbf{y}) = \mathbf{y}$. We then choose $\text{FFN}(\mathbf{y}) = \mathbf{y} + \mathbf{W}_2 \text{ReLU}(\mathbf{W}_1 \mathbf{y})$ with

$$\mathbf{W}_1 = \begin{bmatrix} -\mathbf{I}_d & \mathbf{0}_{d \times (4d + d_e + d_t)} & \mathbf{I}_d & \mathbf{0}_{d \times (3d+1)} \\ \mathbf{I}_d & \mathbf{0}_{d \times (4d + d_e + d_t)} & -\mathbf{I}_d & \mathbf{0}_{d \times (3d+1)} \end{bmatrix} \in \mathbb{R}^{(2d) \times D}$$

and

$$\mathbf{W}_2 = \begin{bmatrix} \mathbf{0}_{(d + d_e + d_t) \times (d)} & \mathbf{0}_{(d + d_e + d_t) \times (d)} \\ \mathbf{I}_d & -\mathbf{I}_d \\ \mathbf{0}_{(7d+1) \times (d)} & \mathbf{0}_{(7d+1) \times (d)} \end{bmatrix} \in \mathbb{R}^{D \times (2d)}.$$

We further apply another multiplication module $f_{\text{mult},1}$ to rescale the gradient computed by the transformer block by the learning rate $\eta_1$. We can check that the first-step GD is represented by $f_{\text{GD},1} = f_{\text{mult},1} \circ \mathcal{TB} \circ f_{\text{mult}}$.

**Size of Transformer Blocks for Approximating The First GD Iteration** We summarize in the following table the resulting network size of transformer blocks for implementing the first GD iteration when initialized with $\mathbf{s}_i = \mathbf{0}$.

Table 1: Transformer size for approximating the first GD iteration

| Input dimension | $D \times N$ with $D = 9d + d_e + d_t + 1$ |
|---|---|
| # of blocks $L$ | $1 + 2L_{\text{mult}}$ with $L_{\text{mult}} = \mathcal{O}(\log(\|\mathbf{x}\|_\infty \|\mathbf{s}\|_\infty Nd/\epsilon))$ |
| # of attention heads $M$ | 1 |
| Parameter bound $B$ | $\mathcal{O}(d(\|\mathbf{x}\|_\infty + \|\mathbf{s}\|_\infty))$ |

Here the parameter bound $B$ is obtained by noting that the magnitude of weight parameters in the multiplication module is at most $\mathcal{O}(\|\mathbf{x}\|_\infty + \|\mathbf{s}\|_\infty)$ and there are at most $\mathcal{O}(d)$ nonzero weight parameters in each weight matrix. The proof is complete. $\square$

The next lemma presents the construction of next $K - 1$ steps of GD.

**Lemma 10** (Construct next steps of GD). Suppose the input token is

$$\mathbf{y}_i = [\mathbf{x}_i^\top, \mathbf{e}_i^\top, \boldsymbol{\phi}^\top(t), \mathbf{s}_i^\top, \mathbf{0}_d^\top, \mathbf{0}_d^\top, \mathbf{0}_d^\top, f_{\mathrm{mult}}(\alpha_t, \boldsymbol{\mu}_i)^\top, 1, \mathbf{0}_{3d}^\top]^\top,$$

then there exists a transformer $f_{\mathrm{GD},2} \in \mathcal{T}_{\mathrm{raw}}(D, L, M, B)$ that approximately iterates $\mathbf{s}_i$ by following the GD update formula, i.e.,

$$f_{\mathrm{GD},2}(\mathbf{y}_i) = \left[\mathbf{x}_i^\top, \mathbf{e}_i^\top, \boldsymbol{\phi}^\top(t), \quad \mathbf{s}_i^\top - \eta_t \left(\sum_{j=1}^N (\alpha_t^2 \bar{\boldsymbol{\Gamma}}_{ij}\boldsymbol{\Sigma} + \sigma_t^2 \mathbf{I})\mathbf{s}_j^\top + (\mathbf{x}_i - \alpha_t \boldsymbol{\mu}_i)^\top\right) + \boldsymbol{\epsilon}_i^\top, \right.$$

$$\left. \mathbf{0}_d^\top, \mathbf{0}_d^\top, \mathbf{0}_d^\top, f_{\mathrm{mult}}(\alpha_t, \boldsymbol{\mu}_i)^\top, 1, \mathbf{0}_{3d}^\top\right]^\top.$$

Here $\|\boldsymbol{\epsilon}_i'\|_2 \le \epsilon/\sqrt{N}$ for $i \in [N]$. Moreover, the parameters of the networks satisfy

$$D = 9d + d_e + d_t + 1, \quad L = \mathcal{O}(\log(\|\mathbf{x}\|_\infty \|\mathbf{s}\|_\infty Nd/\epsilon)),$$

$$M = 4J, \quad B = \mathcal{O}\big(d(\|\boldsymbol{\Sigma}\|_\infty + r^2)\big) + \|\mathbf{x}\|_\infty + \|\mathbf{s}\|_\infty).$$

*Proof of Lemma 10.* It suffices to construct several transformer blocks to represent one gradient descent iteration with truncated matrix $\bar{\boldsymbol{\Gamma}}$, which takes the form

$$\mathbf{s}_i^{(k+1)} = \mathbf{s}_i^{(k)} - \eta_t \left[\sum_{j=1}^N (\alpha_t^2 \bar{\boldsymbol{\Gamma}}_{ij}\boldsymbol{\Sigma})\mathbf{s}_j^{(k)} + \sigma_t^2 \mathbf{s}_i^{(k)} + (\mathbf{x}_{i,t} - \alpha_t \boldsymbol{\mu}_i)\right]$$

$$= \mathbf{s}_i^{(k)} - \eta_t \underbrace{\sum_{j=1}^N \alpha_t^2 \bar{\boldsymbol{\Gamma}}_{ij}\boldsymbol{\Sigma}\mathbf{s}_j^{(k)}}_{(A)} - \underbrace{\left(\eta_t \sigma_t^2 \mathbf{s}_i^{(k)} + \eta_t(\mathbf{x}_{i,t} - \alpha_t \boldsymbol{\mu}_i)\right)}_{(B)}. \tag{13}$$

To ease the presentation, we consider a fixed time $t$ and drop the subscript $t$. We also drop the superscript $(k)$. In the following, we will use two transformer blocks to approximate $(A)$ and $(B)$ separately. The input to those transformer blocks is $\mathbf{Y} = [\mathbf{y}_1, \ldots, \mathbf{y}_N] \in \mathbb{R}^{D \times N}$, where $D$ is a larger dimension and will be specified shortly. For each column vector $\mathbf{y}_i$, it stores copies of $\mathbf{s}_i$ and other relevant information. Recall each token $\mathbf{y}_i$ is

$$\mathbf{y}_i = [\mathbf{x}_i^\top, \mathbf{e}_i^\top, \boldsymbol{\phi}^\top(t), \mathbf{s}_i^\top, \mathbf{0}_d^\top, \mathbf{0}_d^\top, \mathbf{0}_d^\top, f_{\mathrm{mult}}(\alpha_t, \boldsymbol{\mu}_i)^\top, 1, \mathbf{0}_{3d}^\top]^\top \in \mathbb{R}^{9d + d_e + d_t + 1}.$$

Here we reserve $\mathbf{0}_{3d}^\top$ as the buffer space for the multiplication module (the additional hidden width).

**Multiplication Module** Before diving into approximating terms $(A)$ and $(B)$, we use a multiplication module consisting of a series of transformer blocks to transform each column vector $\mathbf{y}_i$ into

$$\mathbf{y}_i = \left[\mathbf{x}_i^\top, \mathbf{e}_i^\top, \boldsymbol{\phi}^\top(t), \mathbf{s}_i^\top, f_{\mathrm{mult}}(\alpha^2, \mathbf{s}_i^\top), f_{\mathrm{mult}}(\sigma^2, \mathbf{s}_i^\top), \mathbf{0}_d^\top, f_{\mathrm{mult}}(\alpha_t, \boldsymbol{\mu}_i)^\top, 1, \mathbf{0}_{3d}^\top\right]^\top,$$

where $f_{\mathrm{mult}}$ denotes an approximation to the entrywise multiplication realized by the multiplication module. We defer the detailed construction of the multiplication module to Appendix G.3, which utilizes $\mathcal{O}(\log(\|\mathbf{s}\|_\infty / \epsilon_{\mathrm{mult}}))$ transformer blocks to reach the accuracy $\left\|f_{\mathrm{mult}}(\alpha^2, \mathbf{s}) - \alpha^2 \mathbf{s}\right\|_\infty \le \epsilon_{\mathrm{mult}}$.

Compared to the raw input before the multiplication module, we have an easy access to the useful quantities $\alpha^2 \mathbf{s}_i$, $\sigma^2 \mathbf{s}_i$ and $\alpha \boldsymbol{\mu}_i$, which simplifies our next step.

**The First Attention Block for Approximating** $(A)$ Here we will construct a transformer block $\mathcal{TB}_1 = \mathrm{FFN}_1 \circ \mathrm{Attn}_1$ for approximating $(A)$. Despite the huge dimension of matrix $\bar{\boldsymbol{\Gamma}}$, it is Toeplitz due to the uniform time grid $\{h_1, \ldots, h_N\}$. There are at most $N$ different entries in $\bar{\boldsymbol{\Gamma}}$ and the $(i, j)$-th entry only depends on the gap $|i - j|$. As a result, we denote $\gamma_m = \bar{\boldsymbol{\Gamma}}_{ij}\mathbb{1}\{|i - j| = m\}$ and rewrite term $(A)$ as

$$(A) = \eta \sum_{m=0}^{N-1} \sum_{j=1}^N \alpha^2 \gamma_m \mathbb{1}\{|i - j| = m\}\boldsymbol{\Sigma}\mathbf{s}_j.$$

The display above suggests a construction of a multi-head attention layer. Formally, for an arbitrary value of $m$, we construct four attention heads with ReLU activation. By Assumption 1, the indicator function $\mathbb{1}\{|i-j|=m\}$ can be realized by calculating the inner product $\mathbf{e}_i^\top \mathbf{e}_j$ of time embedding. To see this, we observe

$$\mathbf{e}_i^\top \mathbf{e}_j = \frac{1}{2}(2r^2 - \|\mathbf{e}_i - \mathbf{e}_j\|_2^2) = \frac{1}{2}(2r^2 - f^2(|i-j|)).$$

Therefore, it holds that

$$\mathbb{1}\{|i-j|=m\} = \mathbb{1}\left\{\mathbf{e}_i^\top \mathbf{e}_j = r^2 - \frac{1}{2}f^2(m)\right\},$$

since $f$ is strictly increasing. Directly approximating an indicator function using ReLU network can be difficult. Yet we note that $|i-j|$ can only take integer values. Therefore, we can slightly widen the decision band for the indicator function. Specifically, we denote a minimum gap $\Delta = \min_{i=1,\dots,N-1}\{f^2(i+1) - f^2(i)\}$. Thus, we deduce

$$\mathbb{1}\{|i-j|=m\} = \mathbb{1}\left\{\mathbf{e}_i^\top \mathbf{e}_j \in \left[r^2 - \frac{1}{2}f^2(m) - \frac{1}{4}\Delta, r^2 - \frac{1}{2}f^2(m) + \frac{1}{4}\Delta\right]\right\}.$$

We can use four ReLU functions to approximate the right-hand side of the last display. In specific, we construct a trapezoid function as follows,

$$\psi(\mathbf{e}_i^\top \mathbf{e}_j) = \frac{8}{\Delta}\mathrm{ReLU}\left(\mathbf{e}_i^\top \mathbf{e}_j - r^2 + \frac{1}{2}f^2(m) + \frac{1}{4}\Delta\right) - \frac{8}{\Delta}\mathrm{ReLU}\left(\mathbf{e}_i^\top \mathbf{e}_j - r^2 + \frac{1}{2}f^2(m) + \frac{1}{8}\Delta\right)$$

$$- \frac{8}{\Delta}\mathrm{ReLU}\left(\mathbf{e}_i^\top \mathbf{e}_j - r^2 + \frac{1}{2}f^2(m) - \frac{1}{8}\Delta\right) + \frac{8}{\Delta}\mathrm{ReLU}\left(\mathbf{e}_i^\top \mathbf{e}_j - r^2 + \frac{1}{2}f^2(m) - \frac{1}{4}\Delta\right).$$

It is straightforward to check that $\psi = 1$ in a $\frac{4}{\Delta}$-width interval centered at $r^2 - \frac{1}{2}f^2(m)$. To this end, we can use four attention heads to realize the function $\psi$. In particular, for the first attention head, we choose

$$(\mathbf{Q}^1)^\top \mathbf{K}^1 = \mathrm{diag}\left(\left[\mathbf{0}_{d\times d}, \mathbf{I}_{d_e}, \mathbf{0}_{d_t\times d_t}, \mathbf{0}_{(5d)\times(5d)}, -r^2 + \frac{1}{2}f^2(m) + \frac{1}{4}\Delta, \mathbf{0}_{(3d)\times(3d)}\right]\right) \text{ and}$$

$$\mathbf{V}^1 = \begin{bmatrix} \mathbf{0}_{(4d+d_e+d_t)\times(2d+d_e+d_t)} & \mathbf{0}_{(4d+d_e+d_t)\times d} & \mathbf{0}_{(4d+d_e+d_t)\times(6d+1)} \\ \mathbf{0}_{d\times(2d+d_e+d_t)} & \frac{8}{\Delta}\gamma_m\boldsymbol{\Sigma} & \mathbf{0}_{d\times(6d+1)} \\ \mathbf{0}_{(4d+1)\times(2d+d_e+d_t)} & \mathbf{0}_{(4d+1)\times d} & \mathbf{0}_{(4d+1)\times(6d+1)} \end{bmatrix}.$$

It is not difficult to check that this attention head calculates the first ReLU function in $\psi$. Analogously, for the second attention head, we choose

$$(\mathbf{Q}^2)^\top \mathbf{K}^2 = \mathrm{diag}\left(\left[\mathbf{0}_{d\times d}, \mathbf{I}_{d_e}, \mathbf{0}_{d_t\times d_t}, \mathbf{0}_{(5d)\times(5d)}, -r^2 + \frac{1}{2}f^2(m) + \frac{1}{8}\Delta, \mathbf{0}_{(3d)\times(3d)}\right]\right) \text{ and}$$

$$\mathbf{V}^2 = -\mathbf{V}^1$$

for realizing the second ReLU function. The third and fourth attention heads have the following parameters,

$$(\mathbf{Q}^3)^\top \mathbf{K}^3 = \mathrm{diag}\left(\left[\mathbf{0}_{d\times d}, \mathbf{I}_{d_e}, \mathbf{0}_{d_t\times d_t}, \mathbf{0}_{(5d)\times(5d)}, -r^2 + \frac{1}{2}f^2(m) - \frac{1}{8}\Delta, \mathbf{0}_{(3d)\times(3d)}\right]\right),$$

$$\mathbf{V}^3 = -\mathbf{V}^1$$

$$(\mathbf{Q}^4)^\top \mathbf{K}^4 = \mathrm{diag}\left(\left[\mathbf{0}_{d\times d}, \mathbf{I}_{d_e}, \mathbf{0}_{d_t\times d_t}, \mathbf{0}_{(5d)\times(5d)}, -r^2 + \frac{1}{2}f^2(m) - \frac{1}{4}\Delta, \mathbf{0}_{(3d)\times(3d)}\right]\right), \text{ and}$$

$$\mathbf{V}^4 = \mathbf{V}^1.$$

By summing up the output of the four attention heads, we derive the output of such an attention layer. For the $i$-th patch, the output is

$$\Big[\mathbf{x}_i^\top, \mathbf{e}_i^\top, \boldsymbol{\phi}^\top(t), \mathbf{s}_i, f_{\mathrm{mult}}(\alpha^2, \mathbf{s}_i^\top), f_{\mathrm{mult}}(\sigma^2, \mathbf{s}_i^\top),$$

$$\sum_{j=1}^N \gamma_m \psi(\mathbf{e}_i^\top \mathbf{e}_j)\boldsymbol{\Sigma} \cdot f_{\mathrm{mult}}(\alpha^2, \mathbf{s}_j^\top), f_{\mathrm{mult}}(\alpha, \boldsymbol{\mu}_i)^\top, 1, \mathbf{0}_{3d}^\top\Big]^\top.$$

Note that this output calculates for a fixed value of $m$. In order to summing over different values of $m$, we utilize $4J$ attention heads. Here $4J$ attention heads are enough, since $\bar{\mathbf{\Gamma}}_{ij} = 0$ if $|i - j| \geq J$. We have

$$
\texttt{Attn}_1(\mathbf{y}_i) = \Big[ \mathbf{x}_i^\top, \mathbf{e}_i^\top, \boldsymbol{\phi}^\top(t), \mathbf{s}_i, f_{\mathrm{mult}}(\alpha^2, \mathbf{s}_i^\top), f_{\mathrm{mult}}(\sigma^2, \mathbf{s}_i^\top),
$$
$$
\sum_{m=0}^{N-1} \sum_{j=1}^{N} \gamma_m \psi(\mathbf{e}_i^\top \mathbf{e}_j) \mathbf{\Sigma} \cdot f_{\mathrm{mult}}(\alpha^2, \mathbf{s}_j^\top), f_{\mathrm{mult}}(\alpha, \boldsymbol{\mu}_i)^\top, 1, \mathbf{0}_{3d}^\top \Big]^\top.
$$

For the feedforward layer $\texttt{FFN}_1$, we set $\mathbf{W}_1 = \mathbf{0}$, $\mathbf{W}_2 = \mathbf{0}$, $\mathbf{b}_1 = \mathbf{0}$ and $\mathbf{b}_2 = \mathbf{0}$ such that the output of the first attention block is

$$
\mathcal{TB}_1(\mathbf{y}_i) = \Big[ \mathbf{x}_i^\top, \mathbf{e}_i^\top, \boldsymbol{\phi}^\top(t), \mathbf{s}_i, f_{\mathrm{mult}}(\alpha^2, \mathbf{s}_i^\top), f_{\mathrm{mult}}(\sigma^2, \mathbf{s}_i^\top),
$$
$$
\sum_{m=0}^{N-1} \sum_{j=1}^{N} \gamma_m \psi(\mathbf{e}_i^\top \mathbf{e}_j) \mathbf{\Sigma} \cdot f_{\mathrm{mult}}(\alpha^2, \mathbf{s}_j^\top), f_{\mathrm{mult}}(\alpha, \boldsymbol{\mu}_i)^\top, 1, \mathbf{0}_{3d}^\top \Big]^\top.
$$

**The Second Transformer Block for Approximating** $(B)$ Similar to the first block, the goal here is to construct $\mathcal{TB}_2 = \texttt{FFN}_2 \circ \texttt{Attn}_2$ for realizing $(B)$. This is much easier than approximating $(A)$, we only need the feed forward layer while set the attention layer trivial. In particular, we choose $\mathbf{Q}, \mathbf{K}$ and $\mathbf{V}$ being all zero matrices so as to maintain the input to the attention layer. For the feedforward layer $\texttt{FFN}_2$, we choose $\mathbf{W}_1$ as

$$
\mathbf{W}_1 = \begin{bmatrix}
\mathbf{I}_d & \mathbf{0}_{d_e \times d_e} & \mathbf{0}_{d_t \times d_t} & \mathbf{I}_d & \mathbf{0}_{d \times d} & \mathbf{I}_d & \mathbf{I}_d & -\mathbf{I}_d & \mathbf{0}_{d \times (3d+1)} \\
-\mathbf{I}_d & \mathbf{0}_{d_e \times d_e} & \mathbf{0}_{d_t \times d_t} & -\mathbf{I}_d & \mathbf{0}_{d \times d} & -\mathbf{I}_d & -\mathbf{I}_d & \mathbf{I}_d & \mathbf{0}_{d \times (3d+1)} \\
 & & & -\mathbf{I}_d & \mathbf{I}_d & & & & \\
 & & & \mathbf{I}_d & -\mathbf{I}_d & & & & \\
 & & & & & \mathbf{I}_d & & & \\
 & & & & & -\mathbf{I}_d & & & \\
 & & & & & & \mathbf{I}_d & & \\
 & & & & & & -\mathbf{I}_d & &
\end{bmatrix} \in \mathbb{R}^{(8d) \times D},
$$

where the missing values are all zero. Then we have $\mathbf{W}_1 \cdot \texttt{Attn}_2 \circ \mathcal{TB}_1(\mathbf{y}_i)$ as

$$
\begin{bmatrix}
\mathbf{s}_i + \sum_{m=0}^{N-1} \sum_{j=1}^{N} \gamma_m \psi(\mathbf{e}_i^\top \mathbf{e}_j) \mathbf{\Sigma} \cdot f_{\mathrm{mult}}(\alpha^2, \mathbf{s}_j) + f_{\mathrm{mult}}(\sigma^2, \mathbf{s}_i) + (\mathbf{x}_i - f_{\mathrm{mult}}(\alpha, \boldsymbol{\mu}_i)) \\
-\mathbf{s}_i - \sum_{m=0}^{N-1} \sum_{j=1}^{N} \gamma_m \psi(\mathbf{e}_i^\top \mathbf{e}_j) \mathbf{\Sigma} \cdot f_{\mathrm{mult}}(\alpha^2, \mathbf{s}_j) - f_{\mathrm{mult}}(\sigma_t^2, \mathbf{s}_i) - (\mathbf{x}_i - f_{\mathrm{mult}}(\alpha, \boldsymbol{\mu}_i)) \\
-\mathbf{s}_i + f_{\mathrm{mult}}(\alpha^2, \mathbf{s}_i) \\
\mathbf{s}_i - f_{\mathrm{mult}}(\alpha^2, \mathbf{s}_i) \\
f_{\mathrm{mult}}(\sigma^2, \mathbf{s}_i) \\
-f_{\mathrm{mult}}(\sigma^2, \mathbf{s}_i) \\
\sum_{m=0}^{N-1} \sum_{j=1}^{N} \gamma_m \psi(\mathbf{e}_i^\top \mathbf{e}_j) \mathbf{\Sigma} \cdot f_{\mathrm{mult}}(\alpha^2, \mathbf{s}_j) \\
-\sum_{m=0}^{N-1} \sum_{j=1}^{N} \gamma_m \psi(\mathbf{e}_i^\top \mathbf{e}_j) \mathbf{\Sigma} \cdot f_{\mathrm{mult}}(\alpha^2, \mathbf{s}_j)
\end{bmatrix}.
$$

It suffices to choose $\mathbf{b}_1 = \mathbf{b}_2 = \mathbf{0}$ and $\mathbf{W}_2$ equal to

$$
\mathbf{W}_2 = \begin{bmatrix}
 & & \mathbf{0}_{(d+d_e+d_t) \times (8d)} & \\
-\mathbf{I}_d & \mathbf{I}_d & & & \\
 & & -\mathbf{I}_d & \mathbf{I}_d & & \\
 & & & -\mathbf{I}_d & \mathbf{I}_d & \\
 & & & & -\mathbf{I}_d & \mathbf{I}_d \\
 & & \mathbf{0}_{(4d+1) \times (8d)} & &
\end{bmatrix} \in \mathbb{R}^{D \times (8d)},
$$

where missing values are all zero. Using the fact that $\mathrm{ReLU}(x) - \mathrm{ReLU}(-x) = x$, we have

$$
\mathbf{W}_2 \cdot \mathrm{ReLU}(\mathbf{W}_1 \cdot \texttt{Attn}_2 \circ \mathcal{TB}_1(\mathbf{y}_i))
$$
$$
= \Big[ \mathbf{0}_d^\top, \mathbf{0}_{d_e}^\top, \mathbf{0}_{d_t}^\top, -\mathbf{s}_i^\top - \sum_{m=0}^{N-1} \sum_{j=1}^{N} \gamma_m \psi(\mathbf{e}_i^\top \mathbf{e}_j) \mathbf{\Sigma} \cdot f_{\mathrm{mult}}(\alpha^2, \mathbf{s}_j^\top) - f_{\mathrm{mult}}(\sigma^2, \mathbf{s}_i^\top) - (\mathbf{x}_i^\top - f_{\mathrm{mult}}(\alpha, \boldsymbol{\mu}_i^\top)),
$$
$$
\mathbf{s}_i^\top - \sum_{m=0}^{N-1} \sum_{j=1}^{N} \gamma_m \psi(\mathbf{e}_i^\top \mathbf{e}_j) \mathbf{\Sigma} \cdot f_{\mathrm{mult}}(\alpha^2, \mathbf{s}_j^\top), -f_{\mathrm{mult}}(\alpha^2, \mathbf{s}_i^\top), -f_{\mathrm{mult}}(\sigma^2, \mathbf{s}_i^\top), \mathbf{0}_d^\top, 0, \mathbf{0}_{3d}^\top \Big]^\top.
$$

Therefore, by concatenating the two transformer blocks, we have

$$
\mathcal{TB}_2 \circ \mathcal{TB}_1(\mathbf{y}_i) = \Bigg[ \mathbf{x}_i^\top, \boldsymbol{e}_i^\top, \boldsymbol{\phi}^\top(t),
$$

$$
- \sum_{m=0}^{N-1} \sum_{j=1}^{N} \gamma_m \psi(\mathbf{e}_i^\top \mathbf{e}_j) \boldsymbol{\Sigma} \cdot f_{\text{mult}}(\alpha^2, \mathbf{s}_j^\top) - f_{\text{mult}}(\sigma^2, \mathbf{s}_i^\top) - (\mathbf{x}_i^\top - f_{\text{mult}}(\alpha, \boldsymbol{\mu}_i^\top)),
$$

$$
\mathbf{s}_i^\top, \mathbf{0}_d^\top, \mathbf{0}_d^\top, f_{\text{mult}}(\alpha, \boldsymbol{\mu}_i)^\top, 1, \mathbf{0}_{3d}^\top \Bigg]^\top .
$$

Lastly, by implementing another multiplication module $f_{\text{mult}}$ that scales the gradient by the learning rate $\eta_t$, we obtain

$$
f_{\text{mult}} \circ \mathcal{TB}_2 \circ \mathcal{TB}_1(\mathbf{y}_i) = \Bigg[ \mathbf{x}_i^\top, \boldsymbol{e}_i^\top, \boldsymbol{\phi}^\top(t),
$$

$$
f_{\text{mult}}\Bigg( \eta_t, - \sum_{m=0}^{N-1} \sum_{j=1}^{N} \gamma_m \psi(\mathbf{e}_i^\top \mathbf{e}_j) \boldsymbol{\Sigma} \cdot f_{\text{mult}}(\alpha^2, \mathbf{s}_j^\top) - f_{\text{mult}}(\sigma^2, \mathbf{s}_i^\top) + (\mathbf{x}_i^\top - f_{\text{mult}}(\alpha, \boldsymbol{\mu}_i^\top)) \Bigg),
$$

$$
\mathbf{s}_i^\top, \mathbf{0}_d^\top, \mathbf{0}_d^\top, f_{\text{mult}}(\alpha_t, \boldsymbol{\mu}_i)^\top, 1, \mathbf{0}_{3d}^\top \Bigg]^\top .
$$

It is straightforward to implement a feedforward layer to sum up $\mathbf{s}_i$ with the gradient increment in $f_{\text{mult}} \circ \mathcal{TB}_2 \circ \mathcal{TB}_1(\mathbf{y}_i)$. Clearly, the feedforward layer can be realized by a transformer block with a trivial attention layer — similar to $\mathcal{TB}_2$ with much simpler weight matrices. As a result, we have implemented one gradient descent iteration, where the output replaces $\mathbf{s}_i$ in the initial input vector. We denote $f_{\text{GD},2}$ as our network implementation of one GD iteration.

We can repeat the multiplication module followed by the two transformer blocks to approximate the gradient descent for $K$ iterations as needed. Note that, we don't need to calculate $f_{\text{mult}}(\alpha_t, \boldsymbol{\mu}_i)^\top$ after the first step of GD, which leads to the construction of $f_{\text{GD},2}$.

**Bounding Approximation Error**  Examining the output of our approximation for one gradient descent iteration, it is not difficult to observe that the approximation error is determined by the error in the multiplication module. For the $k$-th iteration, we denote the approximation realized by the transformer as

$$
\widehat{\mathbf{s}}_i^{(k+1)} = \widehat{\mathbf{s}}_i^{(k)} + f_{\text{mult}}\Bigg( \eta_t, - \sum_{m=0}^{N-1} \sum_{j=1}^{N} \gamma_m \psi(\mathbf{e}_i^\top \mathbf{e}_j) \boldsymbol{\Sigma} \cdot f_{\text{mult}}(\alpha^2, \widehat{\mathbf{s}}_j^{(k)})
$$

$$
- f_{\text{mult}}(\sigma^2, \widehat{\mathbf{s}}_i^{(k)}) - (\mathbf{x}_i^\top - f_{\text{mult}}(\alpha, \boldsymbol{\mu}_i)) \Bigg).
$$

We compare the output with the exact GD update with last iteration at $\widehat{\mathbf{s}}_i^{(k)}$, which is

$$
\widehat{\mathbf{s}}_{i,\star}^{(k+1)} = \widehat{\mathbf{s}}_i^{(k)} - \eta_t \sum_{m=0}^{N-1} \sum_{j=1}^{N} \gamma_m \psi(\mathbf{e}_i^\top \mathbf{e}_j) \boldsymbol{\Sigma} \cdot \alpha^2 \widehat{\mathbf{s}}_j^{(k)} - \sigma^2 \widehat{\mathbf{s}}_i^{(k)} - (\mathbf{x}_i^\top - \alpha \boldsymbol{\mu}_i).
$$

For any index $i \in \{1, \ldots, N\}$, we bound the difference

$$
\left\| \widehat{\mathbf{s}}_{i,\star}^{(k+1)} - \widehat{\mathbf{s}}_i^{(k+1)} \right\|_2 \overset{(i)}{\leq} \epsilon_{\text{mult}} + \eta_t \sum_{m=0}^{N-1} \sum_{j=1}^{N} \left\| \boldsymbol{\Sigma}(\alpha^2 \mathbf{s}_j^{(k)} - f_{\text{mult}}(\alpha^2, \widehat{\mathbf{s}}_j^{(k)})) \right\|_2
$$

$$
+ \eta_t \left\| \sigma^2 \mathbf{s}_i^{(k)} - f_{\text{mult}}(\sigma^2, \widehat{\mathbf{s}}_i^{(k)}) \right\|_2 + \eta_t \left\| \alpha \boldsymbol{\mu}_i - f_{\text{mult}}(\alpha, \boldsymbol{\mu}_i) \right\|_2
$$

$$
\overset{(ii)}{\leq} \epsilon_{\text{mult}} + \eta_t \left\| \boldsymbol{\Sigma} \right\|_2 N \sqrt{d} \epsilon_{\text{mult}} + 2\eta_t \sqrt{d} \epsilon_{\text{mult}},
$$

where in inequality $(i)$, we use the fact that $\gamma_m \psi(\mathbf{e}_i^\top \mathbf{e}_j) \leq 1$, and in inequality $(ii)$, we plug in the approximation error of multiplication module. By setting $\epsilon_{\text{mult}} = \frac{1}{6} \eta_t^{-1} \left\| \boldsymbol{\Sigma} \right\|_F^{-1} N^{-3/2} d^{-1/2} \epsilon$, we ensure that $\left\| \widehat{\mathbf{s}}_{i,\star}^{(k+1)} - \widehat{\mathbf{s}}_i^{(k+1)} \right\|_2 \leq \epsilon/\sqrt{N}$.

**Size of Transformer Blocks for Approximating One GD Iteration**   Similar to the proof of Lemma 9, we summarize the network size for implementing one GD iteration (except the first iteration).

Table 2: Transformer size for approximating one GD iteration (except the first iteration)

| Input dimension | $D \times N$ with $D = 9d + d_e + d_t + 1$ |
| --- | --- |
| # of blocks $L$ | $2 + L_{\mathrm{mult}}$ with $L_{\mathrm{mult}} = \mathcal{O}(\log(\|\mathbf{x}\|_\infty \|\mathbf{s}\|_\infty Nd/\epsilon))$ |
| # of attention heads $M$ | $4J$ with $J$ the covariance truncation length |
| Parameter bound $B$ | $\mathcal{O}\big(d(\|\mathbf{\Sigma}\|_\infty + r^2)\big) + \|\mathbf{x}\|_\infty + \|\mathbf{s}\|_\infty$ |

To see the parameter bound $B$, we observe that the parameters in the constructed transformer are bounded by $\max\{1, \frac{8}{\Delta}\eta \|\mathbf{\Sigma}\|_\infty, r^2\}$. Here, $\|\mathbf{\Sigma}\|_\infty = \max_{ij}|\Sigma_{ij}|$ denotes the maximum magnitude of entries. Since there are at most $\mathcal{O}(d)$ nonzero weights in each weight matrix, we have the norm of the parameters bounded by

$$\mathcal{O}\big(d(\Delta^{-1}\|\mathbf{\Sigma}\|_\infty + r^2)\big) + \|\mathbf{x}\|_\infty + \|\mathbf{s}\|_\infty = \mathcal{O}\big(d(\|\mathbf{\Sigma}\|_\infty + r^2)\big) + \|\mathbf{x}\|_\infty + \|\mathbf{s}\|_\infty.$$

The proof is complete. $\qquad\square$

### D.3.3   FORMAL PROOF OF LEMMA 6

Recall that the score function is written as

$$\mathbf{s}(\mathbf{v}_t) = -(\alpha_t^2(\mathbf{\Gamma} \otimes \mathbf{\Sigma}) + \sigma_t^2\mathbf{I})^{-1}(\mathbf{v}_t - \alpha_t\boldsymbol{\mu}).$$

Thus, if $\|\mathbf{s}(\mathbf{v}_t)\|_2 \le R_0\sigma_t^{-1}$, we have $\|\mathbf{v}_t - \alpha_t\boldsymbol{\mu}\|_2 \le \big\|\alpha_t^2(\mathbf{\Gamma} \otimes \mathbf{\Sigma}) + \sigma_t^2\mathbf{I}\big\|_2 \|\mathbf{s}(\mathbf{v}_t)\|_2 \le CR_0\sigma_t^{-1}$, where

$$
\begin{aligned}
C &= \big\|(\alpha_t^2(\mathbf{\Gamma} \otimes \mathbf{\Sigma}) + \sigma_t^2\mathbf{I})\big\|_2 \\
&\le \big\|\alpha_t^2(\mathbf{\Gamma} \otimes \mathbf{\Sigma})\big\|_2 + \big\|\sigma_t^2\mathbf{I}\big\|_2 \\
&\le \|\mathbf{\Gamma}\|_{\mathrm{F}} \|\mathbf{\Sigma}\|_2 + 1 \\
&\le \sqrt{N(\ell+1)} \|\mathbf{\Sigma}\|_2 + 1.
\end{aligned}
$$

The last inequality follows from Lemma 4. Thus, the infinity norm of $\mathbf{v}_t$ can be bounded by $\|\mathbf{v}_t\|_\infty \le \|\mathbf{v}_t - \alpha_t\boldsymbol{\mu}\|_\infty + \|\alpha_t\boldsymbol{\mu}\|_\infty \le CR_0\sigma_t^{-1} + \mu_0$. Moreover, by the proof of lemma 5, we can also bound the norm of the truncated score function by

$$
\begin{aligned}
\|\bar{\mathbf{s}}(\mathbf{v}_t)\| &\le \|\bar{\mathbf{s}}(\mathbf{v}_t) - \mathbf{s}(\mathbf{v}_t)\|_2 + \|\mathbf{s}(\mathbf{v}_t)\|_2 \\
&\le \sigma_t^{-4}\|\Delta\mathbf{\Gamma}\|_{\mathrm{F}} \|\mathbf{v}_t - \alpha_t\boldsymbol{\mu}\|_2 + \|\mathbf{s}(\mathbf{v}_t)\|_2 \\
&\le \sigma_t^{-4}\|\Delta\mathbf{\Gamma}\|_{\mathrm{F}}(CR_0\sigma_t^{-1} + \mu_0) + R_0\sigma_t^{-1}.
\end{aligned}
$$

Thus, according to Lemma 4, by choosing $J = \mathcal{O}((\ell\log(N\ell\|\mathbf{\Sigma}\|_{\mathrm{F}}/\sigma_t))^{1/\nu})$, we can ensure that $\|\bar{\mathbf{s}}(\mathbf{v}_t)\| \le 2R_0\sigma_t^{-1}$.

Now we construct the transformers as follows:

$$\widetilde{\mathbf{s}}(\mathbf{x}) = f_{\mathrm{norm}} \circ f_{\mathrm{linear}} \circ \underbrace{f_{\mathrm{GD},2} \circ f_{\mathrm{GD},2} \circ \cdots \circ f_{\mathrm{GD},2}}_{(K-1)\times f_{\mathrm{GD},2}} \circ f_{\mathrm{GD},1} \circ f_\mu \circ f_{\mathrm{in}}.$$

Here $f_{\mathrm{out}}$ extracts the $(d + d_e + d_t + 1)$-th to $(2d + d_e + d_t)$-th rows of the output as the score approximator, and we choose the clipping range as $R = 2R_0\sigma_t^{-1}$.

Moreover, let

$$\widetilde{\mathbf{s}}_0 = f_{\mathrm{linear}} \circ f_{\mathrm{GD},2} \circ f_{\mathrm{GD},2} \circ \cdots \circ f_{\mathrm{GD},2} \circ f_{\mathrm{GD},1} \circ f_\mu \circ f_{\mathrm{in}}$$

to be the score approximator without clipping, then under the condition that $\|\mathbf{s}(\mathbf{v}_t)\|_2 \le R_0\sigma_t^{-1}$, we have $\|\widetilde{\mathbf{s}}(\mathbf{v}_t) - \bar{\mathbf{s}}(\mathbf{v}_t)\|_2 \le \|\widetilde{\mathbf{s}}(\mathbf{v}_t)_0 - \bar{\mathbf{s}}(\mathbf{v}_t)\|_2$.

Now let's bound the difference between $\widetilde{\mathbf{s}}_0(\mathbf{v}_t)$ and $\bar{\mathbf{s}}$. By Lemma 9 and Lemma 10, $\widetilde{\mathbf{s}}_0(\mathbf{v}_t)$ expresses the output after $K$ steps of GD with error level $\sqrt{N} \cdot \epsilon/\sqrt{N} = \epsilon$. Here we multiply $\sqrt{N}$ because we

are considering the entire score function instead of each patch. Then by Lemma 2 and following the proof of Lemma 3, we have

$$\|\widetilde{\mathbf{s}}_0(\mathbf{v}_t) - \bar{\mathbf{s}}(\mathbf{v}_t)\|_2 \leq \frac{(\kappa_t + 1)\epsilon}{2} + \exp\left(-\frac{2K}{\kappa_t + 1}\right) \|\bar{\mathbf{s}}(\mathbf{v}_t)\|_2$$

$$= \frac{(\kappa_t + 1)\epsilon}{2} + \exp\left(-\frac{2K}{\kappa_t + 1}\right) 2R_0\sigma_t^{-1}.$$

Thus, by replacing the error level $\epsilon$ by $2\epsilon/(C\sigma_{t_0}^{-2} + 1) \leq 2\epsilon/(\kappa_t + 1)$ for sufficiently large constant $C$ and $K = \lceil \frac{\kappa_{t_0}+1}{2} \log(4R_0/(\sigma_t\epsilon))\rceil$, we have $\|\widetilde{\mathbf{s}}_0(\mathbf{v}_t) - \bar{\mathbf{s}}(\mathbf{v}_t)\|_2 \leq \epsilon$ for any $t \geq t_0$. Moreover, we remark that each patch of the score approximator lies in the euclidean ball with radius

$$\left\|\widetilde{\mathbf{s}}_i^{(k)}(\mathbf{v}_t)\right\|_2 \leq \epsilon + 2\|\bar{\mathbf{s}}(\mathbf{v}_t)\|_2 \leq 1 + 4R_0\sigma_t^{-1}$$

throughout the network. Here $\widetilde{\mathbf{s}}_i^{(k)}(\mathbf{v}_t)$ represents the $i$-th patch of the score approximator after $k$ GD blocks.

**The Overall Size of Transformer Architecture for Approximating The Score Function** We combine the network sizes in Table 1 and Table 2, which gives rise to a characterization of the overall network size for approximating the score function of Gaussian process data. The result is summarized in the following table.

Table 3: Overall transformer network size for approximating the score function

| Input dimension | $D \times N$ with $D = 9d + d_e + d_t + 1$ |
|---|---|
| # of blocks $L$ | $L_\mu + (2 + L_{\text{mult}})K = \mathcal{O}(\kappa_{t_0} \log(R_0Nd/\epsilon)^2)$ |
| # of attention heads $M$ | $4J = \mathcal{O}\left((\ell \log(N\ell\|\mathbf{\Sigma}\|_F/\sigma_{t_0}))^{1/v}\right)$ |
| Parameter bound $B$ | $\mathcal{O}\left(\log(R_0Nd/(\epsilon\sigma_{t_0}))\sqrt{N}R_0\sigma_{t_0}^{-2}(r^2 + \|\mathbf{\Sigma}\|_\infty)\right)$ |
| Output range $R_t$ | $2R_0\sigma_t^{-1}$ |

Note that to obtain the bound on $L$ and $M$, we substitute the choice of $J$ and $K$ into the network sizes in Tables 1 and 2. The parameter bound is obtained by explicitly evaluating the magnitude $\|\mathbf{s}\|_\infty$ of the score function in Table 2. The proof is complete.

### D.4 OMITTED PROOFS OF LEMMAS ABOUT THE SCORE FUNCTION

*Proof of Lemma 5.* Suppose $\mathbf{v}_t \sim P_t = \mathcal{N}(\alpha_t\boldsymbol{\mu}, \alpha_t^2(\mathbf{\Gamma}\otimes\mathbf{\Sigma}) + \sigma_t^2\mathbf{I}))$. Denote $\mathbf{\Phi}_t = \alpha_t^2(\mathbf{\Gamma}\otimes\mathbf{\Sigma}) + \sigma_t^2\mathbf{I}$ and $\bar{\mathbf{\Phi}}_t = \alpha_t^2(\bar{\mathbf{\Gamma}}\otimes\mathbf{\Sigma}) + \sigma_t^2\mathbf{I}$. Then we have $\mathbf{\Gamma}_t = \bar{\mathbf{\Phi}}_t + \alpha_t^2(\Delta\mathbf{\Gamma}\otimes\mathbf{\Sigma})$. Moreover, by Lemma 4, choosing $J = \mathcal{O}((\log(N\|\mathbf{\Sigma}\|_F/(\epsilon\sigma_t)))$ ensures that $\|\Delta\mathbf{\Gamma}\|_F \leq \epsilon\|\mathbf{\Sigma}\|_F^{-2}\sigma_t^2$. Thus, we have

$$\|\bar{\mathbf{s}} - \nabla\log p_t\|_{L_2(P_t)}^2 = \mathbb{E}_{\mathbf{v}_t}\left[\|\bar{\mathbf{s}}(\mathbf{v}_t) - \nabla\log p_t(\mathbf{v}_t)\|_2^2\right]$$

$$= \mathbb{E}_{\mathbf{v}_t}\left[\left\|(\bar{\mathbf{\Phi}}_t^{-1} - \mathbf{\Gamma}_t^{-1})(\mathbf{v}_t - \alpha_t\boldsymbol{\mu})\right\|_2^2\right]$$

$$= \left\|(\bar{\mathbf{\Phi}}_t^{-1} - \mathbf{\Gamma}_t^{-1})\mathbf{\Gamma}_t^{1/2}\right\|_F^2$$

$$= \text{tr}\left((\bar{\mathbf{\Phi}}_t^{-1} - \mathbf{\Gamma}_t^{-1})\mathbf{\Gamma}_t(\bar{\mathbf{\Phi}}_t^{-1} - \mathbf{\Gamma}_t^{-1})\right)$$

$$\leq \left\|(\bar{\mathbf{\Phi}}_t^{-1} - \mathbf{\Gamma}_t^{-1})\mathbf{\Gamma}_t\right\|_F \left\|\bar{\mathbf{\Phi}}_t^{-1} - \mathbf{\Gamma}_t^{-1}\right\|_F$$

$$\leq \left\|(\bar{\mathbf{\Phi}}_t^{-1} - \mathbf{\Gamma}_t^{-1})\mathbf{\Gamma}_t\right\|_F \left\|(\bar{\mathbf{\Phi}}_t^{-1} - \mathbf{\Gamma}_t^{-1})\mathbf{\Gamma}_t\right\|_F \left\|\mathbf{\Gamma}_t^{-1}\right\|_2$$

$$= \left\|\alpha_t^2\bar{\mathbf{\Phi}}_t^{-1}(\Delta\mathbf{\Gamma}\otimes\mathbf{\Sigma})\right\|_F^2 \left\|(\alpha_t^2(\mathbf{\Gamma}\otimes\mathbf{\Sigma}) + \sigma_t^2\mathbf{I})^{-1}\right\|_2$$

$$\leq \left\|(\alpha_t^2(\bar{\mathbf{\Gamma}}\otimes\mathbf{\Sigma}) + \sigma_t^2\mathbf{I})^{-1}\right\|_2^2 \left\|\alpha_t^2(\Delta\mathbf{\Gamma}\otimes\mathbf{\Sigma})\right\|_F^2 \left\|(\alpha_t^2(\mathbf{\Gamma}\otimes\mathbf{\Sigma}) + \sigma_t^2\mathbf{I})^{-1}\right\|_2$$

$$\leq \alpha_t^4\|\Delta\mathbf{\Gamma}\|_F^2\|\mathbf{\Sigma}\|_F^2\left\|\sigma_t^2\mathbf{I}\right\|_2^{-3}$$

$$\leq \frac{\epsilon^2}{\sigma_t^2}.$$

In the last inequality, we invoke $\|\Delta\mathbf{\Gamma}\|_{\mathrm{F}} \leq \epsilon \|\mathbf{\Sigma}\|_{\mathrm{F}}^{-2} \sigma_t^2$. The proof is complete. $\qquad\square$

*Proof of Lemma 7.* Suppose $\mathbf{v}_t \sim P_t = \mathcal{N}(\alpha_t\boldsymbol{\mu}, \alpha_t^2(\mathbf{\Gamma}\otimes\mathbf{\Sigma})+\sigma_t^2\mathbf{I})$. Denote $\mathbf{\Phi}_t = \alpha_t^2(\mathbf{\Gamma}\otimes\mathbf{\Sigma})+\sigma_t^2\mathbf{I}$ and $\bar{\mathbf{\Phi}}_t = \alpha_t^2(\bar{\mathbf{\Gamma}}\otimes\mathbf{\Sigma})+\sigma_t^2\mathbf{I}$. Then we have

$$
\begin{aligned}
\|\bar{\mathbf{s}}\|_{L_2(P_t)}^2 &= \mathbb{E}_{\mathbf{v}_t}\left[\|\bar{\mathbf{s}}(\mathbf{v}_t) - \nabla\log p_t(\mathbf{v}_t)\|_2^2\right] \\
&= \mathbb{E}_{\mathbf{v}_t}\left[\left\|\bar{\mathbf{\Phi}}_t^{-1}\right\|_2^2\right] \\
&= \left\|\bar{\mathbf{\Phi}}_t^{-1}\mathbf{\Gamma}_t^{1/2}\right\|_{\mathrm{F}}^2 \\
&= \operatorname{tr}\left(\bar{\mathbf{\Phi}}_t^{-1}\mathbf{\Gamma}_t\bar{\mathbf{\Phi}}_t^{-1}\right) \\
&\leq \left\|\bar{\mathbf{\Phi}}_t^{-1}\mathbf{\Gamma}_t\right\|_{\mathrm{F}}\left\|\bar{\mathbf{\Phi}}_t^{-1}\right\|_{\mathrm{F}} \\
&\leq \left\|\bar{\mathbf{\Phi}}_t^{-1}\alpha_t^2(\Delta\mathbf{\Gamma}\otimes\mathbf{\Sigma})+\mathbf{I}\right\|_{\mathrm{F}}\left\|\bar{\mathbf{\Phi}}_t^{-1}\right\|_{\mathrm{F}} \\
&\leq \left(\left\|\bar{\mathbf{\Phi}}_t^{-1}\right\|_2\left\|\alpha_t^2(\Delta\mathbf{\Gamma}\otimes\mathbf{\Sigma})\right\|_{\mathrm{F}}+\sqrt{Nd}\right)\sqrt{Nd}\left\|\bar{\mathbf{\Phi}}_t^{-1}\right\|_2 \\
&\leq \left(\frac{\alpha_t^2\|\Delta\mathbf{\Gamma}\|_{\mathrm{F}}\|\mathbf{\Sigma}\|_{\mathrm{F}}}{\sigma_t^2}+\sqrt{Nd}\right)\frac{\sqrt{Nd}}{\sigma_t^2} \\
&\leq \left(\epsilon+\sqrt{Nd}\right)\frac{\sqrt{Nd}}{\sigma_t^2} \\
&\leq \frac{2Nd}{\sigma_t^2}.
\end{aligned}
$$

Here we invoke $\|\Delta\mathbf{\Gamma}\|_{\mathrm{F}} \leq \|\mathbf{\Sigma}\|_{\mathrm{F}}^{-1}\sigma_t^2$ by Lemma 4 in the second last inequality. $\qquad\square$

*Proof of Lemma 8.* Note that if $\mathbf{v}_t \sim P_t$ we can write $\mathbf{v}_t = \alpha_t\boldsymbol{\mu}-\mathbf{\Phi}_t^{1/2}\mathbf{z}$, where $\mathbf{\Phi}_t = \alpha_t^2\mathbf{\Gamma}\otimes\mathbf{\Sigma}+\sigma_t^2\mathbf{I}$ and $\mathbf{z} \sim \mathcal{N}(\mathbf{0},\mathbf{I})$. Thus, the score function can be written as

$$
\nabla\log p_t(\mathbf{v}_t) = -\mathbf{\Phi}_t^{-1}(\mathbf{v}_t - \alpha_t\boldsymbol{\mu}) = \mathbf{\Phi}_t^{-1/2}\mathbf{z}.
$$

Thus, we have $\|\nabla\log p_t(\mathbf{v}_t)\|_2^2 = \mathbf{z}^\top\mathbf{\Phi}_t^{-1}\mathbf{z}$, which is a quadratic form of the standard Gaussian. By taking $g(\mathbf{z}) = \|\nabla\log p_t(\mathbf{v}_t)\|_2^2$ in Lemma 24, we have

$$
\mathbb{P}\left[\left|\|\nabla\log p_t(\mathbf{v}_t)\|_2^2 - \mathbb{E}[\|\nabla\log p_t(\mathbf{v}_t)\|_2^2]\right| \geq \delta\sqrt{\operatorname{Var}(\|\nabla\log p_t(\mathbf{v}_t)\|_2^2)}\right] \leq 2\exp\left(-C_2\delta\right).
$$

Since we have

$$
\mathbb{E}[\|\nabla\log p_t(\mathbf{v}_t)\|_2^2] = \operatorname{tr}\left(\mathbf{\Phi}_t^{-1}\right) \leq Nd\sigma_t^{-2} \quad\text{and}\quad \operatorname{Var}(\|\nabla\log p_t(\mathbf{v}_t)\|_2^2) \leq \left\|\mathbf{\Phi}_t^{-1}\right\|_{\mathrm{F}}^2 \leq Nd\sigma_t^{-4},
$$

we have $\|\nabla\log p_t(\mathbf{v}_t)\|_2^2 \leq \sigma_t^{-2}\left(N+\delta\sqrt{Nd}\right)$ with probability $1-2\exp\left(-C_2\delta\right)$. $\qquad\square$

### D.5 RESULTS ON TRANSFORMERS WITH SOFTMAX ACTIVATION

In this section, we showcase the capability of softmax transformers in unrolling gradient descent algorithm. In this section, we consider the quadratic kernel function, i.e.,

$$
\gamma(h_i, h_j) = \exp\left(-\|\mathbf{e}_i - \mathbf{e}_j\|_2^2/\ell\right).
$$

In the following lemma, we reproduce our results in Lemma 10 with softmax activation. Moreover, due to the exponential scaling strategy of softmax, we show that one head of attention layer is enough to express the GD with the target function without truncation on $\mathbf{\Gamma}$.

**Lemma 11** (Unroll GD in softmax transformers)**.** Suppose the input token is

$$
\mathbf{y}_i = [\mathbf{x}_i^\top, \mathbf{e}_i^\top, \boldsymbol{\phi}^\top(t), \mathbf{s}_i^\top, \mathbf{0}_d^\top, \mathbf{0}_d^\top, \mathbf{0}_d^\top, f_{\mathrm{mult}}(\alpha_t, \boldsymbol{\mu}_i)^\top, 1, \mathbf{0}_{3d-1}^\top, i]^\top,
$$

then there exists a softmax transformer $f_{\text{GD,softmax}} \in \mathcal{T}_{\text{raw}}(D, \mathcal{O}(\log(\|\mathbf{x}\|_\infty \|\mathbf{s}\|_\infty Nd/\epsilon)),$
$1, \mathcal{O}(N^2 + d + \|\boldsymbol{\Sigma}\|_F + r^2); \text{softmax})$ that approximately iterates $\mathbf{s}_i$ by following the GD update
formula, i.e.,

$$f_{\text{GD,softmax}}(\mathbf{y}_i) = \left[\mathbf{x}_i^\top, \boldsymbol{e}_i^\top, \boldsymbol{\phi}^\top(t), \quad \mathbf{s}_i^\top - \eta_t \left(\sum_{j=1}^N (\alpha_t^2 \boldsymbol{\Gamma}_{ij} \boldsymbol{\Sigma} + \sigma_t^2 \mathbf{I})\mathbf{s}_j^\top - (\mathbf{x}_i - \alpha_t \boldsymbol{\mu}_i)^\top\right) + \boldsymbol{\epsilon}_i, \right.$$

$$\left. \mathbf{0}_d^\top, \mathbf{0}_d^\top, \mathbf{0}_d^\top, f_{\text{mult}}(\alpha_t, \boldsymbol{\mu}_i)^\top, 1, \mathbf{0}_{3d-1}^\top, i\right]^\top.$$

Here $\|\boldsymbol{\epsilon}_i'\|_2 \leq \epsilon/\sqrt{N}$ for $i \in [N]$. Compared with Lemma 10, we add an additional time embedding
$i$ at the end of the input for technical convenience.

*Proof of Lemma 11.* The proof is similar to that of Lemma 10 but with different construction on the
attention layer. We also decompose the gradient descent iteration as

$$\mathbf{s}_i^{(k+1)} = \mathbf{s}_i^{(k)} - \eta_t \left[\sum_{j=1}^N (\alpha_t^2 \boldsymbol{\Gamma}_{ij}\boldsymbol{\Sigma})\mathbf{s}_j^{(k)} + \sigma_t^2 \mathbf{s}_i^{(k)} + (\mathbf{x}_{i,t} - \alpha_t \boldsymbol{\mu}_i)\right]$$

$$= \mathbf{s}_i^{(k)} - \underbrace{\eta_t \sum_{j=1}^N \alpha_t^2 \boldsymbol{\Gamma}_{ij}\boldsymbol{\Sigma}\mathbf{s}_j^{(k)}}_{(A)} - \underbrace{\left(\eta_t \sigma_t^2 \mathbf{s}_i^{(k)} + \eta_t(\mathbf{x}_{i,t} - \alpha_t \boldsymbol{\mu}_i)\right)}_{(B)}. \tag{14}$$

By the proof of Lemma 10, part $(B)$ only utilizes feed-forward networks, thus no changes need
to be made. We will elaborate on approximating $(A)$ using softmax attention layers. To ease the
presentation, we consider a fixed time $t$ and drop the subscript $t$. We also drop the superscript $(k)$.
Recall each token $\mathbf{y}_i$ is

$$\mathbf{y}_i = [\mathbf{x}_i^\top, \boldsymbol{e}_i^\top, \boldsymbol{\phi}^\top(t), \mathbf{s}_i^\top, \mathbf{0}_d^\top, \mathbf{0}_d^\top, \mathbf{0}_d^\top, f_{\text{mult}}(\alpha_t, \boldsymbol{\mu}_i)^\top, 1, \mathbf{0}_{3d-1}^\top, i]^\top \in \mathbb{R}^{9d+d_e+d_t+1}.$$

Here we reserve $\mathbf{0}_{3d}^\top$ as the buffer space for the multiplication module (the additional hidden width).
Before diving into approximating terms $(A)$, we follow the proof of Lemma 10 to use a multiplication
module consisting of a series of transformer blocks to transform each column vector $\mathbf{y}_i$ into

$$\mathbf{y}_i = \left[\mathbf{x}_i^\top, \boldsymbol{e}_i^\top, \boldsymbol{\phi}^\top(t), \mathbf{s}_i^\top, f_{\text{mult}}(\alpha^2, \mathbf{s}_i^\top), f_{\text{mult}}(\sigma^2, \mathbf{s}_i^\top), \mathbf{0}_d^\top, f_{\text{mult}}(\alpha_t, \boldsymbol{\mu}_i)^\top, 1, \mathbf{0}_{3d-1}^\top, i\right]^\top,$$

where $f_{\text{mult}}$ denotes an approximation to the entrywise multiplication realized by the multiplication
module. We defer the detailed construction of the multiplication module to Appendix G.3, which
utilizes $\mathcal{O}(\log(1/\epsilon_{\text{mult}}))$ transformer blocks to reach the accuracy $\left\|f_{\text{mult}}(\alpha^2, \mathbf{s}) - \alpha^2 \mathbf{s}\right\|_\infty \leq \epsilon_{\text{mult}}$.
Here we could still utilize the multiplication module after changing the activation function in attention
layer to softmax because its construction only relies on feed-forward networks, so we can simply set
the attention layers trivial throughout the network in multiplication module. See Appendix G.3 for
more discussions. Now we begin to approximate $(A)$.

**Comparison with ReLU activation**  In softmax transformers, we leverage the exponential scaling
mechanism of the softmax activation to directly construct the entire kernel matrix $\Gamma$ without truncation,
so only one head of attention layer is needed. On the contrary, in ReLU transformers, we only
construct the main diagonals of $\boldsymbol{\Gamma}$ with multiple attention heads. See Lemma 10 for more details.

**Approximate $(A)$**  Here we will construct a transformer block $\mathcal{TB}_1 = \text{FFN}_1 \circ \text{Attn}_1$ for approximating $(A)$. Our construction depends on the following fact:

$$(A) = \eta\alpha^2 \sum_{j=1}^N \exp(-\|\mathbf{e}_i - \mathbf{e}_j\|^2/\ell)\boldsymbol{\Sigma}\mathbf{s}_j$$

$$= \eta\alpha^2 \sum_{j=1}^N \exp((2r^2 - 2\mathbf{e}_i^\top \mathbf{e}_j)/\ell)\boldsymbol{\Sigma}\mathbf{s}_j$$

$$= \eta\alpha^2 D_i \cdot \text{softmax}\left([(2r^2 - 2\mathbf{e}_i^\top \mathbf{e}_1)/\ell, \cdots, (2r^2 - 2\mathbf{e}_i^\top \mathbf{e}_N)/\ell]\right) \cdot [\boldsymbol{\Sigma}\mathbf{s}_1, \ldots, \boldsymbol{\Sigma}\mathbf{s}_N],$$

where $D_i = \sum_{j=1}^{N} \exp((2r^2 - 2\mathbf{e}_i^\top \mathbf{e}_j)/\ell)$ is the normalizing constant in the softmax activation. Thus, we can use one attention head to first realize the unnormalized version:

$$(A') = \alpha^2 \cdot \text{softmax}\big([(2r^2 - 2\mathbf{e}_i^\top \mathbf{e}_1)/\ell, \cdots, (2r^2 - 2\mathbf{e}_i^\top \mathbf{e}_N)/\ell]\big) \cdot [\boldsymbol{\Sigma}\mathbf{s}_1, \ldots, \boldsymbol{\Sigma}\mathbf{s}_N].$$

**Use** `Attn` **to approximate** $(A')$  In particular, for the first attention head, we choose

$$(\mathbf{Q})^\top \mathbf{K} = \text{diag}\left([\mathbf{0}_{d \times d}, -2\ell^{-1}\mathbf{I}_{d_e}, \mathbf{0}_{d_t \times d_t}, \mathbf{0}_{(5d) \times (5d)}, 2r^2\ell^{-1}, \mathbf{0}_{(3d) \times (3d)}]\right) \text{ and}$$

$$\mathbf{V} = \begin{bmatrix} \mathbf{0}_{(4d+d_e+d_t) \times (2d+d_e+d_t)} & \mathbf{0}_{(4d+d_e+d_t) \times d} & \mathbf{0}_{(4d+d_e+d_t) \times (6d+1)} \\ \mathbf{0}_{d \times (2d+d_e+d_t)} & \boldsymbol{\Sigma} & \mathbf{0}_{d \times (6d+1)} \\ \mathbf{0}_{(4d+1) \times (2d+d_e+d_t)} & \mathbf{0}_{(4d+1) \times d} & \mathbf{0}_{(4d+1) \times (6d+1)} \end{bmatrix}.$$

Thus, we have

$$\texttt{Attn}_1(\mathbf{y}_i) = \Big[\mathbf{x}_i^\top, \mathbf{e}_i^\top, \boldsymbol{\phi}^\top(t), \mathbf{s}_i, f_{\text{mult}}(\alpha^2, \mathbf{s}_i^\top), f_{\text{mult}}(\sigma^2, \mathbf{s}_i^\top),$$

$$D_i^{-1} \sum_{j=1}^{N} \exp((2r^2 - 2\mathbf{e}_i\mathbf{e}_j)/\ell)\boldsymbol{\Sigma} \cdot f_{\text{mult}}(\alpha^2, \mathbf{s}_j^\top), f_{\text{mult}}(\alpha_t, \boldsymbol{\mu}_i)^\top, 1, \mathbf{0}_{3d-1}^\top, i\Big]^\top.$$

To multiply our constructed value by the normalizing constant $D_i$, we will construct a feed-forward layer $\texttt{FFN}_1$ such that the output of the first attention block is

$$\mathcal{TB}(\mathbf{y}_i) = \Big[\mathbf{x}_i^\top, \mathbf{e}_i^\top, \boldsymbol{\phi}^\top(t), \mathbf{s}_i, f_{\text{mult}}(\alpha^2, \mathbf{s}_i^\top), f_{\text{mult}}(\sigma^2, \mathbf{s}_i^\top),$$

$$D_i^{-1} \sum_{j=1}^{N} \exp((2r^2 - 2\mathbf{e}_i\mathbf{e}_j)/\ell)\boldsymbol{\Sigma} \cdot f_{\text{mult}}(\alpha^2, \mathbf{s}_j^\top), f_{\text{mult}}(\alpha_t, \boldsymbol{\mu}_i)^\top, 1, \widehat{D_i}, \mathbf{0}_{3d-2}^\top, i\Big]^\top. \tag{15}$$

Here $\widehat{D_i}$ is an approximation to $D_i$. Now we show the approximation strategy to approximate $D_i$.

**Use** `FFN` **to approximate** $D_i$  Note that we can rewrite $D_i$ as

$$D_i = 1 + \sum_{k=1}^{N-1} (\mathbf{1}\{i \geq k+1\} + \mathbf{1}\{i \leq N-k\})g(k).$$

Where $g(k) = \exp(-f(k)^\nu/\ell)$ is the correlation function. Let $\widehat{D_i}^{(m)} = 1 + \sum_{k=1}^{m} (\mathbf{1}\{i \geq k+1\} + \mathbf{1}\{i \leq N-k\})g(k)$ to be an approximation of $D_i$, we know by the proof of Lemma 4, for any $i \in [N]$,

$$|D(i) - D_m(i)| \leq 2 \sum_{k=m+1}^{N-1} g(k) \leq c^{-\nu}\ell \exp\left(-\frac{2c^\nu m^\nu}{\ell}\right).$$

Thus, to derive an $\epsilon$ error bound, we only need $m$ to be in the order of $m = \mathcal{O}\left((\ell \log(1/\epsilon))^{1/\nu}\right)$. For each $k$, we have

$$(\mathbf{1}\{i \geq k+1\} + \mathbf{1}\{i \leq N-k\})g(k)$$
$$= g(k)(\text{ReLU}(i-k) - \text{ReLU}(i-k-1) + \text{ReLU}(N-k+1-i) - \text{ReLU}(N-k-i))$$

holds for any integer $i$. Thus, the entire function $\widehat{D_i}^{(m)}$ can be expressed by $4m = \mathcal{O}\left((\ell \log(1/\epsilon))^{1/\nu}\right)$ ReLU neurons in the feed-forward network. Specifically, we take $\texttt{FFN}(\mathbf{y}) = \mathbf{W}_2\text{RelU}(\mathbf{W}_1\mathbf{y}) + \mathbf{W}_1$ and choose $\mathbf{W}_1$ as

$$\mathbf{W}_1 = \begin{bmatrix} \mathbf{0}_{D-3d-1}^\top & -1 & \mathbf{0}_{3d-1}^\top & 1 \\ \mathbf{0}_{D-3d-1}^\top & -2 & \mathbf{0}_{3d-1}^\top & 1 \\ \vdots & \vdots & \vdots & \vdots \\ \mathbf{0}_{D-3d-1}^\top & -m-1 & \mathbf{0}_{3d-1}^\top & 1 \\ \mathbf{0}_{D-3d-1}^\top & n & \mathbf{0}_{3d-1}^\top & -1 \\ \vdots & \vdots & \vdots & \vdots \\ \mathbf{0}_{D-3d-1}^\top & N-m & \mathbf{0}_{3d-1}^\top & -1 \end{bmatrix} \in \mathbb{R}^{(2m+2) \times D}.$$

Then we have $\mathbf{W}_1 \cdot \texttt{Attn} \circ \mathcal{TB}_1(\mathbf{y}_i)$ as

$$
\begin{bmatrix}
i - 1 \\
i - 2 \\
i - m - 1 \\
N - i \\
N - i - 1 \\
N - i - m
\end{bmatrix}.
$$

We choose $\mathbf{b}_1 = \mathbf{0}$, $\mathbf{b}_2 = [\mathbf{0}_{D-3d-1}^\top, 1, \mathbf{0}_{3d}^\top]$. Denote

$$\mathbf{w}_g = [g(1), g(2) - g(1), g(3) - g(2), \cdots, g(m) - g(m-1), -g(m)]$$

and let

$$
\mathbf{W}_2 =
\begin{bmatrix}
\mathbf{0}_{(D-3d-1)\times(2m+2)} \\
\mathbf{w}_g \ \ \mathbf{w}_g \\
\mathbf{0}_{(3d)\times(2m+2)}
\end{bmatrix},
$$

we can check that this exactly express $\widehat{D}_i^{(m)}$ and store it in the correponding location according to the equation (15). Taking $\mathcal{TB} = \texttt{FFN} \circ \texttt{Attn}$, we reach our target in equation (15). After that, we could apply another multiplication module $f_{\text{mult}}$ to multiply the gradient component by $\widehat{D}_i$, which gives rise to

$$
f_{\text{mult}} \circ \mathcal{TB} = \Big[ \mathbf{x}_i^\top, \mathbf{e}_i^\top, \boldsymbol{\phi}^\top(t), \mathbf{s}_i, f_{\text{mult}}(\alpha^2, \mathbf{s}_i^\top), f_{\text{mult}}(\sigma^2, \mathbf{s}_i^\top),
$$

$$
f_{\text{mult}}\left( \widehat{D}_i, D_i^{-1} \sum_{j=1}^N \exp((2r^2 - 2\mathbf{e}_i \mathbf{e}_j)/\ell)\boldsymbol{\Sigma} \cdot f_{\text{mult}}(\alpha^2, \mathbf{s}_j^\top) \right),
$$

$$
f_{\text{mult}}(\alpha_t, \boldsymbol{\mu}_i)^\top, 1, \widehat{D}_i, \mathbf{0}_{3d-2}^\top, i \Big]^\top.
$$

Then, we can completely follow the proof of Lemma 10 to constuct the gradient and scale the gradient by the learning rate $\eta_t$, and the approximation error of the gradient only adds an additional term induced by approximating $D_i$, which has been well controlled by setting $m = \mathcal{O}\Big( (\ell \log (1/\epsilon))^{1/\nu} \Big)$ to reach a error level of $\epsilon$. We remark that $\widehat{D}_i$ can be approximated for only once throughout the transformer networks and we can reuse it in each GD block.

**Size of Transformer Architecture with Softmax Activation**   We list here the size of transformer architecture equipped with the softmax activation. Readers may draw a quick comparison with the size of ReLU transformer in Table 3.

Table 4: Size of transformer with Softmax activation for approximating score function (with Gaussian covariance function)

| Input dimension | $D \times N$ with $D = 9d + d_e + d_t + 1$ |
|---|---|
| # of blocks $L$ | $2 + L_{\text{mult}}$ |
| # of attention heads $M$ | 1 |
| Parameter bound $B$ | $\mathcal{O}(N^2 + d + \|\boldsymbol{\Sigma}\|_{\text{F}} + r^2))$ |
| Output range $R_t$ | $2R_0 \sigma_t^{-1}$ |

It is observed that using Softmax activation leads to a reduced number of transformer blocks. The reason behind is that the Gaussian covariance function can be approximately represented by one attention block. The proof is complete.

$\square$

With Lemma 11 and following the proof in Appendix D while omitting the part of truncating $\boldsymbol{\Gamma}$, i.e., setting $J = N$, we could similarly construct a softmax-transformers as

$$\widetilde{\mathbf{s}}_{\text{softmax}} = f_{\text{out}} \circ f_{\text{GD,softmax}} \circ \cdots \circ f_{\text{GD,softmax}} \circ f_\mu \circ f_{\text{in}}.$$

such that $\|\widetilde{\mathbf{s}}_{\text{softmax}} - \nabla \log p_t\|_{L_2(P_t)} \leq \sigma_t^{-2}\epsilon$ for any $t \geq t_0$.

# E  OMITTED PROOFS IN SECTION 5

## E.1  PROOF OF THEOREM 2

Throughout this section, we assume Assumption 1 holds, and there exists an absolute constant $C$ as in the statement of Theorem 2 such that $C^{-1} \leq \lambda(\boldsymbol{\Sigma}) \leq C$ and $\|\boldsymbol{\mu}\|_\infty \leq C$, which means

$$\|\boldsymbol{\Sigma}\|_\infty = \mathcal{O}(1), \quad \|\boldsymbol{\Sigma}\|_{\mathrm{op}} = \mathcal{O}(1), \quad \|\boldsymbol{\Sigma}\|_{\mathrm{F}} = \mathcal{O}(\sqrt{d}) \quad \text{and} \quad \|\boldsymbol{\mu}\|_2 \leq \sqrt{Nd}.$$

Let's first analyse the effect of early-stopping time $t_0$ on the accuracy of distribution estimation. The following lemma presents the distance between the truth distribution $P$ and the early-stopped distribution $P_{t_0}$ in $W_2$ distance.

**Lemma 12.** Suppose $P \sim \mathcal{N}(\boldsymbol{\mu}, \boldsymbol{\Gamma} \otimes \boldsymbol{\Sigma})$, taking the early stopping time $t_0 = o(1)$, we have

$$W_2^2(P, P_{t_0}) \leq (1 - \alpha_{t_0})^2 \|\boldsymbol{\mu}\|_2^2 + C_\nu^2 N d \sigma_{t_0}^2.$$

Here $C_\nu = \lambda_{\max}(\boldsymbol{\Gamma} \otimes \boldsymbol{\Sigma}) \vee 1 \leq 1 + \|\boldsymbol{\Sigma}\|_{\mathrm{op}} \ell \lesssim 1 + \ell$ by (11).

This means that $W_2(P, P_{t_0}) \lesssim C_\nu \sqrt{t_0 N d}$. We defer the proof to Appendix E.1.1.

Then we state our theory on score estimation error. Our score estimator $\widehat{\mathbf{s}}_t$ is chosen to minimize the objective function as

$$\widehat{\mathbf{s}}_t = \arg \min_{s_t \in \mathcal{F}} \frac{1}{n} \sum_{i=1}^n \mathbb{E}_{\mathbf{v}_t | \mathbf{v}_0^{(i)}} \left\| \mathbf{s}_t(\mathbf{v}_t) - \nabla \log p_t(\mathbf{v}_t \mid \mathbf{v}_0^{(i)}) \right\|_2^2.$$

for each time step $t \in [0, T]$. The next proposition provides guarantee on the score estimation error, which we will transfer into the distribution estimation error later.

**Proposition 2.** By taking $\epsilon = \frac{1}{ndN}$ in Theorem 1, we have the expected score estimation error bounded by

$$\mathbb{E}_{\mathcal{D}}[\ell(\widehat{\mathbf{s}})] \lesssim (T + \log(1/t_0)) \log \left(\kappa_t ndN/t_0\right)^{4+1/\nu} \cdot \frac{\ell^{1/\nu} \kappa_{t_0} N d^3}{n}.$$

The proof is provided in Appendix E.2. Although our assumption does not ensure the Novikov's condition to hold for sure, according to Chen et al. (2022b), as long as we have bounded second moment for the score estimation error and finite KL divergence w.r.t. the standard Gaussian, we could still adopt Girsanov's Theorem and bound the KL divergence between the two distribution. We restate the lemma as follows:

**Lemma 13** (Proposition D.1 in Oko et al. (2023), see also Theorem 2 in Chen et al. (2022b)). Let $p_0$ be a probability distribution, and let $Y = \{Y_t\}_{t \in [0,T]}$ and $Y' = \{Y'_t\}_{t \in [0,T]}$ be two stochastic processes that satisfy the following SDEs:

$$dY_t = s(Y_t, t)dt + dW_t, \quad Y_0 \sim p_0$$
$$dY'_t = s'(Y'_t, t)dt + dW_t, \quad Y'_0 \sim p_0$$

We further define the distributions of $Y_t$ and $Y'_t$ by $p_t$ and $p_t$. Suppose that

$$\int_{\mathbf{x}} p_t(\mathbf{x}) \|(\mathbf{s} - \mathbf{s}')(\mathbf{x}, t)\|_2^2 \, d\mathbf{v}_t \leq C \tag{16}$$

for any $t \in [0, T]$. Then we have

$$\mathrm{KL}\left(p_T \| p'_T\right) \leq \int_0^T \frac{1}{2} \int_{\mathbf{x}} p_t(\mathbf{x}) \|(\mathbf{s} - \mathbf{s}')(\mathbf{x}, t)\|_2^2 \, d\mathbf{x} dt.$$

Now we are ready to prove Theorem 2.

*Proof of Theorem 2.* Now we transfer the score estimation error to a TV-distance bound using Lemma 13. Note that under our assumptions and for any $\mathbf{s} \in \mathcal{F}$, we have

$$\int_{\mathbf{v}_t} p_t(\mathbf{v}_t) \|\mathbf{s}_t(\mathbf{v}_t) - \nabla \log p_t(\mathbf{v}_t)\|_2^2 \, d\mathbf{v}_t \lesssim \int_{\mathbf{v}_t} p_t(\mathbf{v}_t) \left( \frac{\|\mathbf{v}_t - \alpha_t \boldsymbol{\mu}\|_2^2}{\sigma_t^4} + \frac{C}{\sigma_t^2} \right) d\mathbf{v}_t \lesssim \frac{1}{\sigma_t^2}.$$

Thus, the condition (16) holds for $t_0 \leq t \leq T$, which means that we could apply Girsanov's theorem in this time range. Remember that the backward process is written as

$$d\mathbf{X}_t^{\leftarrow} = \left[ \frac{1}{2} \mathbf{X}_t^{\leftarrow} + \nabla \log p_{T-t}(\mathbf{X}_t^{\leftarrow}) \right] dt + d\overline{\mathbf{W}}_t, \text{ with } \mathbf{X}_0^{\leftarrow} \sim \mathsf{N}(\mathbf{0}, \mathbf{I}).$$

We denote the distribution of $\mathbf{X}_t^{\leftarrow}$ as $P_{T-t}^{\leftarrow}$. In real setting, we replace $\nabla \log p_t$ by its score estimator $\widehat{\mathbf{s}}_t$, which gives rise to the following backward process:

$$d\widehat{\mathbf{X}}_t^{\leftarrow} = \left[ \frac{1}{2} \widehat{\mathbf{X}}_t^{\leftarrow} + \widehat{\mathbf{s}}_{T-t}(\widehat{\mathbf{X}}_t^{\leftarrow}) \right] dt + d\overline{\mathbf{W}}_t, \text{ with } \mathbf{X}_0^{\leftarrow} \sim \mathsf{N}(\mathbf{0}, \mathbf{I}).$$

We denote the generated distribution of $\widehat{\mathbf{X}}_t^{\leftarrow}$ as $\widehat{P}_{T-t}$. Besides, we consider the truth backward process as the inverse process of the forward one, which is defined as

$$d\mathbf{X}_t'^{\leftarrow} = \left[ \frac{1}{2} \mathbf{X}_t'^{\leftarrow} + \nabla \log p_{T-t}(\mathbf{X}_t'^{\leftarrow}) \right] dt + d\overline{\mathbf{W}}_t \text{ with } \mathbf{X}_0'^{\leftarrow} \sim P_T.$$

We denote the distribution of $\mathbf{X}_t'^{\leftarrow}$ by $P_{T-t}'$, then we have $P_t' \sim P_t$ for any $t \leq T$.

Since $\mathbf{X}'^{\leftarrow}$ and $\mathbf{X}^{\leftarrow}$ are obtained through the same backward SDE but with different initial distributions, by Data Processing Inequality and Pinsker's Inequality (see e.g., Lemma 2 in Canonne (2023)), we have

$$\begin{aligned}
\mathrm{TV}(P_{t_0}, P_{t_0}^{\leftarrow}) &= \mathrm{TV}(P_{t_0}', P_{t_0}^{\leftarrow}) \\
&\lesssim \sqrt{\mathrm{KL}(P_{t_0}' \| P_{t_0}^{\leftarrow})} \\
&\lesssim \sqrt{\mathrm{KL}(P_T \| \mathsf{N}(\mathbf{0}, \mathbf{I}))} \\
&\lesssim \sqrt{\mathrm{KL}(P \| \mathsf{N}(\mathbf{0}, \mathbf{I}))} \exp(-T).
\end{aligned}$$

Thus, we could decompose the TV bound into

$$\begin{aligned}
\mathrm{TV}(P_{t_0}, \widehat{P}_{t_0}) &\lesssim \mathrm{TV}(P_{t_0}, P_{t_0}^{\leftarrow}) + \mathrm{TV}(P_{t_0}^{\leftarrow}, \widehat{P}_{t_0}) \\
&\lesssim \exp(-T) + \sqrt{\int_{t_0}^{T} \frac{1}{2} \int_{\mathbf{v}_t} p_t(\mathbf{v}_t) \| \widehat{\mathbf{s}}(\mathbf{v}_t, \mathbf{y}, t) - \nabla \log p_t(\mathbf{v}_t) \|_2^2 \, d\mathbf{v}_t dt}. \quad (17)
\end{aligned}$$

Thus, by taking expectation over the dataset $\mathcal{D}$ and invoking Jensen inequality, we have

$$\begin{aligned}
\mathbb{E}_{\mathcal{D}} \left[ \mathrm{TV}(P_{t_0}, \widehat{P}_{t_0}) \right] &\lesssim \mathbb{E}_{\mathcal{D}} \mathrm{TV}(P_{t_0}, P_{t_0}^{\leftarrow}) + \mathbb{E}_{\mathcal{D}} \mathrm{TV}(P_{t_0}^{\leftarrow}, \widehat{P}_{t_0}) \\
&\lesssim \exp(-T) + \sqrt{(T + \log(1/t_0)) \kappa_t^2 \ell^{1/\nu} \log \left( \kappa_t n d N t_0^{-1} \right)^{4+1/\nu} \frac{N d^3}{n}}.
\end{aligned}$$

By taking $T = \mathcal{O}(\log n)$ and combining the results in Lemma 12, we have completed our proof. $\quad \square$

### E.1.1 PROOF OF LEMMA 12

*Proof.* Since $P \sim \mathcal{N}(\boldsymbol{\mu}, \boldsymbol{\Gamma} \otimes \boldsymbol{\Sigma})$ and $P_t \sim \mathcal{N}(\alpha_t \boldsymbol{\mu}, \alpha_t^2 \boldsymbol{\Gamma} \otimes \boldsymbol{\Sigma} + \sigma_t \mathbf{I})$, by the formula of the $W_2$ distance between two multivariate Gaussian distributions, we have

$$W_2^2(P, P_{t_0}) = \| \boldsymbol{\mu} - \alpha_t \boldsymbol{\mu} \|_2^2 + \left\| (\boldsymbol{\Gamma} \otimes \boldsymbol{\Sigma})^{1/2} - (\alpha_t^2 \boldsymbol{\Gamma} \otimes \boldsymbol{\Sigma} + \sigma_t \mathbf{I})^{1/2} \right\|_{\mathrm{F}}^2.$$

Suppose $\boldsymbol{\Gamma} \otimes \boldsymbol{\Sigma} = \boldsymbol{P}^\top \boldsymbol{D} \boldsymbol{P}$, where $\boldsymbol{P} \in \mathbb{R}^{dN \times dN}$ is an orthogonal matrix and $\boldsymbol{D} = \mathrm{diag}(\lambda_1, \ldots, \lambda_{dN})$ is a diagonal matrix with $\lambda_i \geq 0$. Then we have

$$\begin{aligned}
\left\| (\boldsymbol{\Gamma} \otimes \boldsymbol{\Sigma})^{1/2} - (\alpha_t^2 \boldsymbol{\Gamma} \otimes \boldsymbol{\Sigma} + \sigma_t \mathbf{I})^{1/2} \right\|_{\mathrm{F}}^2 &= \left\| \boldsymbol{P}^\top \boldsymbol{D}^{1/2} \boldsymbol{P}^\top - \boldsymbol{P}^\top (\alpha_t^2 \boldsymbol{D} + \sigma_t^2 \mathbf{I})^{1/2} \boldsymbol{P} \right\|_{\mathrm{F}}^2 \\
&= \left\| \boldsymbol{D}^{1/2} - (\alpha_t^2 \boldsymbol{D} + \sigma_t^2 \mathbf{I})^{1/2} \right\|_{\mathrm{F}}^2 \\
&= \sum_{i=1}^{Nd} \frac{\sigma_t^4 (\lambda_i - 1)^2}{(\sqrt{\lambda_t} + \sqrt{\alpha_t^2 \lambda_i + \sigma_t^2})^2} \\
&\leq \sum_{i=1}^{Nd} \frac{\sigma_t^4 C_\nu^2}{\sigma_t^2} \\
&\leq N d C_\nu \sigma_t^2.
\end{aligned}$$

By taking $t = t_0$ and plugging the inequality above into the expression of $W_2^2(P, P_{t_0})$, we complete our proof. □

### E.2 PROOF OF PROPOSITION 2

#### E.2.1 ADDITIONAL NOTATIONS

For any score estimator $\widehat{\mathbf{s}}_t$, we denote its population loss $\ell$ as:

$$\ell(\widehat{\mathbf{s}}_t) := \int_{t_0}^T \mathrm{d}t \cdot \mathbb{E}_{\mathbf{v}_t \sim p_t} \|\widehat{\mathbf{s}}_t(\mathbf{v}_t) - \nabla \log p_t(\mathbf{v}_t)\|_2^2$$

and we define the empirical loss $\widehat{\ell}$ as

$$\widehat{\ell}(\widehat{\mathbf{s}}_t) = \frac{1}{n} \sum_{i=1}^n \int_{t_0}^T \mathrm{d}t \cdot \mathbb{E}_{\mathbf{v}_t | \mathbf{v}_0^{(i)}} \left\| \widehat{\mathbf{s}}_t(\mathbf{v}_t) + \frac{\mathbf{v}_t - \alpha_t \mathbf{v}_0^{(i)}}{\sigma_t^2} \right\|_2^2.$$

Here, $\mathcal{D} = \left\{ \mathbf{v}_0^{(i)} \right\}_{i \in [n]}$ are $n$ i.i.d samples from true distribution $P(= P_0)$. When taken expectation over the choice of samples, we have $\mathbb{E}_{\mathcal{D}}[\widehat{\ell}(\widehat{\mathbf{s}}_t)] = \ell(\widehat{\mathbf{s}}_t) + C$ according to Vincent (2011) for any score estimator $\widehat{\mathbf{s}}$. Here $C$ is a constant independent with $\widehat{\mathbf{s}}_t$. Our score estimator $\widehat{\mathbf{s}}_t$ is chosen to minimize the objective function as

$$\widehat{\mathbf{s}}_t = \arg \min_{s_t \in \mathcal{F}} \frac{1}{n} \sum_{i=1}^n \mathbb{E}_{\mathbf{v}_t | \mathbf{v}_0^{(i)}} \left\| s_t(\mathbf{v}_t) - \nabla \log p_t(\mathbf{v}_t \mid \mathbf{v}_0^{(i)}) \right\|_2^2.$$

for each time step $t \in [0, T]$.

#### E.2.2 PROOF OF PROPOSITION 2

The proof follows that of Theorem 4.1 in Fu et al. (2024) by neglecting the conditional information. Specifically, we replace Lemma D.1 in Fu et al. (2024) by Lemma 14, which provides an $2Nd(T + \log(1/t_0))(R_s^2 + 1)$ uniform upper bound on the magnitude of the empirical loss function.

**Lemma 14** (Counterpart of Lemma D.1 in Fu et al. (2024))**.** Then for any score estimator $\widehat{\mathbf{s}} \in \mathcal{T}(D, L, M, B, R_s \sqrt{Nd}\sigma_t^{-1})$ and data point $\mathbf{v}_0$, we have single-point score loss bounded by

$$h(\widehat{\mathbf{s}}, \mathbf{v}_0) := \int_{t_0}^T \mathrm{d}t \cdot \mathbb{E}_{\mathbf{v}_t | \mathbf{v}_0} \left\| \widehat{\mathbf{s}}_t(\mathbf{v}_t) + \frac{\mathbf{v}_t - \alpha_t \mathbf{v}_0}{\sigma_t^2} \right\|_2^2 \le 2Nd(T + \log(1/t_0))(R_s^2 + 1). \quad (18)$$

The proof is provided in Appendix E.2.3. Besides, to implement covering number techniques, we also introduce a truncated loss $\widehat{\ell}^{\mathrm{trunc}}$ as in Fu et al. (2024) and bound its difference with the truth loss with small error. We define the truncated loss as

$$\widehat{\ell}^{\mathrm{trunc}}(\widehat{\mathbf{s}}_t) := \frac{1}{n} \sum_{i=1}^n \int_{t_0}^T \mathrm{d}t \cdot \mathbb{E}_{\mathbf{v}_t | \mathbf{v}_0^{(i)}} \left[ \left\| \widehat{\mathbf{s}}_t(\mathbf{v}_t) + \frac{\mathbf{v}_t - \alpha_t \mathbf{v}_0^{(i)}}{\sigma_t^2} \right\|_2^2 \mathbf{1} \left\{ \left\| v_0^{(i)} \right\|_2 \le R_0 \right\} \right].$$

The follow Lemma aligns with equation (D.12) in Fu et al. (2024).

**Lemma 15** (Truncation error of the truncated loss function)**.** There exists a constant $C_R$ such that for any $\epsilon < 1$, by choosing $R_0 = \sqrt{N\gamma_0 \operatorname{tr}(\boldsymbol{\Sigma})} + C_R \log(Nd/\epsilon) \|\boldsymbol{\Gamma}\|_{\mathrm{F}} \|\boldsymbol{\Sigma}\|_{\mathrm{F}} = \mathcal{O}(\log(Nd/\epsilon)\sqrt{Nd})$, we have for any $\widehat{s} \in \mathcal{F}$,

$$\left| \mathbb{E}_{\mathbf{v}_0} \left[ \int_{t_0}^T \mathrm{d}t \cdot \mathbb{E}_{\mathbf{v}_t | \mathbf{v}_0^{(i)}} \left\| \widehat{\mathbf{s}}_t(\mathbf{v}_t) + \frac{\mathbf{v}_t - \alpha_t \mathbf{v}_0}{\sigma_t^2} \right\|_2^2 \mathbf{1} \{ \|\mathbf{v}_0\| \ge R_0 \} \right] \right| \lesssim (T + \log(1/t_0))\epsilon.$$

Moreover, based on the truncated loss function, we have the following results on bounding the difference between the losses of two score estimators that are close to each other, which enables us to apply the results in Appendix E.3.

**Lemma 16.** Given the truncation radius $R_0 > 0$. Suppose $\widehat{\mathbf{s}}^{(1)}, \widehat{\mathbf{s}}^{(2)} \in \mathcal{F}$ such that $\left\| \widehat{\mathbf{s}}_t^{(1)}(\mathbf{v}) - \widehat{\mathbf{s}}_t^{(2)}(\mathbf{v}) \right\|_2 \le \epsilon$ for any $\|\mathbf{v}\|_2 \le R_0 + \sigma_t C \sqrt{Nd} \log(dN/\epsilon)$ and $t \ge t_0$, where $C$ is an absolute constant. Then we have

$$\left| \widehat{\ell}^{\mathrm{trunc}}(\widehat{\mathbf{s}}^{(1)}) - \widehat{\ell}^{\mathrm{trunc}}(\widehat{\mathbf{s}}^{(2)}) \right| \le 2\epsilon(T + \log(1/t_0)) \left( \sqrt{Nd}(R_s + C \log(dN/\epsilon)) + 2R_0 + 1 \right).$$

Besides, combining the results of Lemmas 23 and 16, we direcly have the following results on bounding the covering number of the truncated loss function class, which aligns with Lemma D.3 in Fu et al. (2024).

**Lemma 17** (Counterpart of Lemma D.3 in Fu et al. (2024)). We consider the truncated loss function class defined as

$$
\mathcal{S}(R_0) = \left\{ h^{\text{trunc}}(\mathbf{s}, \cdot) : \mathbb{R}^d \to \mathbb{R} \middle| \mathbf{s} \in \mathcal{F} \right\}. \tag{19}
$$

Here $h^{\text{trunc}}(\mathbf{s}, \mathbf{v}) = h(\mathbf{s}, \mathbf{v}) \mathbf{1} \{ \|\mathbf{v}\|_2 \leq R_0 \}$. Then the log-covering number of this loss truncated function class with output range lying in the Euclidean ball with radius $R_0$ can be bounded by

$$
\log \mathcal{N}(\delta; \mathcal{S}(R_0), \|\cdot\|_\infty) \leq 8D^2 M \cdot \left( L^2 \log L_{\mathcal{F}} A_{\mathcal{F}} + \log \frac{24 R_2 B^2 MLN^{3/2}}{\epsilon_\delta} \right) \tag{20}
$$

where $R_2$ satisfies $R_2 \leq (r + C_{\text{diff}})\sqrt{N} + R_0 + \sigma_t C \sqrt{Nd} \log(dN/\epsilon_\delta)$, and $\epsilon_\delta$ satisfies

$$
\delta = 2\epsilon_\delta (T + \log(1/t_0)) \left( \sqrt{Nd}(R_s + C \log(dN/\epsilon_\delta)) + 2R_0 + 1 \right). \tag{21}
$$

Note that if we take $\delta = (ndNt_0^{-1})^{-C}$ for some constant $C$, we have $\log \epsilon_\delta = \mathcal{O}(\log(ndNt_0^{-1}))$. The proofs of all the supporting lemmas above are provided in Appendix E.2.3.

With the lemmas and statements above, we can completely follow the main proof of Theorem 4.1 of Fu et al. (2024) to prove Proposition 2.

*Proof of Proposition 2.* First we set $d_y = 0$ in Fu et al. (2024) for our unconditioned setting. Then by choosing the covering accuracy $\delta$ in Lemma 17 and taking the truncation range as in Lemma 15, i.e., $R_0 = \mathcal{O}(\log(Nd/\epsilon_\delta)\sqrt{Nd})$, we reproduce (D.17) in Fu et al. (2024) as

$$
\begin{aligned}
\mathbb{E}_{\mathcal{D}}[\ell(\widehat{\mathbf{s}})] &\leq 2 \inf_{\mathbf{s} \in \mathcal{F}} \int_{t_0}^T \mathbb{E}_{\mathbf{v}_t} \|\mathbf{s}(\mathbf{v}_t) - \nabla \log p_t(\mathbf{v}_t)\|_2^2 \, \mathrm{d}t \\
&\quad + \frac{(T + \log(1/t_0))}{n} \cdot \log \mathcal{N} + 2(T + \log(1/t_0))\epsilon_\delta + 7\delta. \\
&\leq 2(T + \log(1/t_0))\epsilon \\
&\quad + \frac{2Nd(T + \log(1/t_0))(R_s^2 + 1)}{n} \cdot 8D^2 M \cdot \left( L^2 \log L_{\mathcal{F}} A_{\mathcal{F}} + \log \frac{24 R_2 B^2 MLN^{3/2}}{\epsilon_\delta} \right) \\
&\quad + 2(T + \log(1/t_0))\epsilon_\delta + 14\epsilon_\delta (T + \log(1/t_0)) \left( \sqrt{Nd}(R_s + C \log(dN/\epsilon_\delta)) + 2R_0 + 1 \right).
\end{aligned}
$$

In the inequality, we invoke (20) and substititue $\delta$ by its expression w.r.t. $\epsilon_\delta$ according to (21). Choosing the both the approximation error and the covering accuracy as $\epsilon = \epsilon_\delta = 1/n$ and plugging the corresponding parameters about the size of the transformer classes according to Theorem 1 gives rise to

$$
\mathbb{E}_{\mathcal{D}}[\ell(\widehat{\mathbf{s}})] \lesssim (T + \log(1/t_0))\kappa_t^2 \ell^{1/\nu} \log \left( \kappa_t ndNt_0^{-1} \right)^{4+1/\nu} \frac{Nd^3}{n}.
$$

The proof is complete. □

### E.2.3 Proofs of Other Supporting Lemmas for Proposition 2

*Proof of Lemma 14.* Notice that when $\mathbf{v}_t \sim p_t(\cdot \mid \mathbf{v}_0^{(i)})$, we have $\mathbf{v}_t - \alpha \mathbf{v}_0^{(i)} = \sigma_t z$ where $z \sim \mathcal{N}(0, \mathbf{I}_{Nd})$ is a standard Gaussian variable. Therefore,

$$
\begin{aligned}
\mathbb{E}_{\mathbf{v}_t|\mathbf{v}_0^{(i)}} \left\| \widehat{\mathbf{s}}_t(\mathbf{v}_t) + \frac{\mathbf{v}_t - \alpha_t \mathbf{v}_0^{(i)}}{\sigma_t^2} \right\|_2^2 &\leq 2\mathbb{E}_{\mathbf{v}_t|\mathbf{v}_0^{(i)}} \|\widehat{\mathbf{s}}_t(\mathbf{v}_t)\|_2^2 + 2\mathbb{E}_{\mathbf{z} \sim \mathcal{N}(0, \mathbf{I})} \|\mathbf{z}/\sigma_t\|_2^2 \\
&\leq 2R_s^2 Nd/\sigma_t^2 + 2Nd/\sigma_t^2 = \frac{2Nd(R_s^2 + 1)}{\sigma_t^2}.
\end{aligned}
$$

Now we can take integral over $t \in [t_0, T]$ and obtain that:

$$\int_{t_0}^T \mathrm{d}t \cdot \mathbb{E}_{\mathbf{v}_t | \mathbf{v}_0^{(i)}} \left\| \widehat{\mathbf{s}}_t(\mathbf{v}_t) + \frac{\mathbf{v}_t - \alpha_t \mathbf{v}_0^{(i)}}{\sigma_t^2} \right\|_2^2 \leq 2Nd(R_s^2 + 1) \cdot \int_{t_0}^T \frac{e^t \mathrm{d}t}{e^t - 1} \leq 2Nd(T + \log(1/t_0))(R_s^2 + 1).$$

Here, we use the fact that

$$\int_{t_0}^T \frac{e^t \mathrm{d}t}{e^t - 1} = \log(e^T - 1) - \log(e^{t_0} - 1) \leq T + \log(1/t_0).$$

The proof is complete. $\qquad\square$

To prove Lemma 15, we first need to bound the range of the data with high probability.

**Lemma 18** (Range of the data). Given $\delta > 0$, with probability $1 - 2n \exp(-C\delta)$ over the dataset, Here $C$ is the absolute constant $C_2/\sqrt{3}$ in Lemma 24.

*Proof of Lemma 18.* This statement is directly related to the concentration of high-dimensional Gaussian distribution. For a random variable $\mathbf{v} \sim \mathcal{N}(\mathbf{0}, \mathbf{\Sigma}_0)$ where $\mathbf{\Sigma}_0 \in \mathbb{R}^{Nd \times Nd}$ is the covariance matrix. Then, we use the polynomial concentration lemma (Lemma 24) and let $g(\cdot) = \|\cdot\|_2^2$ be the 2-degree polynomial applied on the Gaussian. Then, we have:

$$\mathbb{E}[g(v)] = \mathrm{tr}(\mathbf{\Sigma}_0), \quad \mathbb{E}[g(v)^2] = 3 \sum_{i=1}^{Nd} (\mathbf{\Sigma}_0)_{ii}^2 + 2 \sum_{i<j} \left( (\mathbf{\Sigma}_0)_{ij} + (\mathbf{\Sigma}_0)_{ji} \right)^2 \leq 3 \|\mathbf{\Sigma}_0\|_F^2.$$

Therefore, we apply Lemma 24 and conclude that with probability at least $1 - 2\exp(-C\delta)$ (where $C$ is the $C_2$ in Lemma 24), we have:

$$\left| \|v\|_2^2 - \mathbb{E}[\|v\|_2^2] \right| \leq \delta \sqrt{\mathrm{Var}(\|v\|_2^2)} \leq \sqrt{3}\delta \cdot \|\mathbf{\Sigma}_0\|_F.$$

For our case, $\mathbf{\Sigma}_0 = \mathbf{\Gamma} \otimes \mathbf{\Sigma}$, and therefore we can conclude that: with probability at least $1 - 2n \exp(-C\delta)$:

$$\|\mathbf{v}_0^{(i)}\|_2^2 \leq \mathrm{tr}(\mathbf{\Gamma}) \cdot \mathrm{tr}(\mathbf{\Sigma}) + \sqrt{3}\delta \|\mathbf{\Gamma}\|_F \|\mathbf{\Sigma}\|_F \leq N\gamma_0 \mathrm{tr}(\mathbf{\Sigma}) + \sqrt{3}\delta \|\mathbf{\Gamma}\|_F \|\mathbf{\Sigma}\|_F.$$

holds for $\forall i \in [n]$. Finally, we replace $\delta$ with $\delta/\sqrt{3}$ and it comes to our conclusion. $\qquad\square$

Now we are ready to prove Lemma 15.

*Proof of Lemma 15.* This conclusion can be made by combining the two lemmas above. By using Cauchy-Schwarz inequality and Lemma 14, we have

$$\left| \mathbb{E}_{\mathbf{v}_0} \left[ \int_{t_0}^T \mathrm{d}t \cdot \mathbb{E}_{\mathbf{v}_t | \mathbf{v}_0^{(i)}} \left\| \widehat{\mathbf{s}}_t(\mathbf{v}_t) + \frac{\mathbf{v}_t - \alpha_t \mathbf{v}_0^{(i)}}{\sigma_t^2} \right\|_2^2 \mathbf{1}\{\|\mathbf{v}_0\| \geq R_0\} \right] \right|$$

$$\leq 2Nd(R_s^2 + 1)(T + \log(1/t_0)) \cdot \sqrt{\mathbb{P}[\|\mathbf{v}_0\|_2^2 \geq R_0^2]}$$

$$\leq 2Nd(R_s^2 + 1)(T + \log(1/t_0)) \cdot \sqrt{2} \exp\left(-CC_R \log(Nd/\varepsilon)/2\right).$$

Let $C_R = 2/C$, then we have:

$$\left| \mathbb{E}_{\mathbf{v}_0} \left[ \int_{t_0}^T \mathrm{d}t \cdot \mathbb{E}_{\mathbf{v}_t | \mathbf{v}_0^{(i)}} \left\| \widehat{\mathbf{s}}_t(\mathbf{v}_t) + \frac{\mathbf{v}_t - \alpha_t \mathbf{v}_0^{(i)}}{\sigma_t^2} \right\|_2^2 \mathbf{1}\{\|\mathbf{v}_0\|_2 \geq R_0\} \right] \right| \lesssim \varepsilon(T + \log(1/t_0)),$$

which comes to our conclusion. $\qquad\square$

*Proof of Lemma 16.* For any datapoint $\mathbf{v}_0$ such that $\|\mathbf{v}_0\| \leq R_0$ and diffusion time $t \geq t_0$, we have

$$\left| \mathbb{E}_{\mathbf{v}_t|\mathbf{v}_0} \left\| \widehat{\mathbf{s}}_t^{(1)}(\mathbf{v}_t) + \frac{\mathbf{v}_t - \alpha_t \mathbf{v}_0}{\sigma_t^2} \right\|_2^2 - \mathbb{E}_{\mathbf{v}_t|\mathbf{v}_0^{(i)}} \left\| \widehat{\mathbf{s}}_t^{(2)}(\mathbf{v}_t) + \frac{\mathbf{v}_t - \alpha_t \mathbf{v}_0}{\sigma_t} \right\|_2^2 \right|$$

$$= \left| \mathbb{E}_{\mathbf{z} \in \mathcal{N}(\mathbf{0}, \mathbf{I})} \left[ \left\| \widehat{\mathbf{s}}_t^{(1)}(\alpha_t \mathbf{v}_0 + \sigma_t \mathbf{z}) + \frac{\mathbf{z}}{\sigma_t} \right\|_2^2 - \left\| \widehat{\mathbf{s}}_t^{(2)}(\alpha_t \mathbf{v}_0 + \sigma_t \mathbf{z}) + \frac{\mathbf{z}}{\sigma_t} \right\|_2^2 \right] \right|$$

$$= \left| \mathbb{E}_{\mathbf{z} \in \mathcal{N}(\mathbf{0}, \mathbf{I})} \left[ \sigma_t^{-1} \left( (\widehat{\mathbf{s}}_t^{(1)} - \widehat{\mathbf{s}}_t^{(2)})(\alpha_t \mathbf{v}_0 + \sigma_t \mathbf{z}) \right)^\top \left( \sigma_t(\widehat{\mathbf{s}}_t^{(1)} + \widehat{\mathbf{s}}_t^{(2)})(\alpha_t \mathbf{v}_0 + \sigma_t \mathbf{z}) + 2\mathbf{z} \right) \right] \right|$$

$$\leq \sigma_t^{-1} \mathbb{E}_{\mathbf{z} \sim \mathcal{N}(\mathbf{0}, \mathbf{I})}[\|\phi_1(\mathbf{z})\|_2 \|\phi_2(\mathbf{z})\|_2],$$

where $\phi_1(\mathbf{z}) = (\widehat{\mathbf{s}}_t^{(1)} - \widehat{\mathbf{s}}_t^{(2)})(\alpha_t \mathbf{v}_0 + \sigma_t \mathbf{z})$ and $\phi_2(\mathbf{z}) = \sigma_t(\widehat{\mathbf{s}}_t^{(1)} + \widehat{\mathbf{s}}_t^{(2)})(\alpha_t \mathbf{v}_0 + \sigma_t \mathbf{z}) + 2\mathbf{z}$. By the upper bound of $s^{(i)}$, we know that

$$\|\phi_1(\mathbf{z})\|_2 \leq 2R_s\sqrt{Nd}\sigma_t^{-1}, \quad \text{and} \quad \|\phi_2(\mathbf{z})\|_2 \leq 2R_s\sqrt{Nd} + 2\|\mathbf{z}\|_2.$$

Thus, they both have sub-linear growth with respect to $\|\mathbf{z}\|_2$. Now we can decompose $\mathbb{E}_{\mathbf{z} \sim \mathcal{N}(\mathbf{0}, \mathbf{I})}[\|\phi_1(\mathbf{z})\|_2 \|\phi_2(\mathbf{z})\|_2]$ by

$$\mathbb{E}_{\mathbf{z} \sim \mathcal{N}(\mathbf{0}, \mathbf{I})}[\|\phi_1(\mathbf{z})\|_2 \|\phi_2(\mathbf{z})\|_2] = \mathbb{E}_{\mathbf{z} \sim \mathcal{N}(\mathbf{0}, \mathbf{I})}[\|\phi_1(\mathbf{z})\|_2 \|\phi_2(\mathbf{z})\|_2 \mathbf{1}\{\|\alpha_t \mathbf{v}_0 + \sigma_t \mathbf{z}\|_2 \leq R_1\}]$$

$$+ \mathbb{E}_{\mathbf{z} \sim \mathcal{N}(\mathbf{0}, \mathbf{I})}[\|\phi_1(\mathbf{z})\|_2 \|\phi_2(\mathbf{z})\|_2 \mathbf{1}\{\|\alpha_t \mathbf{v}_0 + \sigma_t \mathbf{z}\|_2 \geq R_1\}]$$

$$\leq 2\epsilon(R_s\sqrt{Nd} + \sigma_t^{-1}(R_0 + R_1))$$

$$+ 4R_s\sqrt{Nd}\sigma_t^{-1} \underbrace{\mathbb{E}_{\mathbf{z}}\left[(R_s\sqrt{Nd} + \|\mathbf{z}\|_2)\mathbf{1}\{\|\alpha_t \mathbf{v}_0 + \sigma_t \mathbf{z}\|_2 \geq R_1\}\right]}_{A}.$$

In the inequality, we invoke $\|\phi_1(\mathbf{z})\|_2 \leq \epsilon$ and $\|\mathbf{z}\|_2 \leq \sigma_t^{-1}(R_0 + R_1)$ for $\|\alpha_t \mathbf{v}_0 + \sigma_t \mathbf{z}\|_2 \leq R_1$. By Cauchy inequality, we can bound $A$ by

$$A \leq \mathbb{E}_{\mathbf{z}}\left[(R_s\sqrt{Nd} + \|\mathbf{z}\|_2)^2\right]^{1/2} \Pr\left[\|\alpha_t \mathbf{v}_0 + \sigma_t \mathbf{z}\|_2 \geq R_1\right]^{1/2}$$

$$\leq \sqrt{(2R_s^2 + 2)Nd \cdot \Pr\left[\|\mathbf{z}\|_2 \geq \sigma_t^{-1}(R_1 - \alpha_t R_0)\right]}.$$

Since $\mathbf{z}$ is standard Gaussian, we can set $R_1 = R_0 + \sigma_t C\sqrt{Nd}\log(dN/\epsilon)$ for some absolute constant $C$ so that $A \leq \epsilon/(4\sqrt{Nd})$. Altogether, we have

$$\mathbb{E}_{\mathbf{z} \sim \mathcal{N}(\mathbf{0}, \mathbf{I})}[\|\phi_1(\mathbf{z})\|_2 \|\phi_2(\mathbf{z})\|_2] \leq 2\epsilon\left(R_s\sqrt{Nd} + \sigma_t^{-1}(2R_0 + 1) + C\sqrt{Nd}\log(dN/\epsilon)\right).$$

Plugging the inequality into the expression of $\left| \widehat{\ell}^{\mathrm{trunc}}(\widehat{\mathbf{s}}^{(1)}) - \widehat{\ell}^{\mathrm{trunc}}(\widehat{\mathbf{s}}^{(2)}) \right|$, we have

$$\left| \widehat{\ell}^{\mathrm{trunc}}(\widehat{\mathbf{s}}^{(1)}) - \widehat{\ell}^{\mathrm{trunc}}(\widehat{\mathbf{s}}^{(2)}) \right| \leq \int_{t_0}^{T} \sigma_t^{-1} 2\epsilon\left(R_s\sqrt{Nd} + \sigma_t^{-1}(2R_0 + 1) + C\sqrt{Nd}\log(dN/\epsilon)\right)$$

$$\leq \int_{t_0}^{T} \sigma_t^{-2} 2\epsilon\left(R_s\sqrt{Nd} + (2R_0 + 1) + C\sqrt{Nd}\log(dN/\epsilon)\right)$$

$$\leq 2\epsilon(T + \log(1/t_0))\left(\sqrt{Nd}(R_s + C\log(dN/\epsilon)) + 2R_0 + 1\right).$$

The proof is complete. $\qquad\qquad\qquad\qquad\qquad\qquad\qquad\qquad\qquad\qquad\qquad\qquad\square$

### E.3 COVERING NUMBER OF THE MULTI-LAYER TRANSFORMERS

The score network we apply satisfies the following form:

$$f = f_l \circ f_{l-1} \circ \ldots \circ f_1$$

where the total layer number $l = 2L$ (which consists of $L$ feed-forward layers and $L$ Transformer layers) and $f_1, f_2, \ldots, f_l$ are either attention layers or feed-forward networks whose input and output lies in $\mathbb{R}^{D \times N}$. Denote $\mathcal{B}(R) = \{\mathbf{X} \in \mathbb{R}^{D \times N} : \|\mathbf{X}\|_{\mathrm{F}} \leq R\}$ is a $d$-dimensional ball with radius $R$. Assume there exists a sequence of radius $R_0, R_1, \ldots, R_l > 0$ (which will be determined later)

such that $f_i : \mathcal{B}(R_{i-1}) \to \mathcal{B}(R_i)$ holds for $\forall i = 0, 1, \ldots, l$. Then, we need to compute the covering number with respect to the $l_\infty$ norm of the function space constructed by $f : \mathcal{B}(R_0) \to \mathcal{B}(R_l)$ with the form above.

Notice that for two such functions $f = f_l \circ f_{l-1} \circ \ldots \circ f_1$ and $f' = f'_l \circ f'_{l-1} \circ \ldots \circ f'_1$, the $l_\infty$ norm of their difference can be upper bounded by the following lemma:

**Lemma 19.** For functions $\{f_i : \mathbb{R}^{d \times N} \to \mathbb{R}^{d \times N}\}_{i \in [l]}$ and $\{f'_i : \mathbb{R}^{d \times N} \to \mathbb{R}^{d \times N}\}_{i \in [l]}$, we denote their composition as $f := f_l \circ f_{l-1} \circ \ldots \circ f_1$ and $f' := f'_l \circ f'_{l-1} \circ \ldots \circ f'_1$, where $f, f' : \mathbb{R}^{d \times N} \to \mathbb{R}^{d \times N}$. For any matrix-to-matrix function $g$, we denote its $\|\cdot\|_{F,\infty}$ norm as:
$$\|g\|_{F,\infty} := \sup_{\mathbf{X}} \|g(\mathbf{X})\|_F$$
and its Lipschitz continuity as
$$\mathrm{Lip}(g) := \sup_{\mathbf{X},\mathbf{X}'} \frac{\|g(\mathbf{X}) - g(\mathbf{X}')\|_F}{\|\mathbf{X} - \mathbf{X}'\|_F},$$
which is a simple extension from the Lipschitz continuity with respect to $l_2$ norm of vectors. Then, it holds that:
$$\|f - f'\|_{F,\infty} \le \sum_{i=1}^{l} \|f_i - f'_i\|_{F,\infty} \cdot \prod_{j=i+1}^{l} \mathrm{Lip}(f_j).$$

*Proof.* For any $k = 2, 3, \ldots, l$ and $x \in \mathcal{B}(R_0)$, it holds that:
$$\|f_k \circ \ldots f_1(x) - f'_k \circ \ldots f'_1(x)\|_{F,\infty}$$
$$\le \|f'_k \circ f'_{k-1} \circ \ldots \circ f'_1(\mathbf{X}) - f_k \circ f'_{k-1} \circ \ldots \circ f'_1(\mathbf{X})\|_{F,\infty}$$
$$+ \|f_k \circ f'_{k-1} \circ \ldots \circ f'_1(\mathbf{X}) - f_k \circ f_{k-1} \circ \ldots \circ f_1(\mathbf{X})\|_{F,\infty}$$
$$\le \|f'_k - f_k\|_{F,\infty} + \mathrm{Lip}(f_k) \cdot \|f'_{k-1} \circ \ldots \circ f'_1 - f_{k-1} \circ \ldots \circ f_1\|_{F,\infty}.$$
After taking maximum over $x \in \mathcal{B}(R_0)$, we conclude that
$$\|f_k \circ \ldots f_1 - f'_k \circ \ldots f'_1\|_{F,\infty} \le \|f_k - f'_k\|_{F,\infty} + \mathrm{Lip}(f_k) \cdot \|f_{k-1} \circ \ldots f_1 - f'_{k-1} \circ \ldots f'_1\|_{F,\infty}.$$
By using the method of induction, we can easily derive our conclusion. $\qquad\square$

Now, in order to compute the covering number of the function space constructed by functions $f$ with the form $f = f_l \circ f_{l-1} \circ \ldots \circ f_1$ where $f_i \in \mathcal{F}_i$, we firstly need to bound the Lipschitz constants for the function classes $\mathcal{F}_i$. Also, we need to estimate the covering number of each $\mathcal{F}_i$.

**Lemma 20.** For the function space of feed-forward network
$$\mathcal{F}^{\text{FFN}} = \Big\{ \text{FFN} : \mathbb{R}^{D \times N} \to \mathbb{R}^{D \times N}, \mathbf{Y} \mapsto \mathbf{Y} + \mathbf{W}_2 \cdot \text{ReLU}(\mathbf{W}_1 \mathbf{Y} + \mathbf{b}_2 \mathbf{1}^\top) + \mathbf{b}_1 \mathbf{1}^\top :$$
$$\|\mathbf{W}_1\|_F, \|\mathbf{W}_2\|_F, \|\mathbf{b}_1\|_2, \|\mathbf{b}_2\|_2 < B, \mathbf{Y} \in \mathcal{B}(R) \Big\},$$
then all functions in the class $\mathcal{F}^{\text{FFN}}$ are $(1 + B^2)$-Lipschitz. The covering number can be bounded as:
$$\log \mathcal{N}(\delta; \mathcal{F}^{\text{FFN}}, \|\cdot\|_{F,\infty}) \le 4D^2 \log \frac{12B^2(R + \sqrt{N})}{\delta}.$$

*Proof.* For $\forall f \in \mathcal{F}^{\text{FFN}}$ with $f(\mathbf{Y}) = \mathbf{Y} \mapsto \mathbf{Y} + \mathbf{W}_2 \cdot \text{ReLU}(\mathbf{W}_1 Y + \mathbf{b}_2 \mathbf{1}^\top) + \mathbf{b}_1 \mathbf{1}^\top$, by the additivity of Lipschitz constant and the fact that ReLU is 1-Lipschitz continuous, we can conclude that the Lipschitz constant of function $f$ is no larger than $1 + \|\mathbf{W}_2\|_2 \cdot \|\mathbf{W}_1\|_2 \le 1 + \|\mathbf{W}_1\|_F \|\mathbf{W}_2\|_F < 1 + c^2$, which means $f$ is $(1 + c^2)$-Lipschitz continuous. On the other hand, for two functions $f, g \in \mathcal{F}^{\text{FFN}}$ where $f(\mathbf{Y}) = \mathbf{Y} + \mathbf{W}_2 \cdot \text{ReLU}(\mathbf{W}_1 \mathbf{Y} + \mathbf{b}_2 \mathbf{1}^\top) + \mathbf{b}_1 \mathbf{1}^\top$ and $g(\mathbf{Y}) = \mathbf{Y} + \mathbf{W}'_2 \cdot \text{ReLU}(\mathbf{W}'_1 \mathbf{Y} + \mathbf{b}'_2 \mathbf{1}^\top) + \mathbf{b}'_1 \mathbf{1}^\top$. Then for $\forall \mathbf{Y} \in \mathcal{B}(R)$, we have:
$$\|f(\mathbf{Y}) - g(\mathbf{Y})\|_F \le \sqrt{N} \cdot \|\mathbf{b}_1 - \mathbf{b}'_1\|_2 + \|\mathbf{W}'_2 \cdot (\text{ReLU}(\mathbf{W}_1 \mathbf{Y} + \mathbf{b}_2 \mathbf{1}^\top) - \text{ReLU}(\mathbf{W}'_1 \mathbf{Y} + \mathbf{b}'_2 \mathbf{1}^\top))\|_F$$
$$+ \|(\mathbf{W}_2 - \mathbf{W}'_2) \cdot \text{ReLU}(\mathbf{W}_1 \mathbf{Y} + \mathbf{b}_2 \mathbf{1}^\top)\|_F$$
$$\le \sqrt{N} \cdot \|\mathbf{b}_1 - \mathbf{b}'_1\|_2 + \|\mathbf{W}_2\|_F \cdot \left(\sqrt{N}\|\mathbf{b}_2 - \mathbf{b}'_2\|_2 + R\|\mathbf{W}_1 - \mathbf{W}'_1\|_F\right)$$
$$+ \|\mathbf{W}_2 - \mathbf{W}'_2\|_F \cdot (R\|\mathbf{W}_1\|_F + \sqrt{N}\|\mathbf{b}_2\|_2)$$
$$\le \sqrt{N} \cdot \|\mathbf{b}_1 - \mathbf{b}'_1\|_2 + B\sqrt{N} \cdot \|\mathbf{b}_2 - \mathbf{b}'_2\|_2 + BR \cdot \|\mathbf{W}_1 - \mathbf{W}'_1\|_F + B(R + \sqrt{N}) \cdot \|\mathbf{W}_2 - \mathbf{W}'_2\|_F.$$

Therefore, it holds that:

$$\|f - g\|_{\mathrm{F},\infty} \leq \sqrt{N} \cdot \|\mathbf{b}_1 - \mathbf{b}_1'\|_2 + B\sqrt{N} \cdot \|\mathbf{b}_2 - \mathbf{b}_2'\|_2 + BR \cdot \|\mathbf{W}_1 - \mathbf{W}_1'\|_{\mathrm{F}} + B(R + \sqrt{N}) \cdot \|\mathbf{W}_2 - \mathbf{W}_2'\|_{\mathrm{F}},$$

which leads to the upper bound of covering number:

$$\mathcal{N}(\delta; \mathcal{F}^{\mathtt{FFN}}, \|\cdot\|_{\mathrm{F},\infty}) \leq \mathcal{N}(\delta/4\sqrt{N}; \mathcal{M}_1, \|\cdot\|_2) \cdot N(\delta/4B\sqrt{N}; \mathcal{M}_1, \|\cdot\|_2) \cdot$$
$$\mathcal{N}(\delta/4BR; \mathcal{M}_2, \|\cdot\|_{\mathrm{F}}) \cdot \mathcal{N}(\delta/4B(R + \sqrt{N}); \mathcal{M}_2, \|\cdot\|_{\mathrm{F}})$$

where $\mathcal{M}_1 = \{\mathbf{v} \in \mathbb{R}^D : \|\mathbf{v}\|_2 \leq B\}, \mathcal{M}_2 = \{\mathbf{M} \in \mathbb{R}^{D \times D} : \|\mathbf{M}\|_{\mathrm{F}} < B\}$. It is well-known that for $\forall \varepsilon > 0$, we have $\mathcal{N}(\varepsilon; \mathcal{M}_1, \|\cdot\|_2) \leq (3B/\varepsilon)^D$ and $\mathcal{N}(\varepsilon; \mathcal{M}_2, \|\cdot\|_{\mathrm{F}}) \leq (3B/\varepsilon)^{D^2}$. To sum up, we finally conclude that

$$\log \mathcal{N}(\delta; \mathcal{F}^{\mathtt{FFN}}, \|\cdot\|_{\mathrm{F},\infty}) \leq 4D^2 \log \frac{12B^2(R + \sqrt{N})}{\delta}.$$

$\square$

The proof is complete.

**Lemma 21.** For the function space of attention network

$$\mathcal{F}^{\mathtt{Attn}} = \Big\{ \mathtt{Attn} : \mathbb{R}^{D \times N} \to \mathbb{R}^{D \times N}, \ \mathbf{Y} \mapsto \mathbf{Y} + \sum_{m=1}^{M} \mathbf{V}^m \mathbf{Y} \cdot \sigma \left( (\mathbf{Q}^m \mathbf{Y})^\top \mathbf{K}^m \mathbf{Y} \right) :$$

$$\|\mathbf{V}^m\|_{\mathrm{F}}, \|\mathbf{Q}^m\|_{\mathrm{F}}, \|\mathbf{K}^m\|_{\mathrm{F}} < B \ \text{ for } \forall m \in [M], \mathbf{Y} \in \mathcal{B}(R) \Big\},$$

then all functions in the class $\mathcal{F}^{\mathtt{Attn}}$ are $(1 + BM\sqrt{N} + 2B^3 M R^2 N)$-Lipschitz. The covering number can be bounded as:

$$\log \mathcal{N}(\delta; \mathcal{F}^{\mathtt{Attn}}, \|\cdot\|_{\mathrm{F},\infty}) \leq D^2 M \cdot \log \frac{6B^2 M R^3 N^{3/2}}{\delta}.$$

Here, the metric $\|f\|_{\mathrm{F},\infty} := \sup_{\mathbf{Y} \in \mathcal{B}(R)} \|f(\mathbf{Y})\|_{\mathrm{F}}$.

*Proof.* Notice that for any two functions $f_1, f_2$ and $\mathbf{Y}, \mathbf{Y}'$, we have:

$$\|f_1(\mathbf{Y}) f_2(\mathbf{Y}) - f_1(\mathbf{Y}') f_2(\mathbf{Y}')\|_{\mathrm{F}} \leq \|f_1(\mathbf{Y}')\|_{\mathrm{F}} \cdot \|f_2(\mathbf{Y}) - f_2(\mathbf{Y}')\|_{\mathrm{F}} + \|f_2(\mathbf{Y})\|_{\mathrm{F}} \cdot \|f_1(\mathbf{Y}) - f_1(\mathbf{Y}')\|_{\mathrm{F}}.$$

It means that the Lipschitz constant of $\mathbf{V}^m \mathbf{Y} \cdot \sigma \left( (\mathbf{Q}^m \mathbf{Y})^\top \mathbf{K}^m \mathbf{Y} \right)$ is no larger than $\|\mathbf{V}^m\|_2 \cdot \sqrt{N} + R\sqrt{N} \|\mathbf{V}^m\|_2 \cdot 2\|\mathbf{Q}^{m\top} \mathbf{K}^m\|_{\mathrm{F}} \cdot R\sqrt{N}$. Here, we use the fact that both ReLU and softmax activation function over vectors is 1-Lipschitz under $l_2$ norm and that over matrices is 1-Lipschitz under Frobenius norm. See Appendix G.2. Also, $(\mathbf{Q}^m \mathbf{Y})^\top \mathbf{K}^m \mathbf{Y}$ is $2\|\mathbf{Q}^{m\top} \mathbf{K}^m\|_{\mathrm{F}} \cdot R\sqrt{N}$-Lipschitz under Frobenius norm when $\mathbf{Y} \in \mathcal{B}(R)$. Therefore, in the function class $\mathcal{F}^{\mathtt{Attn}}$, all functions included are $(1 + BM\sqrt{N} + 2B^3 M R^2 N)$-Lipschitz continuous, which comes to our conclusion. For the covering number, given any two functions $f, g \in \mathcal{F}^{\mathtt{Attn}}$ with their corresponding parameter sets $\{\mathbf{V}_1^m, \mathbf{Q}_1^m, \mathbf{K}_1^m : m \in [M]\}$ and $\{\mathbf{V}_2^m, \mathbf{Q}_2^m, \mathbf{K}_2^m : m \in [M]\}$, then:

$$\|f(\mathbf{Y}) - g(\mathbf{Y})\|_{\mathrm{F}} \leq \sum_{m=1}^{M} \left\| (\mathbf{V}_1^m - \mathbf{V}_2^m) \mathbf{Y} \cdot \sigma \left( \mathbf{Y}^\top \mathbf{Q}_1^{m\top} \mathbf{K}_1^m \mathbf{Y} \right) \right\|_{\mathrm{F}}$$

$$+ \sum_{m=1}^{M} \left\| \mathbf{V}_2^m \mathbf{Y} \cdot \left( \sigma \left( \mathbf{Y}^\top \mathbf{Q}_1^{m\top} \mathbf{K}_1^m \mathbf{Y} \right) - \sigma \left( \mathbf{Y}^\top \mathbf{Q}_2^{m\top} \mathbf{K}_2^m \mathbf{Y} \right) \right) \right\|_{\mathrm{F}}$$

$$\leq \sum_{m=1}^{M} \|\mathbf{V}_1^m - \mathbf{V}_2^m\|_{\mathrm{F}} \cdot R\sqrt{N} \cdot \sqrt{N} + \sum_{m=1}^{M} BR\sqrt{N} \cdot \|\mathbf{Y}^\top (\mathbf{B}_2^m - \mathbf{B}_1^m) \mathbf{Y}\|_{\mathrm{F}}$$

$$\leq \sum_{m=1}^{M} \|\mathbf{V}_1^m - \mathbf{V}_2^m\|_{\mathrm{F}} \cdot RN + \sum_{m=1}^{M} BR^3 N^{3/2} \cdot \|\mathbf{B}_2^m - \mathbf{B}_1^m\|_{\mathrm{F}}.$$

Here, $\mathbf{B}_1^m := \mathbf{Q}_1^{m\top}\mathbf{K}_1^m$ and $\mathbf{B}_2^m := \mathbf{Q}_2^{m\top}\mathbf{K}_2^m$. Then, $\|\mathbf{B}_1^m\|_F, \|\mathbf{B}_2^m\|_F \le B^2$ holds for $\forall m \in [M]$. It leads to the following upper bound of the covering number of $\mathcal{F}^{\mathtt{Attn}}$.

$$\mathcal{N}(\delta; \mathcal{F}^{\mathtt{Attn}}, \|\cdot\|_{F,\infty}) \le \prod_{m=1}^{M} \mathcal{N}(\delta/2MRN; \mathcal{M}_1, \|\cdot\|_F) \cdot \prod_{m=1}^{M} \mathcal{N}(\delta/2BMR^3N^{3/2}; \mathcal{M}_2, \|\cdot\|_F)$$

where matrix set $\mathcal{M}_1 = \{V \in \mathbb{R}^{D\times D} : \|\mathbf{V}\|_F \le B\}$ and $\mathcal{M}_2 = \{\mathbf{B} \in \mathbb{R}^{D\times D} : \|\mathbf{B}\|_F \le B^2\}$. To sum up, we conclude that:

$$\log \mathcal{N}(\delta; \mathcal{F}^{\mathtt{Attn}}, \|\cdot\|_{F,\infty}) \le D^2 M \cdot \log \frac{6BMRN}{\delta} + D^2 M \cdot \log \frac{6B^2 MR^3 N^{3/2}}{\delta},$$

which comes to our conclusion. $\qquad\square$

Now, we are ready to combine these results together.

**Lemma 22.** Consider the multi-layer transformers class $\mathcal{T}_{\mathrm{raw}}(D, L, M, B)$ without encoders and decoders. Then the log-covering number with input range bounded by $R_0$ ($\|\mathbf{Y}\|_F \le R_0$) can be bounded by

$$\log \mathcal{N}(\delta; \mathcal{T}_{\mathrm{raw}}, R_0, \|\cdot\|_{F,\infty}) \le 4D^2 M \cdot \left(L^2 \log L_{\mathcal{F}} A_{\mathcal{F}} + \log \frac{12R_0 B^2 MLN^{3/2}}{\delta}\right).$$

Here $L_{\mathcal{F}} = 1 + c^2 \vee (BM\sqrt{N} + 2B^3 MR^2 N)$ and $A_{\mathcal{F}} = (1 + B^2 + 2B\sqrt{N}) \vee (1 + MB\sqrt{N})$.

*Proof.* For the score network class $\mathcal{F} = \mathcal{T}_{\mathrm{raw}}(D, L, M, B)$, we have:

$$\mathcal{F} = \{f_{2L} \circ f_{2L-1} \circ \ldots \circ f_1 : f_i \in \mathcal{F}_i \text{ for } \forall i \in [2L], \text{ each } \mathcal{F}_i \text{ is either } \mathcal{F}^{\mathtt{FFN}} \text{ or } \mathcal{F}^{\mathtt{Attn}}\}.$$

Denote $L_{\mathcal{F}} := 1 + B^2 \vee (BM\sqrt{N} + 2B^3 MR^2 N)$ as the upper bound of Lipschitz constant for both $\mathcal{F}^{\mathtt{FFN}}$ and $\mathcal{F}^{\mathtt{Attn}}$. Notice that for any input $Y \in \mathbb{R}^{D\times N}$ such that $\|Y\|_F \le R_0$, then the output of feed-forward network $f \in \mathcal{F}^{\mathtt{FFN}}$ holds

$$\|f(Y)\|_F \le \|Y\|_F + B(B\|Y\|_F + \sqrt{N}) + B\sqrt{N} = (1 + B^2 + 2B\sqrt{N}) \cdot (R_0 \vee 1).$$

The output of attention network $f \in \mathcal{F}^{\mathtt{Attn}}$ holds that

$$\|f(Y)\|_F \le \|Y\|_F + MB\|Y\|_F \cdot \sqrt{N} \le (1 + MB\sqrt{N})R_0.$$

Here we use the fact that $\|P\|_F \le \sqrt{N}$ holds for all probability matrix $P \in \mathbb{R}^{D\times N}$. Denote $A_{\mathcal{F}} := (1 + B^2 + 2B\sqrt{N}) \vee (1 + MB\sqrt{N}) > 1$ as the signal amplifier of each layer, then we can set the sequence of radius as $R_i = (A_{\mathcal{F}})^i \cdot R_0$ with $R_0 > 1$ and $f_i : \mathcal{B}(R_{i-1}) \to \mathcal{B}(R_i)$. According to Lemma 19, we have:

$$\mathcal{N}(\delta; \mathcal{F}, \|\cdot\|_{F,\infty}) \le \prod_{i=1}^{2L} \mathcal{N}(\delta/2LL_F^{2L-i}; \mathcal{F}_i, \|\cdot\|_{F,\infty}),$$

which leads to

$$\log \mathcal{N}(\delta; \mathcal{F}, \|\cdot\|_{F,\infty})$$
$$\le \sum_{i=1}^{2L} \log \mathcal{N}(\delta/2LL_F^{l-i}; \mathcal{F}_i, \|\cdot\|_{F,\infty})$$
$$\le \sum_{i=1}^{2L} \left(4D^2 \log \frac{12B^2(R_{i-1} + \sqrt{N}) \cdot 2LL_F^{2L-i}}{\delta} \vee D^2 M \cdot \log \frac{6B^2 MR_{i-1}^3 N^{3/2} \cdot 2LL_F^{2L-i}}{\delta}\right)$$
$$\le 8D^2 M \cdot \left(L^2 \log L_{\mathcal{F}} A_{\mathcal{F}} + \log \frac{24R_0 B^2 MLN^{3/2}}{\delta}\right).$$

The proof is complete. $\qquad\square$

Moreover, if we consider the entire transformers architecture with the encoder and decoder layers, we directly have the following results:

**Lemma 23** (Covering number of transformers). Consider the entire transformer architecture $\mathcal{F} = \mathcal{T}(D, L, M, B, R)$ in which the encoder that takes $\mathbf{v}_t \in \mathbb{R}^{Nd}$ as the input and embeds it with the time embedding $\mathbf{e}$ and the diffusion time-step embedding $\boldsymbol{\phi}(t)$, where $\|\mathbf{e}\|_2 = r$ and $\|\boldsymbol{\phi}(t)\|_2 \leq C_{\text{diff}}$ for some absolute constant $C_{\text{diff}} > 0$. Then the log-covering number with initial input range $\|\mathbf{v}_t\|_2 \leq R_0$ is bounded by

$$\log \mathcal{N}(\delta; \mathcal{F}, R_0, \|\cdot\|_{\text{F},\infty}) \leq 8D^2 M \cdot \left( L^2 \log L_{\mathcal{F}} A_{\mathcal{F}} + \log \frac{24 R_1 B^2 M L N^{3/2}}{\delta} \right).$$

Here $R_1 = R_0 + \sqrt{N}(r + C_{\text{diff}})$. This is because the encoder maps $\mathbf{v}_t$ to $\mathbf{Y} \in \mathbb{R}^{D \times N}$ with $\|\mathbf{Y}\|_{\text{F}} \leq \|\mathbf{v}_t\|_2 + r\sqrt{N} + C_{\text{diff}}\sqrt{N}$, and our decoder (defined in Appendix D.3.1) only extracts part of the entries from the output and impose a clipping function with range being $R$ on it, both of which lead to no increase in the covering number.

# F  EXPERIMENTAL DETAILS IN SECTION 6

**Patch Embedding Modification**    To implement our simulation experiments on Gaussian Process data, we slightly adapt the original DiT designed for image generation (Peebles and Xie, 2023). In the original DiT, a pre-trained VAE encoder is deployed to convert image in the training dataset to feature representations and patch embedding is applied by splitting the image into subblocks(patches) and flattening and projecting each subblock to a feature. But in our numeric experiments, we dropped the VAE encoder and split the data at time dimension, with each time step being a patch, and project the patch into feature of higher dimension.

**Kernel Estimation Method**    In our experiment, each sample generated by DiT in is denoted as $\mathbf{S} = [\mathbf{s}_1, ..., \mathbf{s}_N] \in \mathbb{R}^{N \times d}$, where $\mathbf{s}_i = \mathbf{S}[i] \in \mathbb{R}^d$ denotes the data patch at $i$-th time index. In our experiment we take $N = 128$ and $d = 8$ and with $n \in \{10^3, 3200, 10^4, 32000, 10^5\}$ samples in total. To evaluate the quality of generated data, we calculate the mean of data patch at $i$-th time step and covariance matrix between data patch at $i$-th and $j$-th time step as follows and compare them with the theoretical ones.

$$\widehat{\boldsymbol{\mu}}_i = \frac{1}{n} \sum_{k=1}^n \mathbf{S}_k[i], \quad \widehat{\boldsymbol{\Sigma}}_{i,j} = \frac{1}{n} \sum_{k=1}^n (\mathbf{S}_k[i] - \widehat{\boldsymbol{\mu}}_i)(\mathbf{S}_k[j] - \widehat{\boldsymbol{\mu}}_j)^\top.$$

Here $\widehat{\boldsymbol{\mu}}_i \in \mathbb{R}^d$ is the empirical mean of the $i$-th patch and $\widehat{\boldsymbol{\Sigma}}_{i,j}$ is the empirical covariance matrix between the $i$-th and the $j$-th patch. With $\widehat{\boldsymbol{\Sigma}}_{i,j}$, we could estimate the empirical kernel value $\widehat{\gamma}(i,j)$ as

$$\widehat{\gamma}(i,j) = \underset{\alpha}{\operatorname{argmin}} \left\| \widehat{\boldsymbol{\Sigma}}_{i,j} - \alpha \boldsymbol{\Sigma} \right\|_{\text{F}}.$$

where $\boldsymbol{\Sigma} \in \mathbb{R}^{d \times d}$ is the true covariance matrix. After running on each pair of $(i,j) \in \{1, 2, ..., N\}^2$, we can get the empirical kernel $\widehat{\boldsymbol{\Gamma}} = [\widehat{\gamma}(i,j)]_{ij} \in \mathbb{R}^{N \times N}$.

**Relative Error**    In the experiments where we compare sample efficiency across different training data size $n$ and kernel setting $\nu, \ell$, the metric is a relative error of the estimated sample covariance matrix to its ground-truth:

$$\epsilon = \frac{\left\| \widehat{\boldsymbol{\Gamma}} \otimes \widehat{\boldsymbol{\Sigma}} - \boldsymbol{\Gamma} \otimes \boldsymbol{\Sigma} \right\|_{\text{F}}^2}{\left\| (\widehat{\boldsymbol{\Gamma}} \otimes \widehat{\boldsymbol{\Sigma}})_{\text{truth}} - \boldsymbol{\Gamma} \otimes \boldsymbol{\Sigma} \right\|_{\text{F}}^2}.$$

Here $(\widehat{\boldsymbol{\Gamma}} \otimes \widehat{\boldsymbol{\Sigma}})_{\text{truth}}$ is the estimated covariance matrix through the same method but under an equal amount of training data, instead of generated data. This relative error eliminates the influence of different scaling of $\boldsymbol{\Gamma} \otimes \boldsymbol{\Sigma}$ and the concentration error caused by finite samples.

**Query-Key Matrices and Value Matrices**     From Figure  10, we can see that the weights of the query-key matrix $\mathbf{Q}^{\top}\mathbf{K}$ emphasize the time embedding part of the input across different layers. Interestingly, Figure 11 shows that the weight emphasis on the value matrix $\mathbf{V}$ is on the data patch part instead. This observation aligns with our theoretical construction of the score function using the DiT structure. Specifically, the score matrix (determined by the query-key matrix) captures the kernel $\boldsymbol{\Gamma}$, representing temporal dependency, while the value matrix determines the covariance $\boldsymbol{\Sigma}$, representing spatial dependency.

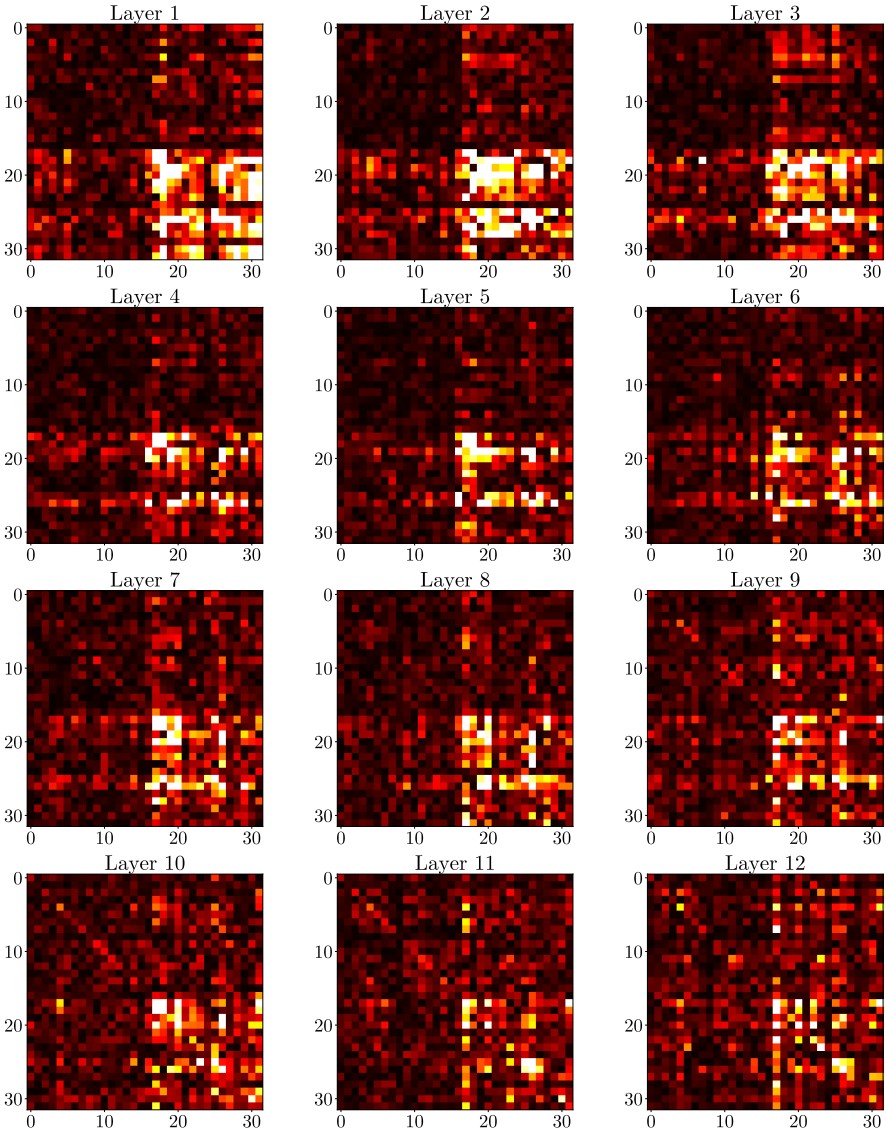

Figure 10: *Query-Key matrices in different transformer attention layers(1~12).*

**Attention Score in Different Attention Layers**     To further demonstrate our theory, we visualize the attention score $\mathbf{Y}_t^{\top}\mathbf{Q}^{\top}\mathbf{K}\mathbf{Y}_t \in \mathbb{R}^{N \times N}$ averaged over $n = 10^5$ data points at different backward diffusion times $t \in \{0, 50, 100, 200, 400, 800, 1000\}$ to observe if it gradually unveils the kernel $\boldsymbol{\Gamma}$ as the backward process progresses. According to Figure 12, in the initial few layers (1~2), we observe the early stages of kernel construction. In the subsequent layers (3~12), the attention score matrix increasingly resembles the kernel, becoming clearer as the backward diffusion process advances.

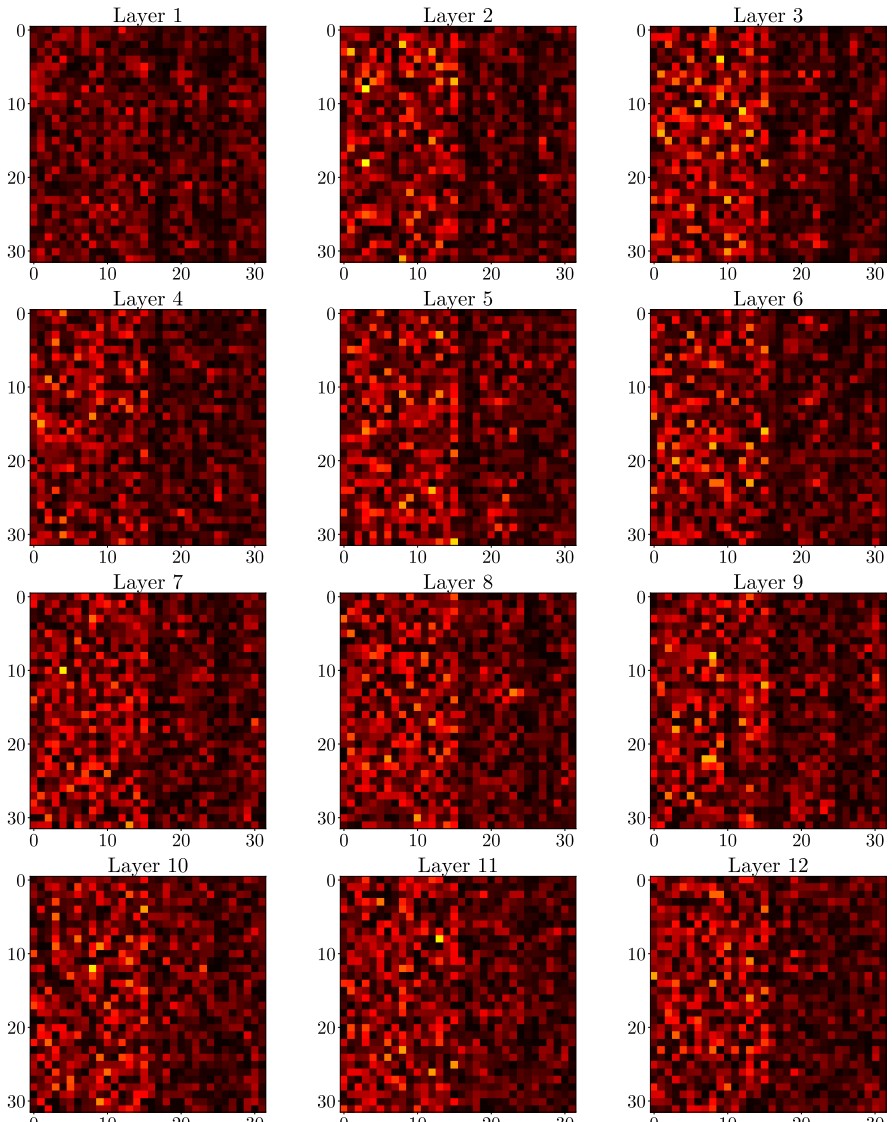

Figure 11: *Value weight matrices* **V** *in different transformer attention layers(1~12).*

**Generated Motions of 2D Balls**    We collect a latent sample from the diffusion transformer and map it to the original 2D space through the pretrained decoder of the VAE, forming a 240-frame video. We present 36 frames in Figure 13 from the video, which are chosen uniformly out of the 240 frames. As shown in the figure, the motion of the ball shows great time consistency and accurately captures the bouncing-back mechanism as expected.

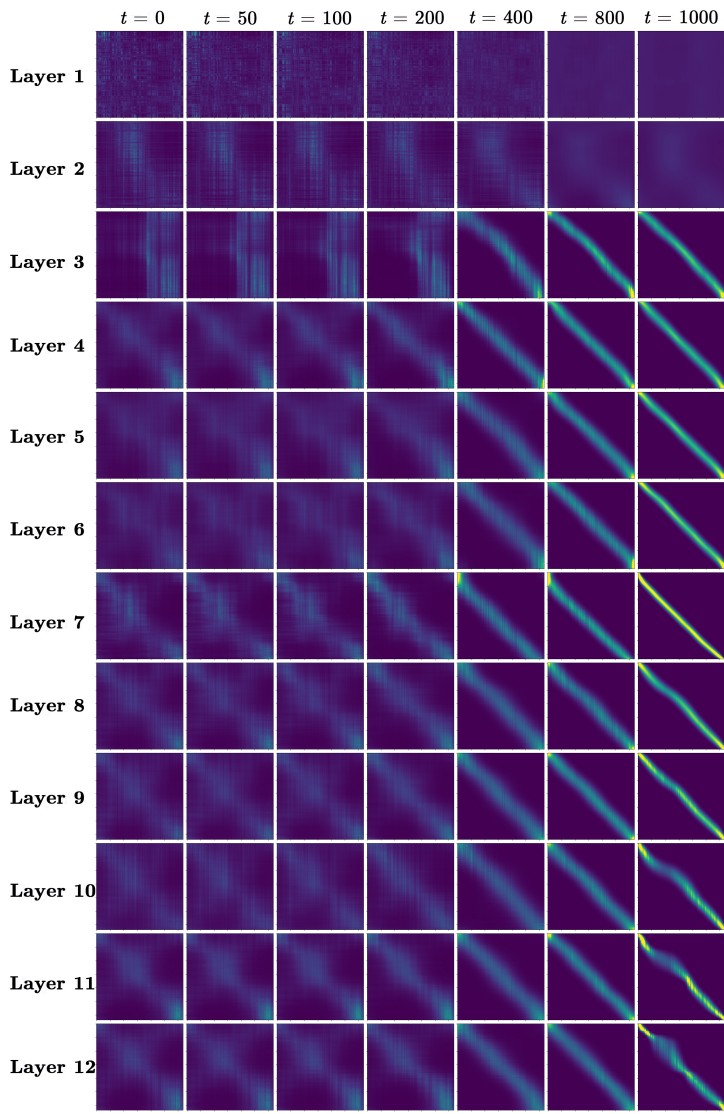

Figure 12: *Attention score matrices in different attention layers and at different steps of the backward process.*

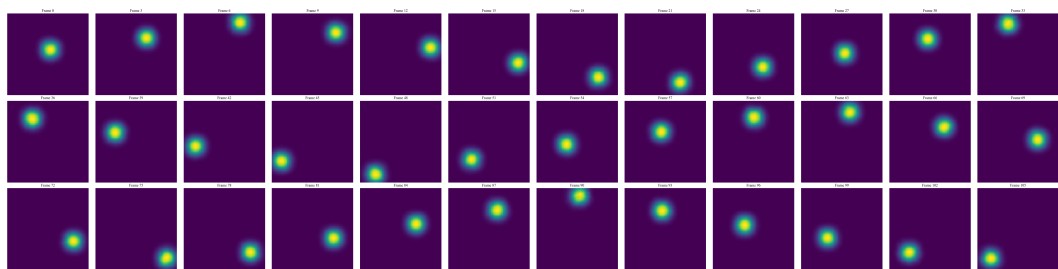

Figure 13: Consecutive frames of a video generated by a trained diffusion transformer with a 2D VAE.

## G  SUPPORTING TECHNICAL RESULTS

### G.1  GAUSSIAN LEMMAS

In this subsection, we will introduce several Lemmas to control the deviation of random variables which polynomially depend on some Gaussian random variables. We will use a slightly modified version of Lemma 30 from Damian et al. (2022).

**Lemma 24.** Let $g$ be a polynomial of degree $p$ and $x \sim \mathcal{N}(0, I_d)$. Then there exists an absolute positive constant $C_p$ depending only on $p$ such that for any $\delta > 1$,

$$\mathbb{P}\left[|g(x) - \mathbb{E}[g(x)]| \geq \delta\sqrt{\mathrm{Var}(g(x))}\right] \leq 2\exp\left(-C_p \delta^{2/p}\right).$$

The next Lemma orginates from Theorem 4.3, Prato and Tubaro (2007).

**Lemma 25.** For any $\ell \in \mathbb{N}$ and $f \in L^2(\mathcal{N}(0, \boldsymbol{I}_d))$ to be a degree $\ell$ polynomial, for any $q \geq 2$, we have

$$\mathbb{E}_{z \sim \gamma}\left[f(z)^q\right] \leq C_{q,\ell}\left(\mathbb{E}_{z \sim \gamma}\left[f(z)^2\right]\right)^{q/2}.$$

where we use $C_{q,\ell}$ to denote some universal constant that only depends on $q, \ell$.

### G.2  LIPSCHITZ CONTINUITY OF ACTIVATION FUNCTIONS

**Lemma 26** (1-Lipschitz continuity of softmax function)**.** For the softmax function $\sigma : \mathbb{R}^d \to \mathbb{R}^d$, it is 1-Lipschitz continuous under $l_2$ norm.

*Proof.* For any $x \in \mathbb{R}^d$, denote $J(x)$ as the Jacobian matrix of softmax function at $x$. Then, the Lipschitz continuity of $\sigma(\cdot)$ under $l_2$ norm can be upper bound by $\sup_{x \in \mathbb{R}^d} \|J(x)\|_{\mathrm{F}}$. By calculation, we have

$$J(x) = \mathrm{Diag}\left(\sigma(x)\right) - \sigma(x)\sigma(x)^\top.$$

Let $\sigma(x) = (p_1, p_2, \ldots, p_d)^\top$ which is a probability vector, then

$$\|J(x)\|_{\mathrm{F}}^2 \leq \sum_{i=1}^{d} p_i^2(1 - p_i)^2 + \sum_{i \neq j}(p_i p_j)^2 \leq \left(\sum_{i=1}^{d} p_i^2\right)^2 \leq \left(\sum_{i=1}^{d} p_i\right)^4 = 1.$$

Therefore $\sup_{x \in \mathbb{R}^d} \|J(x)\|_{\mathrm{F}} \leq 1$, which comes to our conclusion. A direct extension is that the column-wise softmax over matrices is 1-Lipschitz continuous under $\|\cdot\|_{\mathrm{F}}$ norm. $\square$

### G.3  BASICS ON RESNET UNIVERSAL APPROXIMATION THEORY

In this section, we briefly introduce universal approximation theory of ResNet. An $L$-layer ResNet $\mathbf{R}(\mathbf{x}) : \mathbb{R}^d \to \mathbb{R}^{d_o}$ can be defined as

$$\mathbf{R}(\mathbf{x}) = \mathcal{B} \circ \mathrm{FFN}_L \circ \mathrm{FFN}_{L-1} \circ \cdots \circ \mathrm{FFN}_1 \circ \mathcal{A}(\mathbf{x}). \tag{22}$$

Here $\mathcal{A} : \mathbb{R}^d \to \mathbb{R}^{d'}$ and $\mathcal{B} : \mathbb{R}^{d'} \to \mathbb{R}^{d_o}$ are two linear transformations, and $\mathrm{FFN}_i : \mathbb{R}^{d'} \to \mathbb{R}^{d'}$ are basic residual blocks defined as $\mathrm{FFN}_i(\mathbf{y}) = \mathbf{y} + \mathbf{W}_{2,i} \cdot \mathrm{ReLU}(\mathbf{W}_{1,i}\mathbf{y} + \mathbf{b}_{2,i}) + \mathbf{b}_{1,i}$ with $\mathbf{W}_{1,i} \in \mathbb{R}^{d_i \times d'}$, $\mathbf{W}_{2,i} \in \mathbb{R}^{d' \times d_i}$, $\mathbf{b}_{1,i} \in \mathbb{R}^{d'}$ and $\mathbf{b}_{2,i} \in \mathbb{R}^{d_i}$. We denote by $\mathcal{RN}(d, d_o, d', W, L, S, C)$ the set of ResNet functions from $\mathbb{R}^d$ to $\mathbb{R}^{d_o}$ with $L$ layers, $d'$ neurons in each identity layer, maximum width $W = \max_{i \in [L]}\{d_i\}$ and nonzero weights $S$. Moreover, the Frobenius norm of the weight matrices $\mathbf{W}_{j,i}$ and the Euclidean norm of the bias vectors $\boldsymbol{b}_{j,i}$ are uniformly bounded by $C > 0$.

The next lemma shows that we can approximate product operation with ResNet.

**Lemma 27** (Proposition 12 in Liu et al. (2024a))**.** Let $x, y \in [-B, B]$ with $B \geq 1$. Then there exists a ResNet $R \in \mathcal{RN}(2, 1, 3, 4, \mathcal{O}(\log(B/\epsilon), \mathcal{O}(\log(B/\epsilon), \mathcal{O}(B))$ such that

$$|R(x, y) - xy| \leq \epsilon, \quad x, y \in [-B, B]^d \tag{23}$$

holds.

The construction process follows the standard techniques proposed by Yarotsky (2018), which first use feed-forward networks to approximate the square function $f_{\mathrm{sq}}(x) \approx x^2$ and transfer it into a product operator $\times(x, y) = (f_{\mathrm{sq}}(x + y) - f_{\mathrm{sq}}(x - y))/4 \approx xy$.

Furthermore, for $\mathbf{x} \in \mathbb{R}^d$ and $y \in \mathbb{R}$, we can approxiamtely construct the following mapping:

$$f([\mathbf{x}^\top, y]^\top) = y\mathbf{x}. \tag{24}$$

**Corollary 2.** Given $\epsilon > 0$, there exists a ResNet $f_{\text{mult}} \in \mathcal{RN}(d + 1, d, 3d, 4d, \mathcal{O}(\log(B/\epsilon)), d\mathcal{O}(\log(B/\epsilon)), \mathcal{O}(Bd))$ such that $f_{\text{mult}}(y, \mathbf{x}) = y\mathbf{x} + \boldsymbol{\epsilon}$, where $\|\boldsymbol{\epsilon}\|_\infty \leq \epsilon$.

*Proof.* By Lemma 27, we know there exists a ResNet $R : \mathbb{R}^2 \to \mathbb{R}$ satisfies

$$R \in \mathcal{RN}(2, 1, 3, 4, \mathcal{O}(\log(B/\epsilon)), \mathcal{O}(\log(B/\epsilon)), \mathcal{O}(1))$$

and

$$|R(x, y) - xy| \leq \epsilon, \ \ x, y \in [-B, B].$$

Then let's consider the following two steps of mapping:

$$\mathcal{A}([\mathbf{x}^\top, y]^\top) = [x_1, y, x_2, y, \cdots, x_d, y]^\top,$$
$$\mathcal{B}([x_1, y, x_2, y, \cdots, x_d, y]^\top) = [R(x_1, y), R(x_1, y), \cdots, R(x_d, y)]^\top.$$

Here we could choose $\mathcal{A}$ to be a $(d + 1) \times 2d$ matrix

$$\mathcal{A} = \begin{bmatrix} 1 & 0 & \cdots & 0 & 0 \\ 0 & 0 & \cdots & 0 & 1 \\ 0 & 1 & \cdots & 0 & 0 \\ 0 & 0 & \cdots & 0 & 1 \\ \vdots & & \ddots & & \vdots \\ 0 & 0 & \cdots & 1 & 0 \\ 0 & 0 & \cdots & 0 & 1 \end{bmatrix},$$

and take $\mathcal{B}$ as parallelization of $d$ homogeneous networks $R$. Let $f_{\text{mult}} = \mathcal{B} \circ \mathcal{A}$, we know $f_{\text{mult}} \in \mathcal{RN}(d+1, d, 3d, 4d, \mathcal{O}(\log(B/\epsilon)), d\mathcal{O}(\log(B/\epsilon)), \mathcal{O}(d))$, and it approximately realizes the mapping (24) with $L_\infty$ approximation error being $\epsilon$ for the first $d$ entries and no error for the last entry. $\square$

More generally, if we consider the input to be $[x^\top, y, \mathbf{z}]^\top$ and want to construct a (FFN-only) transformers that approximately maps the input to $[yx^\top, y, \mathbf{z}]^\top$, we have the following results:

**Corollary 3.** Suppose the input to be $\mathbf{Y} = [\mathbf{y}_1, \mathbf{y}_2, \ldots, \mathbf{y}_N] \in \mathbb{R}^{D \times N}$ with $\mathbf{y}_i = [\mathbf{x}_i^\top, \mathbf{0}_{2d}^\top, w_i, \mathbf{z}_i^\top]$, where $\mathbf{x}_i \in [-B, B]^d$, $w_i \in [-B, B]$ and $\mathbf{z}_i \in \mathbb{R}^{d_z}$. Given any $\epsilon > 0$, there exists a (FFN-only) transformers

$$\mathbf{f}_{\text{mult}} = \text{FFN}_L \circ \text{FFN}_{L-1} \circ \cdots \circ \text{FFN}_1$$

with $L = \mathcal{O}(\log(B/\epsilon))$ layers that approximately multiplies each component $\mathbf{x}_i$ with the weight $w_i$, which keeping other dimensions the same. This can be formally written as

$$\mathbf{f}_{\text{mult}}(\mathbf{Y}) = \begin{bmatrix} f_{\text{mult}}(w_1, \mathbf{x}_1) & \cdots & f_{\text{mult}}(w_N, \mathbf{x}_N) \\ \mathbf{0}_{2d} & \cdots & \mathbf{0}_{2d} \\ w_1 & \cdots & w_N \\ \mathbf{z}_1 & \cdots & \mathbf{z}_N \end{bmatrix}, \ \text{where} \ \ \|f_{\text{mult}}(w_i, \mathbf{x}_i) - w_i\mathbf{x}_i\|_\infty \leq \epsilon.$$

The inner dimension of the FFNs is at most $8d$. Moreover, the number of nonzero coefficients in each weight matrices or bias vectors is at most $\mathcal{O}(d)$, and the norm of the matrices and bias are all bounded by $\mathcal{O}(Bd)$.

Here we require the buffer variables $\mathbf{0}_{2d}$ in the input because Corollary 2 needs $3d$ neurons in each dimension to store the calculation results that are necessary for constructing the product function. Thus, we add $\mathbf{0}_{2d}$ so that $[\mathbf{x}^\top, \mathbf{0}_{2d}^\top] \in \mathbb{R}^{3d}$ for construction convenience.

*Proof of Corollary 3.* By Corollary 2, there exists a ResNet $f_{\text{mult}} : \mathbb{R}^{d+1} \to \mathbb{R}^d$ such that

$$f_{\text{mult}}(y, \mathbf{x}) = \mathcal{B} \circ \text{FFN}_L \circ \text{FFN}_{L-1} \circ \cdots \circ \text{FFN}_1 \circ \mathcal{A}(\mathbf{x}, y)$$

with $f_{\text{mult}}(y, \mathbf{x}) = y\mathbf{x} + \boldsymbol{\epsilon}$, where $\|\boldsymbol{\epsilon}\|_\infty \leq \epsilon$. Since the FFNs in the transformers are position-wise, we only need to consider the mapping of one token while others are completely the same. For each $i$, suppose $\text{FFN}_i : \mathbb{R}^{3d} \to \mathbb{R}^{3d}$ satisfies

$$\text{FFN}_i(\mathbf{v}) = \mathbf{v} + \mathbf{W}_{2,i} \cdot \text{ReLU}(\mathbf{W}_{1,i}\mathbf{v} + \mathbf{b}_{2,i}) + \mathbf{b}_{1,i}$$

with $\mathbf{W}_{1,i} \in \mathbb{R}^{d_i \times 3d}$, $\mathbf{W}_{2,i} \in \mathbb{R}^{3d \times d_i}$, $\mathbf{b}_{1,i} \in \mathbb{R}^{3d}$ and $\mathbf{b}_{2,i} \in \mathbb{R}^{d_i}$. Here $d_i \leq 4d$ for all $1 \leq i \leq L$. For the first FFN, let

$$
\text{FFN}'_1 \left( \begin{bmatrix} \mathbf{x} \\ \mathbf{0}_{2d} \\ w \\ \mathbf{z} \end{bmatrix} \right) = \begin{bmatrix} \mathbf{x} \\ \mathbf{0}_{2d} \\ w \\ \mathbf{z} \end{bmatrix} + \begin{bmatrix} \mathbf{W}_{2,1} \\ \mathbf{0}_{d_z \times d_i} \end{bmatrix} \cdot \text{ReLU} \left( \begin{bmatrix} \mathbf{W}_{1,1} \mathcal{A}, \mathbf{0}_{d_i \times (d_z + 2d + 1)} \end{bmatrix} \begin{bmatrix} \mathbf{x} \\ w \\ \mathbf{0}_{2d} \\ \mathbf{z} \end{bmatrix} + \mathbf{b}_{2,1} \right) + \begin{bmatrix} \mathbf{b}_{1,1} \\ \mathbf{0}_{d_z} \end{bmatrix}
$$

$$
= \begin{bmatrix} \text{FFN}_1 \circ \mathcal{A}(\mathbf{x}, y) \\ w \\ \mathbf{z} \end{bmatrix}.
$$

For $2 \leq i \leq L$, suppose the input is $[\mathbf{v}^\top, w, \mathbf{z}^\top]^\top = [\mathbf{v}^\top, w, \mathbf{z}^\top]^\top$ with $\mathbf{v} \in \mathbb{R}^{3d}$ and $\mathbf{z} \in \mathbb{R}^{d_z}$, let

$$
\text{FFN}'_i \left( \begin{bmatrix} \mathbf{v} \\ w \\ \mathbf{z} \end{bmatrix} \right) = \begin{bmatrix} \mathbf{v} \\ w \\ \mathbf{z} \end{bmatrix} + \begin{bmatrix} \mathbf{W}_{2,i} \\ \mathbf{0}_{d'_z \times d_i} \end{bmatrix} \cdot \text{ReLU} \left( \begin{bmatrix} \mathbf{W}_{1,i}, \mathbf{0}_{d_i \times d'_z} \end{bmatrix} \begin{bmatrix} \mathbf{v} \\ w \\ \mathbf{z} \end{bmatrix} + \mathbf{b}_{2,i} \right) + \begin{bmatrix} \mathbf{b}_{1,i} \\ \mathbf{0}_{d'_z} \end{bmatrix} = \begin{bmatrix} \text{FFN}_i(\mathbf{v}) \\ \mathbf{z} \end{bmatrix}.
$$

Here $d'_z = d_z + 1$. For the final layer, let

$$
\text{FFN}'_{L+1} \left( \begin{bmatrix} \mathbf{v} \\ w \\ \mathbf{z} \end{bmatrix} \right) = \begin{bmatrix} \mathbf{v} \\ w \\ \mathbf{z} \end{bmatrix} + \boldsymbol{W}_{2,L+1} \cdot \text{ReLU} \left( \begin{bmatrix} \mathcal{B} & \mathbf{0}_{d \times d'_z} \\ -\mathcal{B} & \mathbf{0}_{d \times d'_z} \\ \mathbf{I}_{3d} & \mathbf{0}_{d \times d'_z} \\ -\mathbf{I}_{3d} & \mathbf{0}_{d \times d'_z} \end{bmatrix} \begin{bmatrix} \mathbf{v} \\ w \\ \mathbf{z} \end{bmatrix} \right) = \begin{bmatrix} \mathcal{B}\mathbf{v} \\ \mathbf{0}_{2d} \\ w \\ \mathbf{z} \end{bmatrix}.
$$

Here

$$
\boldsymbol{W}_{2,L+1} = \begin{bmatrix} \text{diag}(\mathbf{1}_d, \mathbf{0}_{2d}) & -\text{diag}(\mathbf{1}_d, \mathbf{0}_{2d}) & -\mathbf{I}_{3d} & \mathbf{I}_{3d} \\ \mathbf{0}_{(2d+d'_z) \times d} & \mathbf{0}_{(2d+d'_z) \times d} & \mathbf{0}_{(2d+d'_z) \times 3d} & \mathbf{0}_{(2d+d'_z) \times 3d} \end{bmatrix}.
$$

Thus, we have finished constructing the (FFN-only) transformers we want by taking

$$
\mathbf{f}_{\texttt{mult}} = \text{FFN}'_{L+1} \circ \text{FFN}_{L-1} \circ \cdots \circ \text{FFN}'_1,
$$

and the hidden dimension of the FFNs is at most $8d$. Moreover, by the definition of the original $\text{FFN}_i$ and the new $\text{FFN}'_i$, the number of nonzero coefficients in each weight matrix or bias vector is at most $\mathcal{O}(d)$, and the norm of the matrices and bias are all bounded by $\mathcal{O}(d)$. The proof is complete. $\qquad\square$

### G.4 ASYMPTOTIC RESULTS ON THE SPECTRUM OF TOEPLITZ MATRICES

In this section, we provide some existing results on the spectrum of Toeplitz matrix when both its size and bandwidth go to infinity. For a Toeplitz matrix $\mathbf{T} \in \mathbb{R}^{N \times N}$, its $(i, j)$-th component $\mathbf{T}_{ij} = a_{|i-j|}$ only depends on its distance to diagonal. When $k > M$, we have $a_k = 0$ where $M$ is known as the bandwidth. Denote $\lambda_1, \lambda_2, \ldots, \lambda_N$ to be the eigenvalues of $\mathbf{T}$, with multiplicities counted and let $\mu_N := \frac{1}{N} \sum_{i=1}^N \delta_{\lambda_i}$ be the empirical distribution of the spectrum. In the asymptotic case where $M, N \to \infty$, we focus on the behavior of $\mu_N$. Denote $F_N(x)$ to be the cumulative distribution function of $\mu_N$, Kargin (2009) proposes the result that $F_N(x)$ converges to the standard Gaussian distribution when the Toeplitz matrix follows that $\mathbb{E}a_k = 0$, $\mathbb{E}a_k^2 = \frac{1}{M}$, $\sup_{k,N} \mathbb{E}|\sqrt{M}a_k|^4 < C < \infty$ and most importantly, the band-to-size ratio $\frac{M}{N} \to 0$. While the band-to-size ratio $\frac{M}{N} \to c \in (0, 1)$, the spectrum distribution $F_N(x)$ converges to some non-Gaussian distribution $\Psi_c$. Some other statistical works such as Hartman and Wintner (1950); Tilli (1998); Delsarte and Genin (2005) study the spectrum of generalized Toeplitz matrices by using Fourier expansion.

In our case, the condition number $\kappa_{t_0}$ will keep in a constant range if the sequence $\{a_k\}$ introduced above sharply decays, while grow as $N$ goes larger if $\{a_k\}$ slowly decays. In the former case, we can treat $\kappa_{t_0}$ as a constant, which does not affect our analysis. In the latter case, since we have a natural upper bound of

$$
\kappa_{t_0} \leq \sigma_{t_0}^{-2} \lambda_{\max}(\mathbf{\Gamma} \otimes \mathbf{\Sigma}) \lesssim \ell \sigma_{t_0}^{-2} \lesssim t_0^{-1},
$$

by taking $t_0 = n^{-1/3}$ in Theorem 2, we can still obtain a $n^{-1/6}$-convergence rate in both $W_2$ distance and TV distance.

