# OpenReview forum: "Diffusion Transformer Captures Spatial-Temporal Dependencies: A Theory for Gaussian Process Data"
_ICLR.cc/2025/Conference — ICLR 2025 Poster_

### Official Review · Reviewer_FvaY · 2024-10-20

**Soundness:** 3
**Presentation:** 3
**Contribution:** 4
**Rating:** 8
**Confidence:** 3

**Summary:**

In this paper, authors explore the reasons for Diffusion Transformer (DiT)’s capability of capturing spatial-temporal dependencies within the sequential data. They theoretically establish score approximation and distribution estimation guarantee of DiT for learning Gaussian process data. Specifically, they replace the score function in diffusion models with gradient descent process defined by truncated covariance, which are further implemented with transformer models. Authors conduct some experiments to prove the effectiveness of their theory under their assumptions.

**Strengths:**

1. This paper studies the principle behind the Diffusion Transformer and provides a theoretical explanation of its capability of modeling spatial-temporal correlations within data, which is the very first trial in related fields.
2. Authors propose a novel score function approximation theory as well as the sample complexity bound. All the lemmas and theorems are well defined with sufficient mathematical proofs.
3. The correctness of the proposed theory has been evaluated on both synthesized and real data through experiments.

**Weaknesses:**

1. Although the overall writing is relatively clear, there’re some suggestions:
1.a) The object of this paper is sequential data, which has the axis of time (real world time, $h$). while in the diffusion process, there’s another axis of diffusion step ($t$), and authors also name it as ‘time’. To avoid confusion, authors should use some other descriptions to make it clearer.
1.b) The symbols should be consistent. E.g., the definitions of $v_t$, $\mathcal{D}$ and $v_0^t$, the use of subscripts of data $\mathbf{x}$ are confusing.
1.c) The font size of Figures could be larger, e.g. Fig.2, otherwise the equations and words are difficult to recognize under normal physical paper size.
2. Although the effectiveness of the theory has been proved on synthesized Gaussian process data and semi-synthetic video data. More real data with complex contents should also be tested on.

**Questions:**

1. The theorems proposed in the paper are based on the authors’ assumptions, then:
1.a) Does it mean that the theory is only valid under these assumptions, or does the model only work on the data satisfying these assumptions?
1.b) How to ensure that the assumptions can represent the case in real sequential data?
1.c) How do authors derive these assumptions? E.g., in Assumption 1, why the definition of the covariance function has that formula, are there any theoretical basis for it?

---

> ### Author Response · Authors · 2024-11-22
> **Response to Reviewer FvaY**
>
> We thank the reviewer for the valuable comments and suggestions. We would appreciate it if you could champion our paper during the discussions!
>
> >Although the overall writing is relatively clear, there’re some suggestions: 1.a) The object of this paper is sequential data, which has the axis of time (real world time, h). while in the diffusion process, there’s another axis of diffusion step (t), and authors also name it as ‘time’. To avoid confusion, authors should use some other descriptions to make it clearer. 1.b) The symbols should be consistent. E.g., the definitions of vt, D and v0t the use of subscripts of data x are confusing. 1.c) The font size of Figures could be larger, e.g. Fig.2, otherwise the equations and words are difficult to recognize under normal physical paper size.
>
> We thank the reviewer for the advice on the meaning of “time” and “diffusion timesteps”, notation consistency and font size. We’ve changed all the “time” related to diffusion process to “diffusion timesteps” (always denoted as $t$, $t_0$ or $T$), and we refer the all the “time”s that appear in “time index” or “time embedding” to Gaussian Process (always denoted as numbers from $1$ to $N$ or $h_1,h_2,\dots,h_N$). We have also adjusted the font size of the figures for a better readability.
>
> >Although the effectiveness of the theory has been proved on synthesized Gaussian process data and semi-synthetic video data. More real data with complex contents should also be tested on.
>
> We agree that experiments on real data offer stronger evidence of the effectiveness of our theory, the scale of real-data experiments like video generation is too large and beyond the scope of a theory-based work. The success of DiT in sequential data (like video data) has been empirically verified by a large number of works, such as SORA, openSora, kling AI, e.t.c.  It's worth mentioning that we've conducted new experiments comparing DiTs with Unets and examining the training dynamics of DiTs, which is shown in the  **general response** and **Appendix A** for the reviewer's interest.
>
> >The theorems proposed in the paper are based on the authors’ assumptions, then: 1.a) Does it mean that the theory is only valid under these assumptions, or does the model only work on the data satisfying these assumptions? b) How to ensure that the assumptions can represent the case in real sequential data?
>
> We believe our theoretical framework is applicable to a broad class of data distribution. Gaussian Process is one of the most common ways of modeling sequential data. Please see our **general response** for a detailed discussion on the assumptions.
>
> Our theory can be applied well **beyond our assumptions and simplification**. Indeed, our results suggest that as long as the score function can be represented by an efficient algorithm and transformers can implement the algorithm, then it is expected that DiT will capture the sequential distribution well. We believe this analytical framework can be applied to nonstationary Gaussian process and broad sequential data distributions. However, rigorous theoretical analysis requires further investigation.
>
> > How do authors derive these assumptions? E.g., in Assumption 1, why the definition of the covariance function has that formula, are there any theoretical basis for it?
>
> The covariance function is derived from the definition of the stationary Gaussian process, which has been widely assumed in statistics literature when studying the inference and prediction of sequential data . The structure of the covariance matrix is required for our approximation theory (Theorem 1), but not the core intuition of our proof, i.e, algorithm unrolling. Please also see our **general response** for a detailed discussion about the assumptions of the Gaussian process.

---

> > ### Comment · Reviewer_FvaY · 2024-11-24
> >
> > Thank you for the feedback. The additional information has addressed most of my concerns. I believe after careful modifications as you promised, the paper will be better and clearer. Therefore, I think this paper is above the acceptance bar of the conference and I will keep my rating. Good luck to you.

---

### Official Review · Reviewer_zotT · 2024-10-30

**Soundness:** 2
**Presentation:** 3
**Contribution:** 3
**Rating:** 6
**Confidence:** 2

**Summary:**

This paper provides a theoretical study on the limits of the learning capabilities of transformer-based diffusion models on learning spatial-temporal data. The authors achieve this by reformulating the task of learning the score function of Gaussian process data as a gradient descent algorithm. This algorithm can then be unrolled using a transformer archtitecture where the attention layers can be studied to evaluate how spatial-temporal dependencies are learned.
The results are thoroughly motivated and dervied and validated using a few numerical examples.

I believe that the theoretical results could have an impact in terms of understanding the learning of temporal dependencies but currently the practical results are not extensive and the take away messages are not clear enough

**Strengths:**

- studying gaussian processes to asses spatio-temporal dependencies is inherently reasonable and could be used as an important theoretical framework to study neural architectures.

- The method mathematically is well grounded and derived

- High-quality figures visualizing key components of the contribution

**Weaknesses:**

- While the results convincingly show that transformers can learn spatial-temporal dependencies, the experiments only show examples of properly learned tasks. Given that theoretical boundaries were derived, I would have loved more experiments investigating these limits. For example something like Figure 1 but with a degradation for higher length. The same goes for Section 6.2. What I am missing is a clear connection between model complexity and performance.

- Lack of comparison to other methods: I think it would be helpful to see how typical diffusion models, including Unet-based architectures, perform on the task of learning Gaussian processes. Clear results could help in shifting the design towards transformer-based diffusion architectures.

- Generalization to real-world problems: There are no experiments showing how well the insights generated by the experiments translate to real-world problems. You present the ball experiment but even there you have to consider how well your experiments align with the assumption of using Gaussian process data (cmp. l. 504-506). Real-world problems will often not follow Gaussian process dynamics

- It would be nice if you could state some take-home messages for the reading. Ideally some general information such as design choices for the model architecture.

**Questions:**

- The learned temporal dependencies presented in Figure 5 look very similar to what you get when you just plot the attention between two uncorrelated matrices with positional embedding (l. 300). What is the difference between these maps and the ones you present, e.g., in Fig. 5

---

> ### Author Response · Authors · 2024-11-22
> **Response to Reviewer zotT**
>
> We thank the reviewer for the valuable comments and suggestions.
>
> >While the results convincingly show that transformers can learn spatial-temporal dependencies, the experiments only show examples of properly learned tasks. Given that theoretical boundaries were derived, I would have loved more experiments investigating these limits. For example something like Figure 1 but with a degradation for higher length. The same goes for Section 6.2. What I am missing is a clear connection between model complexity and performance.
> >Lack of comparison to other methods: I think it would be helpful to see how typical diffusion models, including Unet-based architectures, perform on the task of learning Gaussian processes. Clear results could help in shifting the design towards transformer-based diffusion architectures.
>
> We thank the reviewer for the advice on experiments. We conducted new experiments with different sequence length (N) and model size (the number of transformer blocks in DiT) and also compare DiTs with UNets. Please see our **general response** and **Appendix A** in our updated manuscript for details.
>
> >Generalization to real-world problems: There are no experiments showing how well the insights generated by the experiments translate to real-world problems. You present the ball experiment but even there you have to consider how well your experiments align with the assumption of using Gaussian process data (cmp. l. 504-506). Real-world problems will often not follow Gaussian process dynamics
> Regarding the practical implication of our theory, please also see our general response.
> >It would be nice if you could state some take-home messages for the reading. Ideally some general information such as design choices for the model architecture.
>
> Our take-home message is that diffusion transformers capture spatial-temporal correlation in sequential data through its unique attention mechanism. For the guidance on model architecture, as shown in our theory, when learning sequential data with length $N$ and a decaying temporal dependency,  it is suggested to equip the diffusion transformers with **$O(\log N)$ transformer blocks** for the best performance as predicted by our theory and experiments. Moreover, according to the roles of each components in transformers, if we want to improve the temporal correlations among tokens, we may try making more emphasis on the attention block (such as raising the latent dimension of **query and key weight matrices**), when we want to improve the single-frame quality, we may try emphasizing the **value weight matrices** and the **residual blocks**, or improving the latent encoder outside the DiT architecture. Also, our truncation technique (described in Figure 2) shows that we may impose  a **window-limited attention** with window length depending on the temporal correlation to reduce inter-timestep computation without losing too many temporal dependencies.
>
> >The learned temporal dependencies presented in Figure 5 look very similar to what you get when you just plot the attention between two uncorrelated matrices with positional embedding (l. 300). What is the difference between these maps and the ones you present, e.g., in Fig. 5
>
> We remark that the decay of the temporal dependencies can be  different from that of two uncorrelated matrices because our theory is applicable to general $\ell$ and $\nu$. In the training process, we actually observe that the temporal dependencies of generated samples are gradually adapting to the temporal dependencies of the truth data. Please see **Figure 9 in Appendix A** for details.

---

> ### Author Response · Authors · 2024-12-02
> **Thank you for your time and effort in reviewing our paper.**
>
> Dear Reviewer zotT,
>
> Thank you for your valuable feedback on our paper. We believe our revisions have fully addressed your concerns and significantly improved our work.
>
> If you have any additional questions, we're available for discussion.
>
> Your insights have been instrumental in enhancing our paper. Thank you again for your thorough review.
>
> Best regards,
>
> The Authors

---

### Official Review · Reviewer_fqSV · 2024-11-04

**Soundness:** 3
**Presentation:** 3
**Contribution:** 3
**Rating:** 8
**Confidence:** 4

**Summary:**

This paper scrutinises diffusion transformers under a simplified setting where the input time series is assumed to follow a Gaussian process. The paper carefully derives the analytical forms of the dynamics and provides score approximation as well as data-distribution estimation guarantees in the simplified setting of interest. The authors cast this paper as a first step towards building more sophisticated theories.

**Strengths:**

I really like the framing in this paper. While sounding like toy settings, GPs can be quite general and could be a good starting point of analysis. I especially appreciate the analytical forms, for example in the score approximator, and how the covariance function relates to positional embeddings. The paper is also quite well written. I have gone through parts of the proofs and they made sense - although I have not exhaustively verified every step.

**Weaknesses:**

- Can we have a section in the appendix where all the assumptions and simplifications are listed? This would really ease the conclusions that the reader can draw from this paper.
- Ln. 397: What's a "properly transformer" architecture?
- Could the authors give more elaborations on what motivates the choices of $\epsilon$ and $T$ in Thm. 2?
- What about empirically testing the learning on GPLVMs? Wouldn't this be a natural fit?
- There are some duplicate looking sections in the appendix, for example relating to the size of the transformer blocks. Can't we gather those in a big table instead of repeating them after each relevant section?

**Questions:**

Please see weaknesses.

---

> ### Author Response · Authors · 2024-11-22
> **Response to Reviewer fqSV**
>
> We thank the reviewer for the valuable comments and suggestions. We would appreciate it if you could champion our paper during the discussions!
>
> >Can we have a section in the appendix where all the assumptions and simplifications are listed? This would really ease the conclusions that the reader can draw from this paper.
> >There are some duplicate looking sections in the appendix, for example relating to the size of the transformer blocks. Can't we gather those in a big table instead of repeating them after each relevant section?
>
> Thanks for the suggestion. We have added a short paragraph in **Appendix C** to list assumptions and simplifications we have made. We have also presented the network sizes using tables at corresponding places. The overall network size is listed in Table 3, which combines the results from Tables 1 and 2.
> >Ln. 397: What's a "properly transformer" architecture?
>
> By proper transformer architecture, we refer to the constructed transformer $\mathcal{T}(D, L, M, B, R_t)$ in Theorem 1. With this choice, we establish sample complexity in Theorem 2. Note that, the configuration parameters in $\mathcal{T}$ are chosen dependent on the sample size, ensuring that the transformer is expressive enough to accurately represent the score function and succinct meanwhile for efficient statistical learning.
> >Could the authors give more elaborations on what motivates the choices of ϵ and T  in Thm. 2?
>
> We choose $\epsilon$ depending on the sample size due to **bias-variance trade-off** when learning the score function. We note from the first inequality at the beginning of the Proof of Proposition 2. that $\epsilon$ is the bias (transformer approximation error) and all the remaining terms are statistical variance when learning the score function using $n$ i.i.d. samples. Given $\epsilon = \epsilon_\delta$, we observe that smaller $\epsilon$ leads to large variance and large $\epsilon$ leads to small variance but large bias. Therefore, we optimally choose $\epsilon$ so that the bias and variance have the same order of dependence on sample size $n$.
> The choice of $T$ is relevant to the **forward mixing error** in inequality $(17)$. As we terminate the forward process at a finite time $T$, the terminal distribution is not exactly standard Gaussian. However, as long as $T$ is sufficiently large, the terminal distribution is close to Gaussian and we show that the convergence speed is **exponential**. Therefore, it suffices to choose $T = \log n$ to match the $1/\sqrt{n}$ statistical error.
> >What about empirically testing the learning on GPLVMs? Wouldn't this be a natural fit?
>
> From our understanding (if correct), GPLVM is a multi-output regression model which serves as  a dimension reduction tool. However, GPLVM assumes the data are sampled from one Gaussian process, while we are considering learning and generating the whole Gaussian process. Could you provide more information on how to empirically test the learning on GPLVMs if possible? Thanks!

---

> > ### Comment · Reviewer_fqSV · 2024-11-22
> >
> > I thank the authors for the response and will keep my rating. GPLVM is a generative method where each observed dimension of the data is assumed to be generated by a separate GP (latents are shared). So no it's not a single Gaussian. In this work you also make the simplification that the data comes from a GP. Hence, I thought GPLVM would also be a natural fit to study. Said differently, maybe one can study the case where the data comes from a GPLVM or a similar case where each column of data is generated by a GP.

---

> ### Author Response · Authors · 2024-11-22
> **Response**
>
> Thank you for supporting our work and explaining the GPLVM!
>
> We think GPLVM could be a viable tool for modeling the Gaussian process data in our work, given its flexibility to learn low-dimensional representation of high-dimensional sequences. We note that GPLVM would work differently compared to diffusion models, where the latter capture the sequential data distribution by estimating the score function. Given stacked sequential data $X \in \mathbb{R}^{d \times N}$, GPLVM aims to find a mapping $f$ that can transform a low-dimensional latent Gaussian process data $Z \in \mathbb{R}^{q \times N}$ to match the distribution of $X$, where $q \ll d$. The training of GPLVM is often likelihood based. Our understanding is that GPLVM works similarly to variational autoencoder, with the ability to handle sequential data.
>
> It is a very interesting direction to consider the observed sequential data being generated by a GPLVM. In this case, the score function may be more complicated and involve the differentiation of the mapping $f$. We believe that the temporal and spatial correlation should still be learned by the attention mechanism in a DiT as long as $f$ assumes certain smoothness regularity. A rigorous analysis requires future investigation though.

---

### Official Review · Reviewer_XNMz · 2024-11-05

**Soundness:** 2
**Presentation:** 3
**Contribution:** 2
**Rating:** 5
**Confidence:** 4

**Summary:**

This paper explores transformers as denoisers within diffusion models for generating spatiotemporal data, motivated by recent advancements in video foundation models, such as SORAH.
The approach begins by assuming a Gaussian process as the underlying data distribution and simplifies it to a scale-varying Gaussian process where the covariance changes only in scale over time. In this framework, the score function is related to the inverse of a modified covariance matrix. Since directly calculating this inverse is computationally prohibitive for high-dimensional latent vectors (dimension = d * N), the paper introduces a convex quadratic relaxation solvable via gradient descent. The authors then design a transformer architecture inspired by unrolled gradient descent steps. They also derive sample complexity bounds for the transformer’s ability to learn the covariance based on the smoothness and decay properties of the kernel matrix.
The paper includes basic experiments with synthetic Gaussian process data and a simple ball-motion scenario to validate the proposed method.

**Strengths:**

- The paper addresses a timely and relevant problem, aiming to understand why spatiotemporal transformers effectively learn distributions in spatiotemporal data.

- The writing is clear, making the methodology and findings accessible.

**Weaknesses:**

- The goal of the paper lacks clarity. If the aim is to demonstrate the expressiveness of transformers for correlated or spatiotemporal data, this doesn’t necessarily relate to diffusion models and could be examined independently in a more simplified setting.

-The assumption of a Gaussian process with scale-wise stationarity is a strong simplification, limiting the applicability to realistic distributions. Consequently, the results offer limited insights into transformer design for broader, more complex distributions.

-The experiments are weak and unconvincing. To substantiate the theory, more realistic data should be used, and comparisons with alternative architectures, such as UNet (3D or 2+1D), known for capturing spatiotemporal correlations, would strengthen the argument.

**Questions:**

- In the quadratic formulation in Equation (3), the score function estimation remains non-separable over timesteps (i.e., across x1,t, ..., xN,t). Would further decomposition allow for a distributed solution with minimal inter-timestep communication?

- Prior work, such as [Sahiner et al., 2022], has already shown that attention layers capture correlations and reflect low-rank priors, so it’s unsurprising that attention proves effective for correlated and stationary data.

Sahiner, A., Ergen, T., Ozturkler, B., Pauly, J., Mardani, M., & Pilanci, M. (2022, June). Unraveling attention via convex duality: Analysis and interpretations of vision transformers. In the International Conference on Machine Learning (pp. 19050-19088). PMLR.

---

> ### Author Response · Authors · 2024-11-22
> **Response to Reviewer XNMz**
>
> We thank the reviewer for the valuable comments and suggestions.
>
> >The goal of the paper lacks clarity. If the aim is to demonstrate the expressiveness of transformers for correlated or spatiotemporal data, this doesn’t necessarily relate to diffusion models and could be examined independently in a more simplified setting.
>
> The goal of our paper is to **theoretically** investigate how diffusion transformers capture spatial-temporal dependencies for efficiently **learning sequential data distributions**. This is the core question raised in the introduction section on Page 2. To answer the question, we establish a sample complexity bound in Theorem 2, which relies on our developed transformer approximation theory for the score function in Gaussian process data. In other words, our efficient learning guarantees are a consequence of the **modeling power of both diffusion processes and transformers**.
> We acknowledge that transformers are known to be universal approximators in the existing literature, however, our study presents a very **different perspective**. In diffusion models, learning data distributions is conducted by estimating the score function, which is distinct from using a transformer directly for sequential data modeling in an autoregressive manner. To be more specific, for Gaussian process data, we closely examine the structures in the score function, and show that transformers can unroll a gradient descent algorithm for approximating the score function. This result is highly relevant in the context of diffusion models, yet goes **beyond the conventional study of expressiveness of transformers**.
>
> >The assumption of a Gaussian process with scale-wise stationarity is a strong simplification, limiting the applicability to realistic distributions. Consequently, the results offer limited insights into transformer design for broader, more complex distributions.
>
> Please see our **general response** for a discussion about the assumption of Gaussian process and the practical implications of our theoretical results. Our paper is **theory driven** and derives the first sample complexity bound for diffusion transformers in learning Gaussian process data. The data assumption aims to provide clear theoretical insights while maintaining technical cleanness. In particular, under the assumption, we precisely characterize the spatial and temporal dependencies and show how these dependencies are learned in a transformer by performing algorithmic unrolling. Such theoretical insights generalize to broader and more complex distributions such as time series and kinetic dynamics, as long as the two essential ingredients are established: 1) the score function of the distribution is representable by an efficient algorithm, and 2) a transformer can approximate each iteration of the algorithm. Yet, rigorous theoretical analysis requires further investigation beyond the current paper.
>
> For sequential data modeling, Gaussian processes are **used frequently** [3, 4]. Meanwhile, we have discussed the relevance of our results and assumptions to real applications in the last paragraph on Page 3. Our study aligns with the commonly used latent diffusion models, where raw data will be transformed into a low-dimensional embedding, presumably close to Gaussian [1, 2].
>
> We recognize the importance of transformer architecture design for better sequential data modeling. We believe that understanding how transformers represent spatial-temporal dependencies is the **first step towards a principled answer to architectural innovation**.
>
> >The experiments are weak and unconvincing. To substantiate the theory, more realistic data should be used, and comparisons with alternative architectures, such as UNet (3D or 2+1D), known for capturing spatiotemporal correlations, would strengthen the argument.
>
>
> We conducted new experiments on Unets, where we find that DiT  outperforms Unets in capturing spatial-temporal correlations especially for long-horizon dependencies. Please see our **general response** and **Appendix A**  in our updated manuscript for the details of our experimental results.

---

> ### Author Response · Authors · 2024-11-22
> **Response part 2**
>
> >In the quadratic formulation in Equation (3), the score function estimation remains non-separable over timesteps (i.e., across x1,t, ..., xN,t). Would further decomposition allow for a distributed solution with minimal inter-timestep communication?
>
> Although the score function estimation remains non-separable, we remark that our truncation technique (described in Figure 2) actually serves as a tool to reduce inter-timestep communication, which is similar to adding a **sliding window in the attention block** that deactivate the interaction between tokens that have  small correlations.
>
> >Prior work, such as [Sahiner et al., 2022], has already shown that attention layers capture correlations and reflect low-rank priors, so it’s unsurprising that attention proves effective for correlated and stationary data.
>
> We thank the reviewer for mentioning this work. Although both [Sahiner et al., 2022] and our work aims to interpret attention mechanism, the setting and viewpoint are different. We make emphasis on learning **sequential data** with transformer-based diffusion models, and leverage an algorithm-unrolling strategy for approximation and generalization theory, while  [Sahiner et al., 2022]  makes emphasis on vision transformers and provides analysis in a more optimization-based viewpoint. We will add a discussion of  [Sahiner et al., 2022]  in the revised version of our paper.
>
> We appreciate the suggestions on the presentation of our work. We will emphasize the generality of our work and revise the confusing sentences for better presentation. We would be grateful if the reviewer could reconsider the evaluation based on our contributions. If you have any further questions, please do not hesitate to comment.

---

> > ### Author Response · Authors · 2024-11-25
> > **Thank you for your time and effort in reviewing our paper.**
> >
> > Dear Reviewer XNMz,
> >
> > Thank you for your valuable feedback on our paper. We believe our revisions have fully addressed your concerns and significantly improved our work.
> >
> >
> > If you have any additional questions, we're available for discussion until November 26 (AOE). Otherwise, we would greatly appreciate it if you could reconsider your score based on our revisions.
> >
> >
> > Your insights have been instrumental in enhancing our paper. Thank you again for your thorough review.
> >
> >
> > Best regards,
> >
> > The Authors

---

> ### Author Response · Authors · 2024-12-02
> **Thank you for your time and effort in reviewing our paper.**
>
> Dear Reviewer XNMz,
>
> Thank you for your valuable feedback on our paper. We believe our revisions have fully addressed your concerns and significantly improved our work.
>
> If you have any additional questions, we're available for discussion. Otherwise, we would greatly appreciate it if you could reconsider your score based on our revisions.
>
> Your insights have been instrumental in enhancing our paper. Thank you again for your thorough review.
>
> Best regards,
>
> The Authors

---

### Author Response · Authors · 2024-11-22
**General response to all reviewers**

We would like to thank the reviewers for their valuable comments and suggestions. Here, we elaborate on the contributions of our theory and demonstrate how transformers efficiently learn spatial-temporal dependencies  through further experiments.

**1.The generality of our theory (not limited in our assumption)**

We agree with Reviewers XNMz and  zotT that directly assuming real-world data to follow a Gaussian process is a strong assumption. However, as discussed in our paper, our theory also applies to **latent diffusion models** (e.g., [1], [2]), where an autoencoding model maps real-world data to a low-dimensional latent space, making the latent distribution more suitable for score-based generative models. With this autoencoder, the distribution of the low-dimensional representation could be simpler, so the Gaussian process assumption is milder in this case. Additionally, some variants of VAEs perform inference on a latent Gaussian process model, assuming the prior distribution of the latent representation follows a Gaussian process for convenience in sequential data modeling (such as [3] and [4]). These works align well with our assumptions.

Moreover, our theoretical insights and analytical tools developed in the paper can be **extended to broader applications**. Firstly, we theoretically elucidate **the role of each transformer block** (see the discussion following Theorem 1 and Figure 3) in learning the score function of sequential data. These observations are strongly corroborated by the experiments in the paper (Section 5). Secondly, our **algorithm unrolling theory** of diffusion transformers applies as long as there is an efficient algorithm for computing the score function and each iteration of the algorithm can be implemented by transformer blocks. Therefore, our analytical tools can be generalized to time series, kinetic dynamics, etc.

**2.Further experiments on DiTs and comparison with UNets**

Please see **Appendix A** for additional numerical results. We summarize our major discoveries.

a) **Comparison with UNet-based diffusion**

We implement a UNet-based diffusion model with a similar size as the DiT to learn Gaussian Process data. After training, we independently generate samples to test the performance. We show that the spatial-temporal correlation generated by the DiT is close to the ground truth. While the UNet-based diffusion model struggles in capturing the spatial dependencies and exhibit clear inaccuracy for learning temporal dependencies.

b) **Sequence length and transformer size v.s. performance**

We examine the influence of the sequence length $N$ and the number of transformer blocks $L$ on the performance of DiT for learning Gaussian process data. We demonstrate that the estimation error increases mildly as the sequence length increases, which aligns with our $\sqrt{N}$-dependence in Theorem 2. We also show that the performance of DiT improves as the number of transformer blocks increases, yet at a marginally diminishing speed if $L$ is sufficiently large. This supports our construction of a transformer with $O(\log N)$ blocks.




[1] Blattmann et, al. “Stable video diffusion: Scaling latent video diffusion models to large datasets”

[2] Wang, et,al. Lavie: “High-quality video generation with cascaded latent diffusion models”

[3] Casale, et al. “Gaussian Process Prior Variational Autoencoders”, NeurIPS 2018.

[4] Fortuin, et al. “GP-VAE: Deep Probabilistic Time Series Imputation”, AISTATS 2020.

---

### Meta-Review · Area_Chair_HRkX · 2024-12-15

**Metareview:**

The paper develops a theoretical framework to explain how diffusion transformers (DiTs) capture spatial-temporal dependencies in sequential data, using Gaussian processes as a simplified model. It introduces a novel score function approximation and establishes sample complexity bounds, supported by mathematical proofs and experiments on synthetic Gaussian process data and semi-synthetic tasks. Strengths include a strong theoretical contribution to understanding transformers, well-founded mathematical derivations, and high-quality visualizations. Weaknesses include the reliance on Gaussian process assumptions, limited practical experiments on real-world data, and insufficient comparisons to alternative architectures like UNets. Missing elements include more diverse datasets and guidelines for architectural design beyond theoretical insights. The paper’s significant theoretical contributions and clear presentation justify acceptance, with suggestions to expand empirical validation and real-world applicability in future work.

**Additional Comments On Reviewer Discussion:**

During the rebuttal period, reviewers raised concerns about the reliance on Gaussian process assumptions, limited real-world experiments, and the need for comparisons to other architectures like UNets. They also requested clarifications on notation, assumptions, and the relationship between model complexity and performance. The authors addressed these by conducting new experiments comparing DiTs to UNets, analyzing training dynamics, and providing detailed clarifications on assumptions and theoretical bounds. They also added clearer notation and improved figure readability. While some reviewers noted the ongoing limitations of empirical validation, the authors’ comprehensive responses and additional experiments addressed major concerns, leading to a consensus for acceptance based on the paper’s strong theoretical contributions and improved clarity.

---

### Decision · Program_Chairs · 2025-01-22

Accept (Poster)